# Single-molecule epitranscriptomic analysis of full-length HIV-1 RNAs reveals functional roles of site-specific m⁶As

Alice Baek[1,2,3,12], Ga-Eun Lee[1,2,3,4,12], Sarah Golconda[1,2,3], Asif Rayhan[5], Anastasios A. Manganaris[4,6], Shuliang Chen[1,2], Nagaraja Tirumuru[1,2], Hannah Yu[1,2,3], Shihyoung Kim[1,2,3], Christopher Kimmel[2,4], Olivier Zablocki[7,8], Matthew B. Sullivan [7,8,9], Balasubrahmanyam Addepalli[5], Li Wu [10] & Sanggu Kim [1,2,3,4,11] ✉

Although the significance of chemical modifications on RNA is acknowledged, the evolutionary benefits and specific roles in human immunodeficiency virus (HIV-1) replication remain elusive. Most studies have provided only population-averaged values of modifications for fragmented RNAs at low resolution and have relied on indirect analyses of phenotypic effects by perturbing host effectors. Here we analysed chemical modifications on HIV-1 RNAs at the full-length, single RNA level and nucleotide resolution using direct RNA sequencing methods. Our data reveal an unexpectedly simple HIV-1 modification landscape, highlighting three predominant N⁶-methyladenosine (m⁶A) modifications near the 3' end. More densely installed in spliced viral messenger RNAs than in genomic RNAs, these m⁶As play a crucial role in maintaining normal levels of HIV-1 RNA splicing and translation. HIV-1 generates diverse RNA subspecies with distinct m⁶A ensembles, and maintaining multiple of these m⁶As on its RNAs provides additional stability and resilience to HIV-1 replication, suggesting an unexplored viral RNA-level evolutionary strategy.

RNAs are highly structured macromolecules with various post-transcriptional modifications, including 3' polyadenylation, splicing and chemical modifications. Since the late 1950s, more than 300 types of chemical modifications (epitranscriptomes) have been identified[1,2], adding another layer of complexity to RNA biology. These modifications control a wide range of cellular and viral processes and are associated with more than 100 human diseases[2,3]. Studying these modifications, however, has been slow and laborious due to technical limitations inherent in the sequencing of native RNAs[4].

Human immunodeficiency virus (HIV-1) has a substantially higher number of chemical modifications on its RNAs than typical cellular transcripts[5,6]. However, the evolutionary benefits and HIV-1-specific

[1]Center for Retrovirus Research, Ohio State University, Columbus, OH, USA. [2]Department of Veterinary Biosciences, Ohio State University, Columbus, OH, USA. [3]Infectious Diseases Institute, Ohio State University, Columbus, OH, USA. [4]Translational Data Analytics Institute, Ohio State University, Columbus, OH, USA. [5]Rieveschl Laboratories for Mass Spectrometry, Department of Chemistry, University of Cincinnati, Cincinnati, OH, USA. [6]Department of Computer Science and Engineering, Ohio State University, Columbus, OH, USA. [7]Center of Microbiome Science, Ohio State University, Columbus, OH, USA. [8]Department of Microbiology, Ohio State University, Columbus, OH, USA. [9]Department of Civil, Environmental and Geodetic Engineering, Ohio State University, Columbus, OH, USA. [10]Department of Microbiology and Immunology, Carver College of Medicine, University of Iowa, Iowa City, IA, USA. [11]Center for RNA Biology, Ohio State University, Columbus, OH, USA. [12]These authors contributed equally: Alice Baek, Ga-Eun Lee. ✉e-mail: kim.6477@osu.edu

roles of these modifications in viral replication and various RNA functions remain unclear and sometimes even controversial, showing both pro- and anti-viral effects depending on the virus type, replication stage or tested cell type[3,7–11]. Most RNA modification studies so far have relied on indirect analyses of the phenotypic effects of perturbing host effectors (known as writers, erasers and readers)[6,8,10,12–14], neglecting the potential site-specific and context-dependent roles of chemical modifications[15–19]. Although studies using short-read sequencing have mapped several common modifications onto the HIV-1 genome, including $N^6$-methyladenosine (m6A), 5-methylcytosine (m5C), 2′-O-methylation (Nm) and $N^4$-acetylcytidine (ac4C), they have provided only low-resolution and population-average values of modifications of a given type for fragmented RNAs[6,8,12–14]. The site-specific roles of individual modifications and their ensembles on the same RNA strand remain largely unknown.

Nanopore direct RNA sequencing (DRS) is a powerful tool that can analyse individual strands of native RNAs as they continuously pass through nanopores[20]. This unique technology allows for a simultaneous evaluation of key features of RNAs at the single molecule level, including RNA sequences, chemical modifications, splicing isoforms, 3′ polyadenylation and absolute quantitation and profiling of a heterogeneous pool of RNA transcripts[21–23]. DRS is also free from the experimental biases associated with current short-read sequencing methods[24]. However, DRS faces challenges when analysing long RNA molecules and RNAs that cause motor enzyme stalls[25–27] and when analysing chemical modifications at the single RNA level. In this Article, we present several technical innovations, including full-length DRS and read-level binary classification methods, which maximize the potential of DRS technology for the study of HIV-1 RNA biology. We found three dominant and site-specific m6A modifications on the 3′ end of the HIV-1 RNA genome and characterize their functional significance in regulating viral replication at the individual RNA level.

## Results

### Nanopore DRS of full-length HIV-1 RNA

DRS of long and complex RNA molecules, such as premature transcripts (13–18 kb), cellular Xist, mitochondrial messenger RNAs, and plant and virus RNAs, has been challenging[25–28]. In our study, initial conventional DRS procedures failed to generate more than 13 reads (0.01% recovery) of full-length HIV-1 sequences in eight out of nine runs (Fig. 1a,b and Extended Data Fig. 1). Given that HIV-1 RNAs are 2–30 times more modified than typical cellular mRNAs[5,12] and have complex secondary and tertiary structures[29,30]—features known to stall reverse transcription[24,31](RT), an optional step known to improve the DRS throughput[20,27]—we reasoned that inefficient linearization of HIV-1 RNA by RT might be the cause of the failure. To alleviate this, we established a multiplex RT using oligonucleotide primers specific to different parts of the HIV-1 genome to improve full-length DRS (Fig. 1a,b and Extended Data Fig. 1). Under optimized conditions using 111 different primers, we generated a total of 810, 1,797 and 2,655 reads of full-length unspliced (US; ~9 kb), partially spliced (PS; ~4 kb) and

completely spliced (CS; ~2 kb) HIV-1 RNAs, respectively, as well as a total of 3,985 reads of full-length virion RNAs (Supplementary Table 5). The improved DRS with multiplex RT enabled a comprehensive analysis of individual reads of HIV-1 RNAs in virions and in virus-producing cells.

### The modification landscape reveals site-specific m6As

Previous mass spectrometry studies have estimated that approximately 80–200 modifications of various kinds exist per HIV-1 RNA genome[5,12]. The locations and the functions of site-specific modifications remain unclear due to the challenges to identify their precise locations. To identify site-specific modifications on a whole-genome scale, we used a two-step signal-refinement process that involved the use of in vitro transcribed (IVT) HIV-1 RNAs as a non-modified RNA control (Methods). With our optimized conditions, the Tombo analysis[32] generated highly reproducible per-read modification (P value) signals in our repeated experiments (Extended Data Fig. 2). The results from Tombo and other tested software tools, including Eligos2 (ref. 33), Nanocompore[34] and xPore[35], consistently revealed a small number of prominent modification signals on the 3′ end of the HIV-1 genome (Fig. 1c and Extended Data Fig. 3). These prominent signals probably point to site-specific modifications; the signals from non-site-specific modifications are diluted in these population-based analyses (Fig. 1c).

To identify the most notable and consistent site-specific modifications, we compared the modification signals generated by three different tools—Tombo, Eligos2 and Nanocompore, each analysing different aspects of DRS signals, such as ionic current levels[32], base-calling error rates[33] and dwell time[34,36]. Given these tools can detect various kinds of chemical modifications, we selected the top 149, 167 and 156 modification signals, respectively, from these tools and cross-compared them (Methods and Supplementary Fig. 7). Despite the high reproducibility of all three tools (Extended Data Fig. 3), we identified only seven common peaks, reflecting the variable detection efficiencies when using different DRS signal features[36]. Notably, among the seven peaks, five were located at or adjacent to the known m6A motifs (DRACH: D, A/G/U; R, A/G; H, A/C/U)[37]. One DRACH peak at position A17 was excluded because we found the signals near the end of the reads to be inherently unstable (Methods and Supplementary Fig. 4). The remaining four DRACH sites (A8079, A8110, A8975 and A8989) consistently exhibited strong modification signals across all our tests (Fig. 1c and Supplementary Figs. 5 and 6) and were highly conserved among HIV-1 subtype B in the HIV database (the Los Alamos National Laboratory database; Fig. 1e), indicating the importance of these sites in circulating viruses. These sites also coincided with the major m6A peaks in previous short-read sequencing studies (Extended Data Fig. 4)[6,8]. Similar modification signals were also observed in HIV-1-infected CD4+ T cells (Extended Data Fig. 4d). Considering 242 DRACH sites present in the HIV-1 genome, the predominant modification signals in these few DRACH sites suggest their strong site specificity in terms of m6A installation or its functions.

Moreover, the 25 common modification peaks detected by both Tombo and Eligos2 were also significantly enriched near the DRACH

**Fig. 1 | DRS of full-length HIV-1 RNA points to the site-specific function of m6As. a**, A schematic view of multiplex RT. The polyadenylated RNAs were selectively sequenced using a RT adaptor (RTA). **b**, Read length distribution. Unlike the conventional methods, our multiplex RT with 111 oligos enabled a consistent and reproducible recovery of full-length HIV-1 RNA. The arrowheads denote 9.2 Kb virion RNA (left) and intracellular US, PS and CS HIV-1 RNAs (right). Rep1–3 (left) and rep1–4 (right) denote repeated experiments using independently prepared samples. **c**, A total of 3,985 full-length virion RNA reads, 5,411 reads of IVT RNAs (canonical control) and 1,450 IVT RNA subreads (baseline control) were analysed using Tombo-MSC, Eligos2 and Nanocompore. 'd values' from Tombo-MSC, 'odd ratios' from Eligos2 and absolute values (ABS) of 'logit LOR score' from Nanopore are shown. Site-specific modifications exhibit a robust and distinct signal, while the signals from non-site-specific modifications

are diluted in these population-level analyses. **d**, To identify modification signals common in the three analyses, the top 149 peaks of Tombo-MSC (d value >0.05 based), 167 of Eligos2 data (odd ratios >2.4) and 156 peaks from Nanocompore results (logit LOR score >0.73) were cross-compared (Supplementary Fig. 7). A total of 25 signal peaks common in Tombo-MSC and Eligos2 (open circles) and seven peaks common in all three analyses (purple circles) are shown. Crosses denote DRACH sites. The 25 common sites are significantly enriched in DRACH sites (Methods). **e**, The magnified HIV-1 genome from 7,916 to 9,172 of HIV-1 genome (NL4-3 strain) and sequence logo plots for circulating HIV-1 (Los Alamos Database; https://www.hiv.lanl.gov/) are shown. Two adjacent sites (8,975 and 8,989) were located immediately upstream of the G-quadruplexes (G4s) in the U3, and the m6A at 8,079 and 8,110 are located immediately upstream of the potential G4 within the rev/env region downstream of the A7 splicing acceptor.

sites (Fig. 1d) and in the m⁶A-reader binding sites (Supplementary Table 4)[6,8]. In contrast, we did not find any notable associations between these common peaks and other modifications, such as the Nm, m⁵C or

ac⁴C sites[12,13,38] (Extended Data Fig. 5a). Given mass spectrometry estimates a large number of chemical modifications on the HIV-1 genome—particularly, Nm, m⁵C and ac⁴C are several-fold more frequent or at least

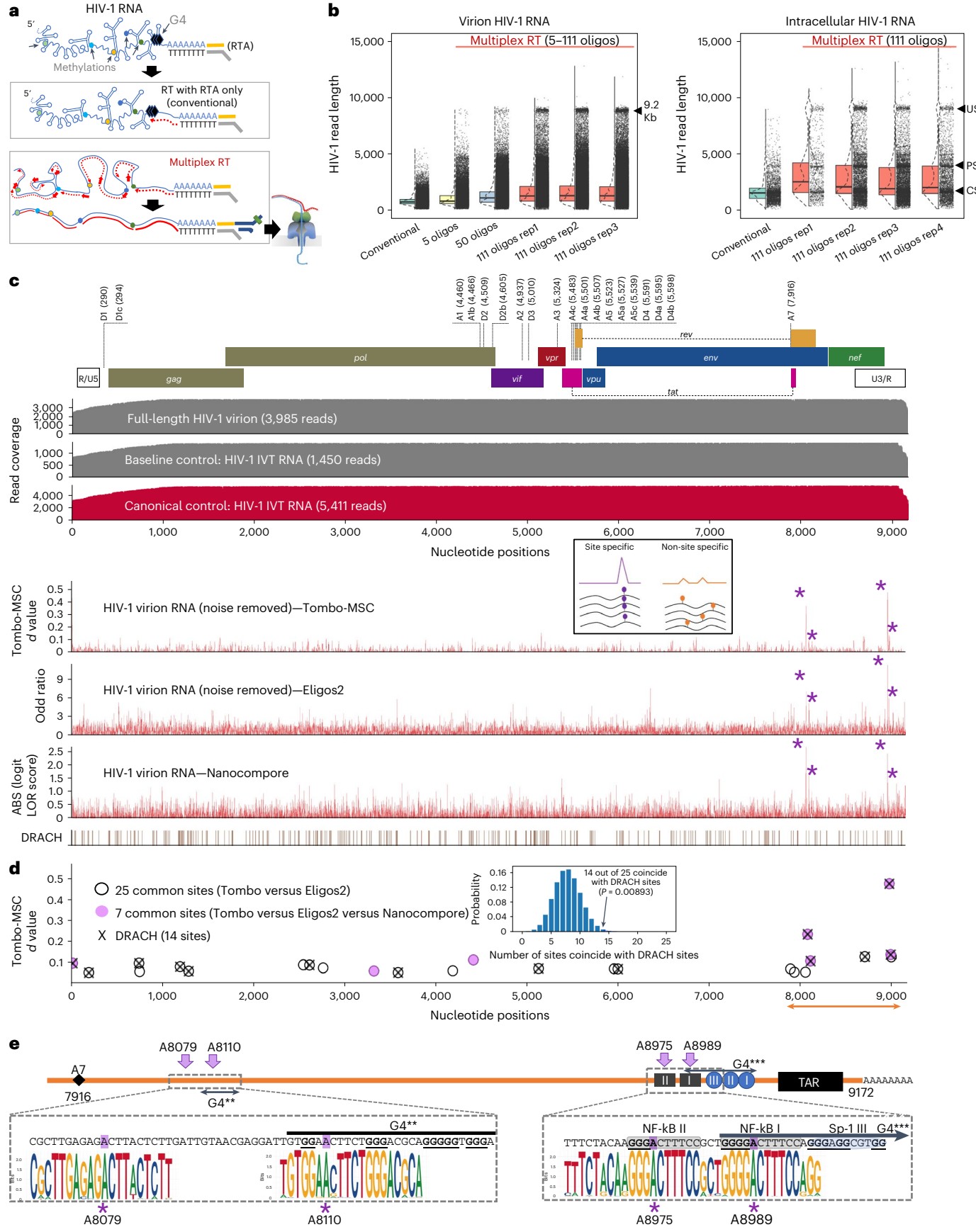

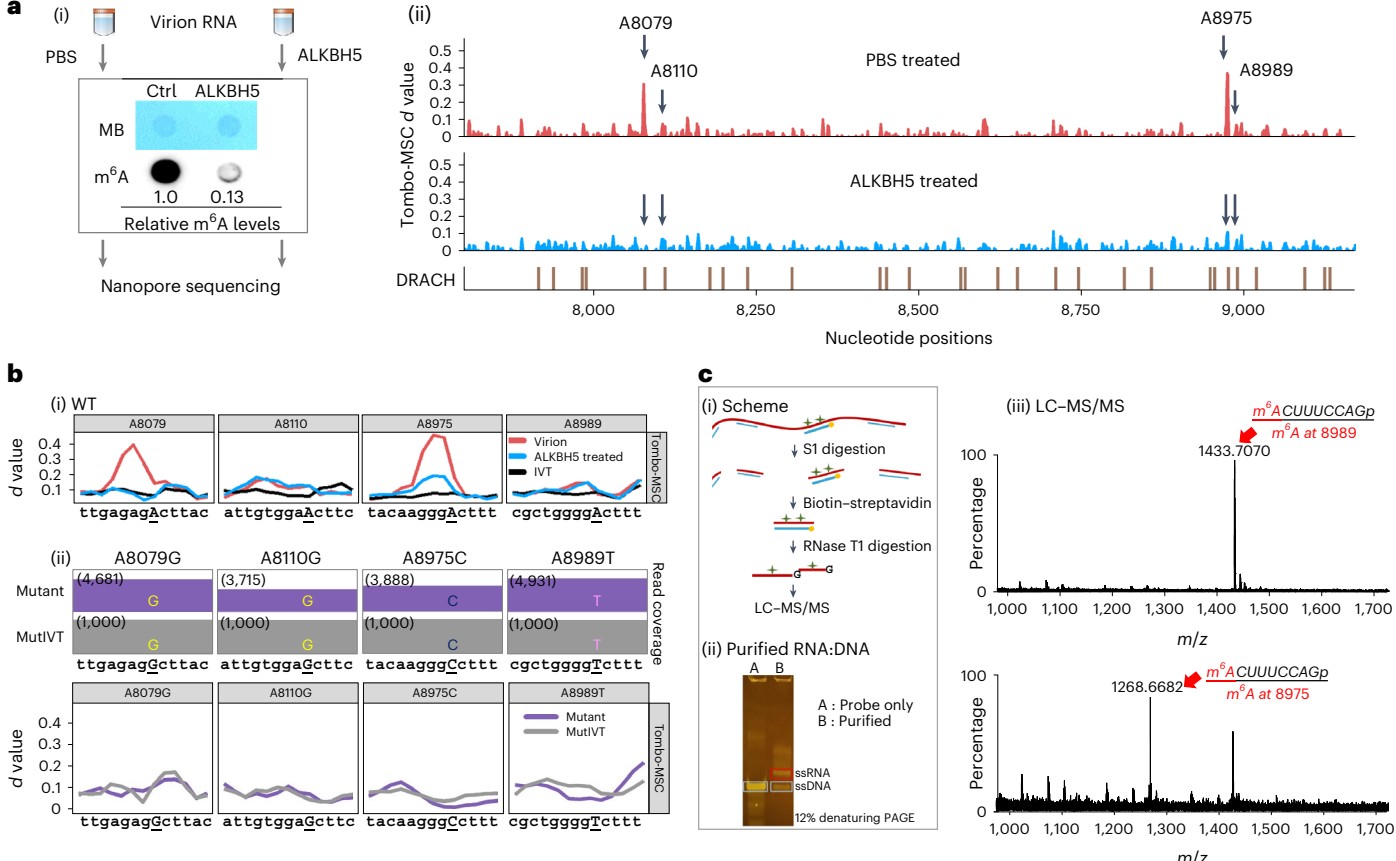

**Fig. 2 | Confirming dominant m⁶As on HIV-1 RNA at the single nucleotide resolution. a**, ALKBH5 treatment reduced m⁶A signals of HIV-1 virion RNA. A schematic view of ALKBH5 treatment is shown in (i). An immunoblot assay showed an 87% reduction in m⁶A signals following ALKBH5 treatment (i). DRS Tombo-MSC $d$ values comparing ALKBH5-treated (blue) and non-treated (PBS; red) RNAs are shown in (ii). **b**, Site-directed mutagenesis eliminated modification signals. Tombo $d$ values ($y$ axis) comparing WT (red), ALKBH5-treated WT RNAs (blue) and WT IVT RNAs (black) are shown in (i). Tombo analysis of four mutant NL4-3 RNAs, including A8079G, A8110G, A8975C and A8989T, using IVT controls

with the same mutations is shown in (ii). Top: the read depth of mutant HIV-1 RNA (purple bars) and IVT controls (MutIVT; grey bars). All mutants showed effective removal of modification signals. **c**, Oligonucleotide LC–MS/MS using RNase T1. A schematic view of RNA sample preparation is shown in (i). The target RNA fragments are purified using biotinylated DNA probes and subjected to RNase T1 digestion and LC–MS/MS. Purified target RNAs and DNA probes are shown in a 12% denaturing polyacrylamide gel electrophoresis (PAGE) (ii). Oligonucleotide LC–MS/MS confirmed adenosine methylations at 8,989 (top) and 8,975 (bottom) (iii). MB, methylene blue; Ctrl, untreated control.

as common as m⁶As[5,12]—it is notable that there are only three to four high stoichiometry modification sites, while all other modifications are either undetectable or barely above the detection threshold in these population-level analyses (Fig. 1c,d). These results suggest that Nm, m⁵C and ac⁴C modifications are generally less site-specific than m⁶As.

**Confirmation of m⁶As at the single-nucleotide resolution**

To evaluate the existence of the four most probable m⁶As, we first analysed HIV-1 virion RNA after in vitro treatment with an m⁶A eraser, ALKBH5 (ref. 39). All of the four m⁶A sites showed varying levels of signal reduction in the Tombo, Eligos2, Nanom6A[40] and dwell-time[41] analyses (Fig. 2a and Extended Data Fig. 5b). Next, we introduced a point mutation to each of the four most probable m⁶A sites (A8079G, A8110G, A8975C and A8989T) of an HIV-1 pro-virus plasmid (pNL4-3). All mutants showed a complete absence of modification signals when compared with IVT RNAs with identical mutations (Fig. 2b). Last, we confirmed the two pro- spective m⁶As at positions A8975 and A8989 by oligonucleotide liquid chromatography coupled to tandem mass spectrometry (LC–MS/MS) (Fig. 2c and Supplementary Fig. 8)[42]. As significant m⁶A modification signals were consistently observed at the A8079 site in all our tests, we selected A8079, A8975 and A8989 for further investigation of their site-specific roles. DRS of synthetic oligonucleotides with m⁶A at A8079, A8975 or A8989 further supports the presence of m⁶As at these three

sites (Fig. 5a(ii)). Despite multiple attempts, we could not confirm m⁶A methylation at A8110 by LC–MS/MS due to the insufficient enrichment of RNA fragments containing A8110. Read-level quantification assays, including m6Anet and Nanom6A (Supplementary Fig. 6), suggest that m⁶A at A8110 is a relatively low stoichiometry methylation.

**Knocking out all three m⁶As affects HIV-1 fitness**

The functions of m⁶A modifications are determined by host effectors that catalyse, recognize and remove such modifications (known as writ- ers, readers and erasers, respectively)[3,43]. Mounting evidence also sug- gest that these modifications have site-specific and context-dependent roles, controlling local RNA–protein interactions by modulating the RNA structures where the interactions occur[15–19]. m⁶As play impor- tant roles in regulating various aspects of RNA biology, including RNA structure, splicing, translation, metabolism and translocation within cells and promote HIV-1 replication in general[3,44]. However, our current understanding is primarily based on indirect analyses of the phenotypic effects of perturbing m⁶A writers, readers or erasers in host cells, which overlook the potential site-specific roles of the modifications. Some findings remain controversial, showing inconsistent results depending on the replication stages, cell types and assays used in the studies[3,44].

To directly analyse the functions of m⁶As on HIV-1 RNA, we gen- erated m⁶A-knockout viruses using site-directed mutagenesis and

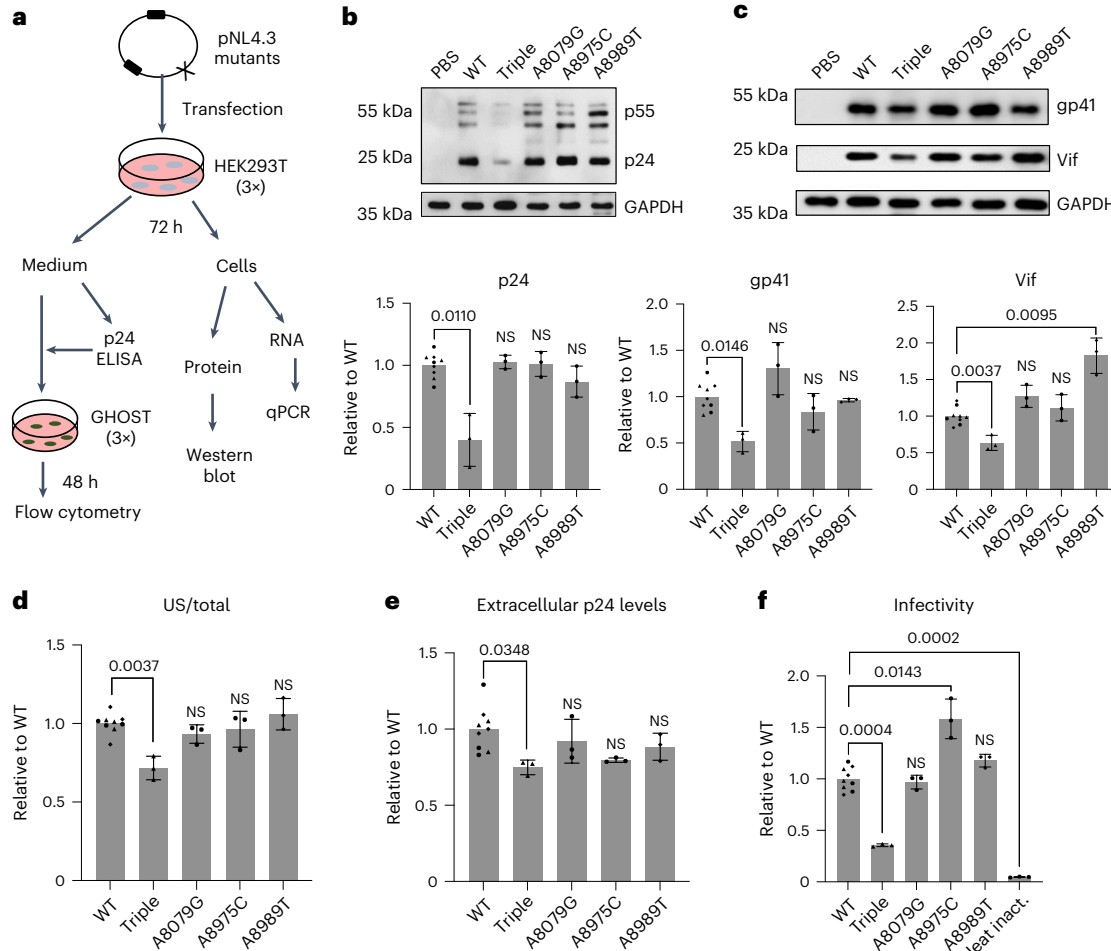

**Fig. 3 | Knocking out all the three dominant m⁶As on HIV-1 RNA, but not the single m⁶A, affects viral fitness. a**, A schematic view of experimental procedures. qPCR, quantitative PCR. **b**,**c**, Intracellular Gag protein (**b**) and gp41 and Vif expression (**c**) were significantly reduced by the triple mutation ($P = 0.0110$, $P = 0.0146$ and $P = 0.0037$, respectively), but not by the single mutations. Western blot results were quantitated by densitometry (bar charts below the gel images). WT results were set as 1. Triple mutant (Triple) and single mutants (A8079G, A8975C and A8989T) are shown in comparison. **d**, Triple mutants showed a significant reduction in US RNAs ($P = 0.0037$), while single mutants did not. **e**, p24 release into the medium was significantly reduced by triple mutations

($P = 0.0348$), but not by single mutations. **f**, An infection assay using an equimolar p24-containing medium showed a significant decrease in viral infectivity ($P = 0.0004$). The number of infected cells was determined by measuring GFP expression in GHOST reporter cells using flow cytometry. A heat-inactivated (heat inact.) HIV-1 was used as a negative control. A8079 is in a region where *rev* and *env* genes overlap. A8079G is silent for *rev* but changes glutamine to glycine at position 771 of gp41 (the cytoplasmic domain of envelope). A8975C and A8989T are in the U3. All the bar graphs in this figure are presented as mean values ± s.d. Two-tailed *t*-tests; $n = 3$ for triple (triangles), A8079G, A8975C (circles) and A8989T (diamonds); $n = 3$ for WT in each experiment (three experiments, total $n = 9$). NS, not significant.

evaluated key steps of viral replication in wild-type (WT) host cells (Fig. 3a). Although the m⁶As were effectively removed (Fig. 2b), none of the single mutations resulted in significant reductions in any of the tested replication steps, including total HIV-1 US RNA production, viral protein expression (Gag, Vif and envelope gp41), virion production (extracellular p24 levels) and infection of reporter cells (Fig. 3 and Extended Data Fig. 6). However, the triple mutation of all three m⁶A sites significantly reduced US RNA levels (Fig. 3d). HIV-1-infected CD4⁺ T cells (Jurkat) also showed similar reduction of US RNA (Extended Data Fig. 6b(ii)). It is known that the loss or reduction of US RNA results in a drastic reduction in viral fitness[45] because US RNAs are essential for producing the structural proteins (Gag/Gag-Pol) and genomic RNA. As expected, all subsequent steps, including p24 production, virion release and viral infectivity, were also significantly reduced (Fig. 3b,e,f).

**Triple m⁶A mutations induce an over-splicing phenotype**

Given the critical importance of RNA splicing in HIV-1 replication, particularly in controlling US RNA levels[45], we investigated the roles of the three m⁶As in HIV-1 alternative splicing (Fig. 4 and Extended Data Fig. 7).

HIV-1 produces over 50 different forms of spliced RNA, an extraordinarily high level of alternative splicing[46]. All HIV-1 RNAs are produced as a full-length initially and remain US (genomic RNA for virion packaging or mRNA for *gag/gag-pol*) or spliced into CS (major mRNA for *nef*, *rev* or *tat*) and PS (major mRNA for *vif*, *vpr* or *env/vpu*) (Fig. 4b). RNA modifications have been suggested to affect HIV splicing[3]. While DRS can effectively disentangle complex RNA isoforms[21,22,27], analysis of HIV-1 RNAs has been impractical due to poor full-length sequencing. Here, using the new multiplex RT method, we were able to reproducibly generate full-length reads of approximately 2 kb of CS, 4 kb of PS and 9 kb of US RNA, with recovery rates of 54.1%, 31.5% and 34.9%, respectively (Fig. 4a). We successfully assigned 94.8% of these full-length reads to 196 exon combinations, including 53 major isoforms[46], without any notable ambiguity (Fig. 4a). The read counts were generally consistent with the densitometric quantification of PCR amplicons of the CS and PS isoforms (Fig. 4c and Supplementary Table 6).

Regarding total HIV-1 RNA production, we observed no significant differences between the WT HIV-1 and triple mutants (Fig. 4d(i)). As expected from the molecular biology tests described above, the

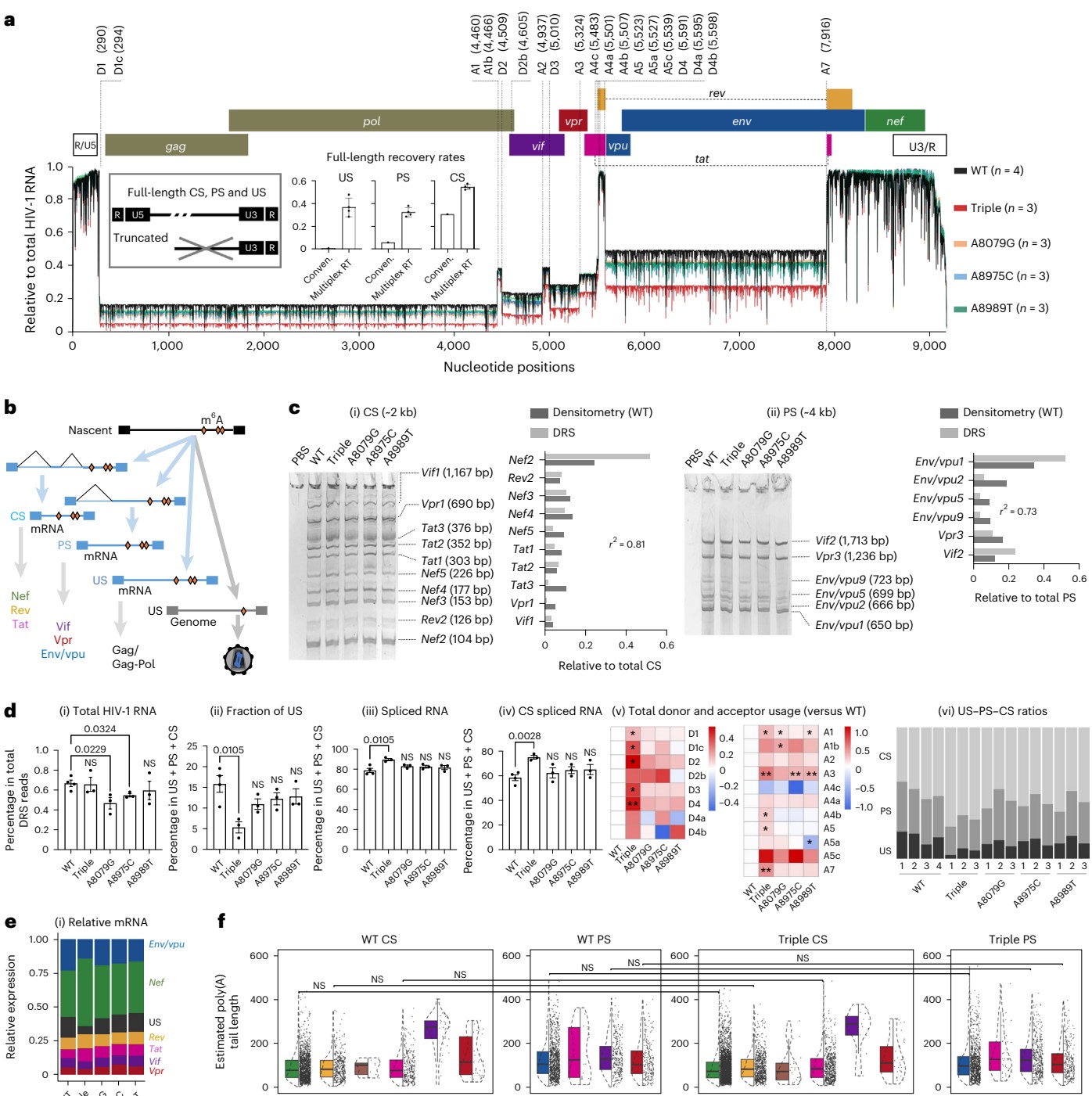

**Fig. 4 | The triple m⁶A mutation induces over splicing of HIV-1 RNA. a**, Full-length intracellular HIV-1 RNA were mapped onto the reference. A total of 94.8% of full-length reads were successfully assigned to 196 exon combinations without any notable ambiguity for splicing donors (D1–D4) and acceptors (A1–A7). The box plots show the full-length recovery rates by conventional (conven.) and multiplex RT methods. **b**, A schematic view of HIV-1 RNA production. **c**, Absolute counting of DRS data showed a general agreement with the densitometry quantification of RT–PCR amplicons of CS ($r^2 = 0.81$ for most prominent bands) (i) and PS ($r^2 = 0.73$ for six most prominent bands) (ii) RNAs. WT HIV-1 DRS data ($n = 4$) were combined into a single dataset to quantitate individual isoforms. **d**, While total HIV-1 RNA remained at similar levels (i), the total US RNAs were significantly reduced in triple mutant-producing HEK293T cells (ii). Data are presented as mean values ± s.d. $P = 0.0105$; two-tailed $t$-test; WT, $n = 4$; triple, A8079G, A8975C and A8989T: $n = 3$; biologically independent samples (ii). The fractions of total spliced RNAs (based on the D1 and D1c usage; $P = 0.0105$) (iii)

and CS RNAs (based on the A7 usage; $P = 0.0028$) (iv) were significantly higher than WT. Donor and acceptor usage rates (shown in $\log_2$ scale heat map; *$P < 0.05$ and **$P < 0.01$, Student's $t$-test) are generally higher in triple mutant-producing cells than those in WT-producing cells (v). The increased donor and acceptor usage rates resulted in an increase in CS ratios (vi). **e**, All mutants showed similar levels of *env/vpu* and *vif* mRNAs (i), while gp41 and Vif protein translation rates per mRNA (ii; calculated by dividing western blot densitometry results with mRNA levels) were lower in triple mutant-producing cells than in single mutant-producing cells. **f**, The lengths of the 3′ poly(A) tail in protein-specific mRNAs are shown. WT and triple mutants showed no significant difference (two-tailed Kolmogorov–Smirnov test: box, first to last quartiles; whiskers, 1.5× interquartile range; centre line, median; points, individual data values; violin, distribution of density; sORF, short open reading frames). The lengths of poly (A) tail, varied among CS, PS, US and virion RNAs (Extended Data Fig. 8).

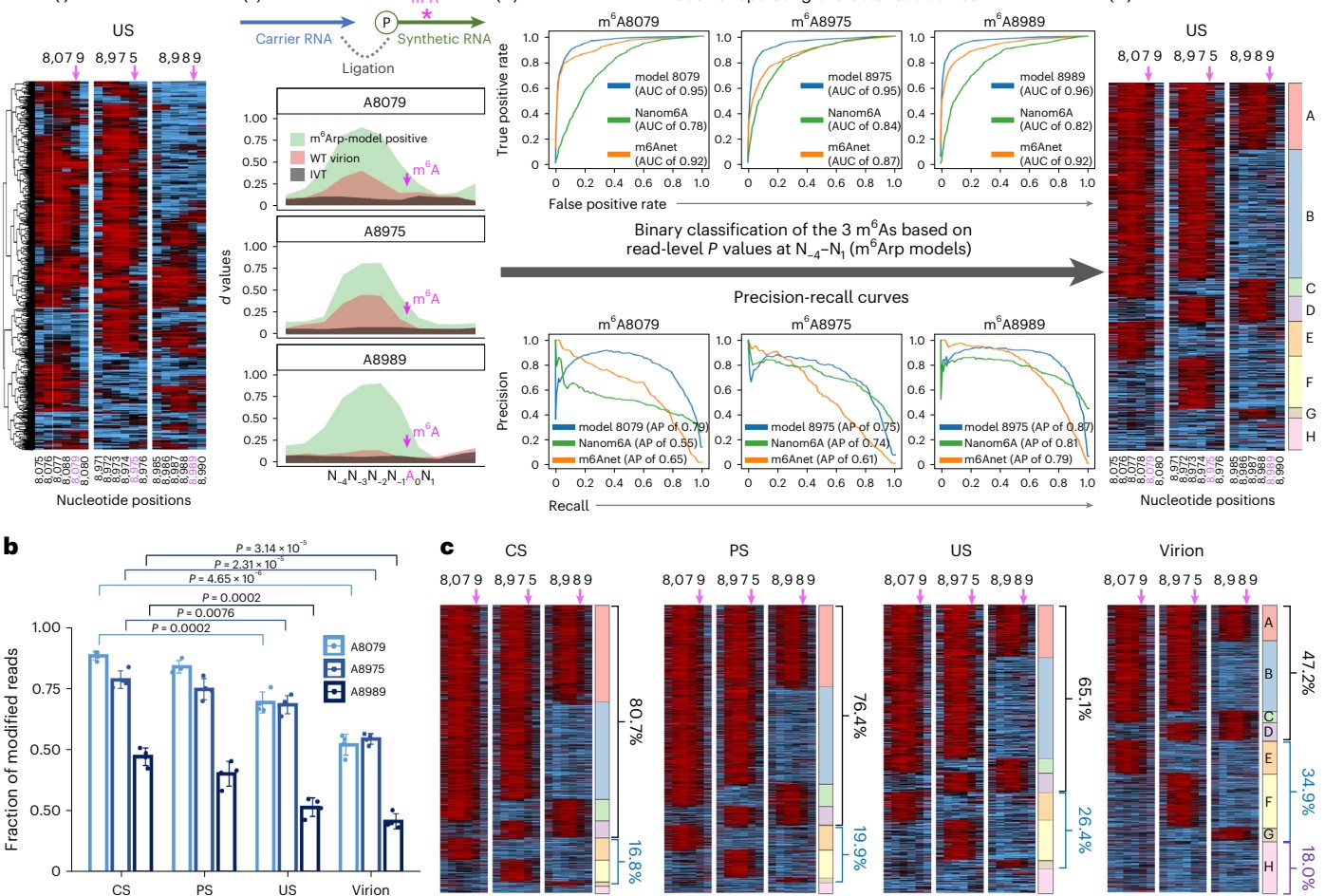

**Fig. 5 | Read-level binary classification identifies HIV-1 RNA subspecies with distinct m⁶As. a**, The development of read-level binary-classification models (m⁶Arp models) for the three predominant m⁶A sites. The heat map view shows heterogeneous RNA reads (rows) clustered on the basis of Tombo-MSC per-read $P$ values at positions −4 to +1 relative to the A8079, A8975 and A8989 sites ($N_{-4}$, $N_{-3}$, $N_{-2}$, $N_{-1}$, $A_0$, $N_1$; $A_0$, m⁶A marked by purple arrows (i)). To train the models, we generated three sets of positive and negative training datasets (Extended Data Fig. 9 and Supplementary Table 8) (ii). Positive control (green) and WT virion RNA reads (brown) showed a common shift of $d$ values to positions −4 to +1 relative to the m⁶A sites. Our pretrained models showed superior AUROC (AUC; top) and

area under the precision-recall curve (AP; bottom) than m6ANet and nanom6A (iii). The m⁶Arp models effectively determined m⁶As for each read and identified RNA subspecies with distinct ensembles of these m⁶As (subspecies A–H) (iv). **b**, A read-level estimation of m⁶A stoichiometry of at the three sites for CS, PS, US and virion. Data are presented as mean values ± s.d. Two-tailed $t$-test; $n = 4$ intracellular RNA, $n = 4$ virion RNA and biologically independent samples. **c**, Differential distribution of m⁶A RNA subspecies in CS, PS, US and virion RNA. A total of 97.5% of CS RNAs and 96.3% of PS RNAs have ≥1 m⁶As (subspecies A–G). The fraction of subspecies with ≥2 m⁶As (A–D) was highest in CS (80.7%) and lowest in virion (47.2%).

fraction of US RNA was significantly lower in the triple mutants than in the WT (Fig. 4d(ii)). Since cells rarely tolerate US or incompletely spliced transcripts, HIV-1 must heavily suppress its RNA splicing to maintain sufficient levels of US RNA[45]. However, triple mutants showed a significantly increased usage of D1 donor (Fig. 4d(iii), which occurs in all spliced RNA), A7 acceptor (Fig. 4d(iv), which occurs for all CS) and all other donors and acceptors (Fig. 4d(v)). Consequently, the 'over splicing' by the triple mutants significantly reduced US RNAs, while relatively increasing the CS portion (Fig. 4d(vi)). Single mutant viruses also showed an increase in CS RNA but maintained a higher level of US RNA than did the triple mutants (Fig. 4d(vi)).

### Triple m⁶A mutations reduced HIV-1 protein translation

In addition to its role in RNA splicing, m⁶A has been associated with RNA translation, metabolism and 3′ polyadenylation[43,44,47]. Both HIV-1 Vif and envelope proteins are mainly translated from PS RNA. Although the PS RNA levels (Fig. 4d(vi)) and mRNAs for Vif and envelope (Fig. 4e) were maintained at relatively similar levels among cells producing any mutants, intracellular Vif and envelope gp41 proteins were significantly

reduced by triple mutations, but not by any of the single mutations (Fig. 3c), indicating that inefficient translation of Vif and envelope mRNAs by triple mutants compared with those by single mutants or WT. The length of 3′ poly(A) tails is known to have important implications for RNA translation and metabolism[23]. Our analysis confirmed that there were no notable differences in poly(A) tail lengths between the WT and mutant RNA isoforms in this regard (Fig. 4f and Extended Data Fig. 8). These results suggest the regulatory roles of these m⁶As in viral RNA translation.

### Individual RNA-level analysis of site-specific m⁶As

To investigate the functions of the three m⁶As at the single RNA molecule level, it is crucial to determine the presence of m⁶As accurately and without bias for each read and at different sites. We have developed new read-level binary classification methods that are specific to each of the three m⁶As (m⁶Arp models; for details, see Methods). These methods are based on the read-level $P$ value patterns surrounding these sites (Fig. 5a(ii) and Extended Data Fig. 9), consistent with those of cellular transcripts[48]. The generation of per-read $P$ values is highly

reproducible under our optimized conditions ($r^2 > 0.999$ with >25,000 IVT reads), and the *P* values remained consistent in repeated experiments (Extended Data Fig. 2c,d).

Our pretrained m⁶Arp models showed an area under the receiver operating characteristics curve (AUROC) ranging from 0.95 to 0.97, with false-positive rates (FPRs) of 8.40–10.80% and false-negative rates (FNRs) of 8.90–12.40% for the three m⁶As (Supplementary Table 7). Our models out-performed Nanom6A[40] and m6Anet[49], which are *k*-mer-based methods optimized for whole-transcriptome analysis (Fig. 5a(iii)). Moreover, the performance of our models, which evaluate m⁶A presence one read at a time, is unaffected by the number of reads or the data composition of test samples (that is, data sparsity and imbalance problems)[35,36,49]. These features of our models enabled us to accurately determine RNA subspecies with distinct ensembles of the three m⁶As (subspecies A–H) (Fig. 5a(iv)) and to compare these RNA subspecies in various settings.

### Higher m⁶A stoichiometry on HIV-1 mRNAs than genomic RNA

Our models also demonstrated a strong linearity of quantification ($r^2 > 0.9982$) (Extended Data Fig. 9g). Consistent with a recent report demonstrating reduced m⁶A levels in the genomic RNAs[7], we found the stoichiometry of the three m⁶As were significantly higher on translating mRNAs (CS and PS RNA) than that on genomic (virion) RNA (Fig. 5b). The estimates from other tools, including Nanom6A, m6Anet and Tombo, were consistent with our findings (Extended Data Fig. 9h). These m⁶As were most frequently detected on CS, showing average 88.5% (±1.8 s.d.), 78.7% (±3.6 s.d.) and 47.2% (±3.6 s.d.) of m⁶A modifications at A8079, A8975 and A8989, respectively.

Interestingly, read-level analyses of RNA subspecies revealed that virtually all CS and PS reads had at least one of these m⁶As (subspecies A–G, collectively accounting for 97.5% and 96.3% of all CS and PS reads, respectively), while the fraction dropped to 82.1% in virion RNA (Fig. 5c). Moreover, RNA subspecies with multiple m⁶As (subspecies A–D) accounted for a predominant portion in the CS and PS RNAs (80.7% and 76.4%, respectively), whereas the portion was substantially lower in the virion RNAs (47.2%). US RNAs, consisting of both translating mRNA and genomic RNA types (Fig. 4b), showed a mixed character of mRNAs and virion RNAs (genomic RNA) as expected. The group H (lacking the three m⁶As) was mostly not spliced and highly enriched in genomic RNAs. These results, therefore, further support the important roles of these m⁶As in splicing and translation. Having a relatively lower number of m⁶As on the genomic RNA may be favoured during the virion packaging and during viral RT where m⁶As are reported to be inhibitory[7,8].

### Redundant roles of the m⁶As in regulating RNA isoforms

We analysed splicing patterns of these RNA subspecies to evaluate the roles of each of these three m⁶As and their ensembles. Consistent with splicing patterns on a total population scale, all WT subspecies showed substantially lower donor (for example, D1, D4 and A5) and acceptor (for example, A7) usages than the triple mutants (Fig. 6a), pointing to the suppressive roles of these m⁶As. Among the subspecies A–G, however, there were only moderate differences in splicing patterns and donor or acceptor usages. Having at least one of the three m⁶As, regardless of the position or the number of m⁶As installed, was sufficient for these RNAs to control splicing events and produce all major splicing isoforms (Fig. 6a).

Given the potential functional redundancy of these m⁶As, we then asked why HIV-1 maintains excessive m⁶As on its RNAs. To explore additive or synergistic effects of these m⁶As, we hypothesized that having multiple m⁶As on its RNA molecules ('subspecies A–D' in Fig. 5c) is vital for HIV-1 to maintain normal levels of viral replication. Given that all single mutants exhibited no significant reduction in most of their replication stages (Fig. 3), we investigated whether the single mutants (1) selectively enrich multiple-m⁶A-containing RNAs in their RNA pool

and/or (2) deposit new m⁶As at other DRACH sites in response to the loss of a major m⁶A. We found indistinguishable or only moderate differences in the m⁶A stoichiometry (Fig. 6b,c) and m⁶A landscape (Extended Data Fig. 10a) between the single mutants and the WT HIV-1. These results suggest no significant additive effects of the three m⁶As on HIV-1 replication, except for a moderate increase in alternative splicing in the single mutants.

RNA subspecies of single mutants also exhibited similar splicing donor or acceptor usages (Extended Data Fig. 10b–d) compared with those of WT (Fig. 6a(v)), suggesting no apparent functional changes of these m⁶As in single mutants.

In the context of RNA population-level evolutionary responses, our data suggest that the functional redundancy of m⁶As on the HIV-1 RNA best aligns with the bet-hedging mode among the three core modes of evolutionary response, including adaptive tracking, plasticity in phenotype (or function) and bet hedging[50]. HIV-1 may tolerate these multiple redundant m⁶As to minimize the risk of losing them, for example, by unpredictable random mutagenesis ($1 \times 10^{-5}$ to $1 \times 10^{-3}$ mutations per bp per cycle for HIV-1 (ref. 51)). Interestingly, all the single mutants maintained at least one of the three m⁶As in most of their CS (87.7–94.7%) and PS (81.9–93.9%) RNAs, levels comparable with the WT HIV-1 (Fig. 6d). Despite causing substantial loss of m⁶A at the population level, single mutations had only a marginal effect on overall viral fitness. The loss of all three, however, eroded the potential of RNA communities to sustain their control over splicing and translation, and adversely affected the various stages of the HIV-1 life cycle (Fig. 3). The possibility that single mutants may exhibit phenotypes under more stringent assay conditions, nevertheless, cannot be excluded.

## Discussion

In this study, we made substantial strides in understanding the HIV-1 epitranscriptome through technological innovations enabling a full-length and individual RNA-level analysis of long and complex RNAs. Our analysis revealed that HIV-1 maintains functionally redundant m⁶As almost exclusively at the three DRACH sites (A8079, A8975 and A8989 in HIV-1$_{NL4-3}$; equivalent to A8089, A8985 and A8999 of HIV-1$_{HBX2}$ strain, respectively) near the 3' end, out of a total 242 DRACH sites on its RNA. Nearly all (>96%) spliced mRNAs of HIV-1 have at least one of these m⁶As, with each labelling up to 89%. They do not exhibit any notable changes in m⁶A site specificity even after losing the major m⁶A due to mutation(s). The remarkable site specificity of m⁶As and their exceptionally high stoichiometry to the HIV-1 genome, relative to those of cellular mRNAs[43], suggest HIV-1-specific and context-dependent roles of these m⁶As.

m⁶A deposition on cellular RNAs is largely regulated through the 'targeted suppression' of RNA-binding protein (RBP) complexes (for example, the exon junction complexes)[52–54]. The m⁶A sites of HIV-1 mirror the typical m⁶A patterns on cellular mRNAs[43], located downstream of the last exon junction (A7) and adjacent to the stop codons (*tat* and *nef* stop codons). Unlike cellular RNAs, however, HIV-1 shows no differences in m⁶A site selection between spliced and US mRNAs. Furthermore, the US RNAs of HIV-1 exhibit markedly lower m⁶A stoichiometry than spliced mRNAs, in contrast to the trends observed in cellular US RNAs[52]. Given that m⁶A deposition can be influenced by transcriptional context and speed[55–57], as well as RNA–RBP interactions[52–54], the distinct m⁶A site specificity and stoichiometry of HIV-1 RNA may reflect virus's unique RBP–RNA interactions and/or transcriptional contexts of HIV-1 RNAs, distinct from those of cellular mRNAs.

The differential m⁶A stoichiometry between HIV-1's spliced mRNAs and genomic RNAs may also reflect their unique RBP–RNA interactions[58,59] and/or transcriptional contexts of HIV-1 RNAs in different fate paths (mRNA or genomic RNA)[60,61]. A recent study also suggested a selective demethylation of m⁶As on genomic RNA by a Gag–FTO complex[7], which may occur independently of differential m⁶A deposition. The fine-tuning of m⁶A levels between HIV-1 mRNA and genomic RNAs

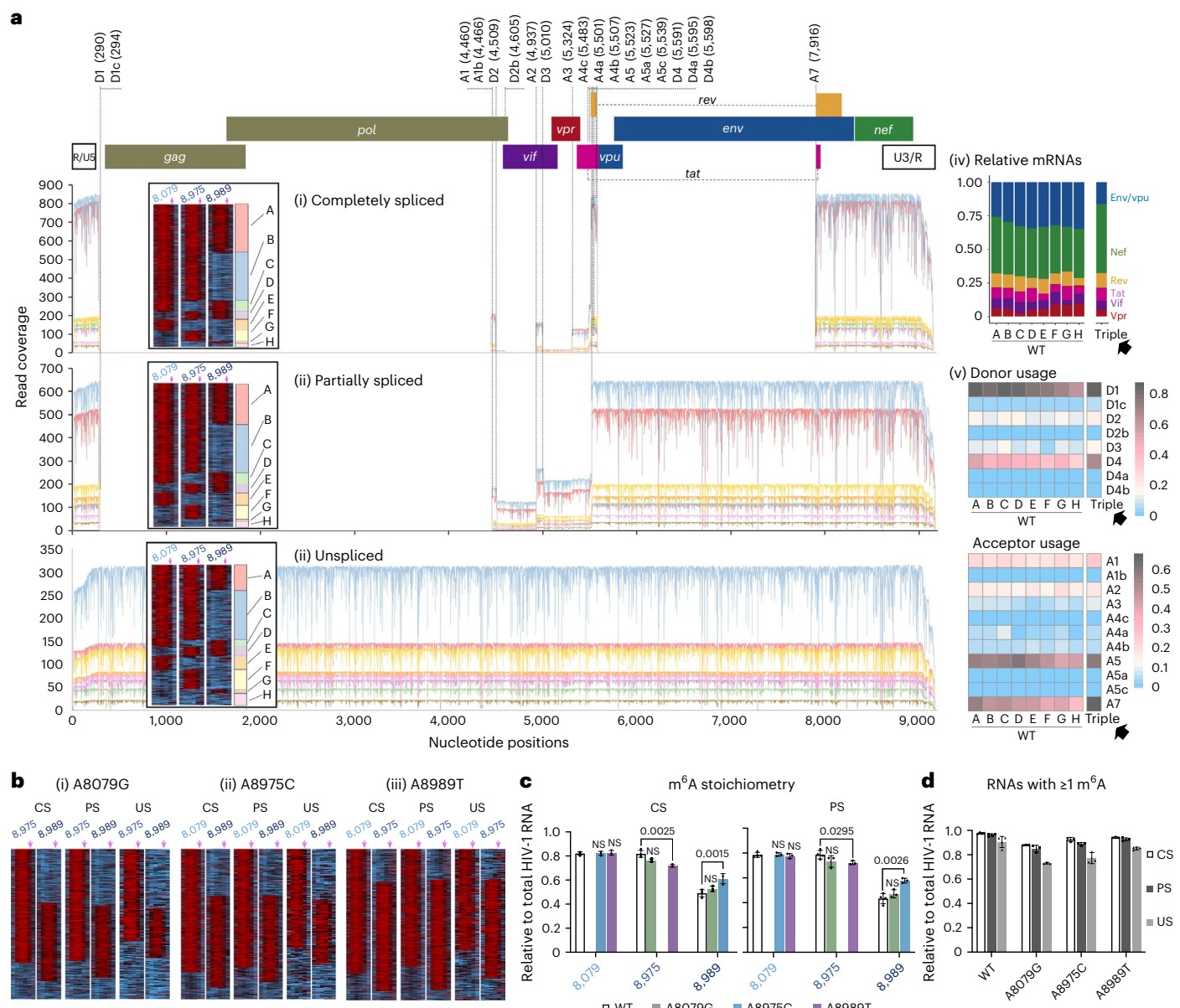

**Fig. 6 | Intramolecular HIV-1 RNA m⁶A heterogeneity and functional redundancy. a**, Splicing donor and acceptor usages were analysed for WT subspecies A–H of CS (i), PS (ii) and US (iii) RNA. All WT subspecies showed substantial differences in mRNA contents (iv) and splicing donor and acceptor usages (v) compared with those of the triple mutant (black arrows on iv–v representing a baseline control of RNAs lacking all three m⁶As). Among WT subspecies, however, all showed only marginal differences in generating splicing isoforms (iv–v). **b**, Heat map views for A8079G (i), A8975C (ii) and A8989T (iii). **c**, Comparison of m⁶A stoichiometry at the three m⁶A positions. All single

mutants showed indistinguishable or only moderate differences in m⁶A stoichiometry compared with the WT. One exception was that the A8975C mutant showed a moderate (1.3-fold) increase in the stoichiometry of the neighbouring 8,989 m⁶A. **d**, The fractions (%) of RNAs with ≥1 m⁶As in the CS, PS and US groups were compared. Knocking out one of the three m⁶As still allows HIV-1 to maintain ≥1 m⁶A in 87.7–94.7% and 81.9–93.9% of CS and PS RNAs. Data are presented as mean values ± s.d. Two-tailed *t*-test; WT, *n* = 4; A8079G, A8975C and A8989T, *n* = 3; biologically independent samples.

may help maximize viral translation while minimizing the inhibitory effects of m⁶As during virion packaging[7] and RT[8]. Further investigation is required to better understand the exact mechanisms.

The three site-specific m⁶As also exhibit functional features partially distinct from m⁶As in cellular RNAs. Recent studies have revealed that cytoplasmic m⁶A readers, YTHDF1–3, share their binding sites on cellular RNAs and facilitate RNA degradation[62,63]. HIV-1 also exhibited shared binding sites among YTHDF1–3 (refs. 6,8), but unlike these reports, we found that the total HIV-1 RNA copies remained similar in both WT- and triple mutant-producing cells. Instead, triple mutation affected the translation efficiency of viral mRNAs, resulting in substantially lower viral protein levels. Although the precise mechanisms

remain unclear[62,63], m⁶As in untranslated regions (UTRs) have been reported to stimulate translation for cellular mRNAs[64–68]. The impact of m⁶As on HIV-1 RNA expression has yielded inconsistent results among previous cellular perturbation and quantitative PCR-based studies[6,8,10].

The connection between m⁶As and RNA splicing is also intricate and probably context dependent. The impact of m⁶As on individual genes seems to be heterogeneous in whole-transcriptome studies[69–71]. The timing of m⁶A deposition (occurring before or after splicing) is key to understanding the connection, but it appears to be complex[52–55,70–73]. Notably, gene-specific or virus-specific investigations have established clearer links between m⁶As and alternative splicing, involving m⁶A writers[74–77], the nuclear reader YTHDC1[9,10,78] and erasers[7,79,80], as

well as interactions with splicing regulatory elements[16–19,69,81]. However, the impact of YTHDC1 knockdown on HIV-1 alternative splicing appeared controversial in two recent studies[9,10]. G-quadruplexes (G4s), co-localized with the three major m[6]A sites, might also influence alternative splicing[82,83]. The m[6]A-G4 co-localization has been reported for other viruses[84]. Additionally, m[6]A-mediated translational enhancement of viral regulatory protein, particularly Rev, could affect the production of HIV-1 US RNA[45].

Overall, we demonstrate that a full-length, single-molecule-level analysis using DRS can provide new opportunities to untangle the complexity of the RNA biology. The new methods and analytical standards presented herein can serve as a useful reference for future investigations of RNAs of interest and various RNA viruses in the ever-expanding RNA virosphere.

## Methods

### Extraction of HIV-1 virion RNA

HEK293T cells (CRL-3216, American Type Culture Collection) were transfected with the HIV-1 pro-viral DNA construct pNL4-3 by polyethylenimine as described[85]. The cell culture medium was exchanged with fresh medium at 6 h post-transfection and the supernatant was collected at 72 h for virion RNA extraction and p24 particle release measurement by the HIV-1 p24 enzyme-linked immunosorbent assay (ELISA) kit (Abcam). Total RNA from the cells was extracted by TRI reagent (Sigma-Aldrich) following the manufacturer's instructions. Total viral particles were concentrated and centrifuged for 1 h 40 min at 28,000$g$ at 4 °C with a 10% sucrose gradient. After discarding the supernatant, the pellet was resolved using 160 µl of 1× Hanks' balanced salt solution, followed by DNase treatment for 30 min at 37 °C. The sample was incubated in 1 ml TRIzol for 5 min, followed by the addition of 200 µl chloroform and shaking for 30 s. The tube was then centrifuged for 10 min at 12,000$g$ at 4 °C. The clear upper aqueous layer, which contains RNA, was transferred to a new 1.5 ml tube and 0.7 ml of isopropanol was added. After 10 min incubation at room temperature, the tube was centrifuged for 10 min at 12,000$g$ at 4 °C. The supernatant was discarded, and then the pellets were resuspended in 1 ml of 70% ethanol, followed by another centrifugation for 5 min at 7,500$g$. The RNA pellet was air-dried in the hood and then resuspended in 30 µl of diethyl pyrocarbonate-treated water.

### Nanopore DRS of full-length HIV-1 RNA

For HIV-1 RNA DRS, 1 µg of virion RNA or 10 µg of total cellular RNA in 9 µl was used for DRS library preparation following the manufacturer's protocol (the Oxford Nanopore DRS, SQK-RNA002) with a modification of the RT step. Mixtures of 111 HIV-1 sequence-specific DNA oligomers (Integrated DNA Technologies) at a copy number ratio of 1:30 (HIV-1 RNA:each oligomer) were annealed to the RNA at 65 °C for 5 min and proceeded to the RTA ligation step. The DNA oligomers used are listed in Supplementary Table 9. The library was run on FLO-MIN106D flow cell for 48 h on a MinION device (Oxford Nanopore Technologies). IVT RNA fragments were sequenced the same way except 4 µg RNA was used for the input.

### DRS data preprocessing

MinKNOW GUI (v.3 or later; Nanopore Technology) was used for sequencing data collection. Multi-fast5 reads were base-called by guppy (v.3.2.8 or higher) using the fast base calling option. The base-called multi-fast5 reads were then converted to single-read fast5s using the Oxford Nanopore Technologies application programming interface, ont_fast5 (v.3.3.0). Fastqs were aligned to the HIV-1 genome reference sequence AF324493.2 from the National Center for Biotechnology Information (NCBI) or the human reference sequence (human genome assembly GRCh38.p13 for Extended Data Fig. 7a) with the options '-ax map-ont' using minimap2 (v.2.24). Unmapped reads were discarded using SAMtools (v.1.6). For in-depth analysis of the 3'-end region, short

reads (read length <2,000) were filtered out using NanoFilt (v.2.7.1). The sequence read length was extracted by aligning sequences against the HIV-1 coding sequences retrieved from HIV-1 genome reference sequence AF324493.2 from NCBI using minimap2 (v.2.24), retaining multiple secondary alignments (parameters -p 0 -N 10) and counting the number of unique read IDs among mapped alignments. Reads were required to be over 8,000 nucleotides to be classified as full-length HIV-1 RNA. Sequencing read coverage depth was calculated using bedtools genomcov of v.2.25.0 and visualized for 1 nt binning size in the plots.

### DRS-mediated detection of RNA modifications

Nanopore DRS can detect various types of chemical modification based on the DRS electrical signals (raw current intensity and dwell time) and/or the detection of modification-induced base-calling errors. For a whole-genome scale analysis of chemical modifications on HIV-1 RNA, we used Tombo (v1.5.1, based on current intensity differences[32]; section 1.2), Eligos2 (v.2.0.0, based on error rates[33]; section 1.3), Nanocompore (v.1.0.4, based on both current intensity and dwell time differences[34]; section 1.4) and Xpore (v.2.1, based on current intensity differences at the individual read level[35]; section 1.4). We also used Nanom6A[40] (v.2.0) and m6Anet[49] (v.1.1.1) for a read-level detection of m[6]As (section 1.5). We then cross-compared the results of Tombo, Eligos2 and Nanocompore to identify the most likely candidates of RNA modification sites (section 1.6). Default options were used for all software used in this study, except where noted.

**Preparation of IVT RNA controls.** We generated three types of HIV-1 IVT datasets: (1) full-length (with the identical nucleotide sequence to NL4.3 RNA from the nucleotide position 1 to 1,973 with a poly(A) tail at the 3' end), (2) half-length (F1 fragment from 1 to 4,587 and F2 fragment from 4,588 to 9,173) and (3) short IVT (7 fragments of 1 to 2 kb covering the whole genome) (Supplementary Fig. 1). The full-length IVT RNA reads were used for the whole-genome scale comparison of RNA modification sites (Fig. 1). The half-length IVT datasets were used to train m[6]Arp models (see the 'Machine learning: determining m6A modifications per-read per-position basis' section). We found the short IVT RNA sets are not suitable for the whole-genome scale analysis due to unreliable modification signals at the ends of RNA reads (see the 'Determining the signal instability at the first and the last 40 nucleotides of DRS reads' section below).

**Tombo analysis.** Tombo software uses raw DRS current intensity data to identify modified bases. DRS raw signals can distinguish canonical and non-canonical bases within the read head of the pore protein, but they also reflect various contexts surrounding the read head, including local RNA structures and neighbouring nucleotide sequences interacting with pore or motor proteins[32,86]. Given the high levels of noise in Tombo de novo analysis, we employed a two-step noise-reduction procedure to refine modification signals as follows:

**Step 1 Tombo-MSC.** Tombo de novo analysis, calculating current intensity deviations at each site using the 'expected canonical signal levels', generated highly noisy modification signals (Supplementary Fig. 2). To reduce the noise, we employed Tombo model–sample–compare (MSC) for a more accurate modification detection using HIV-1 IVT RNA reads, which adjust the 'expected canonical signal levels' for HIV-1 RNA-specific analysis (using the '−sample-only-estimates' option). As expected, Tombo's Tombo-MSC generated HIV-1-specific per-read $P$ values for modification sites and population-level 'estimated fractions of significantly modified reads' (dampened_fraction or $d$ values; Extended Data Fig. 2a), which successfully dampened the majority of noise signals observed in Tombo de novo analysis (Supplementary Fig. 2). For a full-length comparison of modification sites, we used a total of 5,411 of full-length IVT RNA reads as a canonical control for Tombo-MSC

(Fig. 1 and Extended Data Fig. 2a). For an analysis using the half-length IVT data, we used 26,000 reads of the first half (F1) and 28,100 reads of the second half IVTs (F2), which resulted in a similar read coverage (~25,000) for the whole genome (Extended Data Fig. 2b). Tombo-MSC using 25,000 reads of IVT RNAs generated per-read $P$ values that are highly reproducible, showing $r^2 > 0.999$ when compared with the $P$ values generated with 50,000 IVT reads (Extended Data Fig. 2c) and $r^2 = 0.8852 \pm 0.0244$ for four different WT HIV-1 virion RNA datasets prepared independently (Extended Data Fig. 2d).

**Step 2 removal of the baseline noise.** Although significantly reduced compared with Tombo 'de novo' analysis, Tombo-MSC analysis showed a considerable level of the baseline $d$ value noise in our control analysis using 1,450 reads of full-length IVT reads (IVT subreads that were not used as a canonical control; Supplementary Table 8). The $d$ values of IVT subreads coincided with most of $d$ values of native HIV-1 RNA (Supplementary Fig. 3). A subtraction of the baseline noise substantially refined the modification signals of virion RNA (Supplementary Fig. 3).

A similar two-step noise-reduction procedure using IVT subreads was applied to all pretrained detection tools used in this study, including ELIGOS2 (Supplementary Fig. 5), nanom6A and m6Anet (Supplementary Fig. 6). DRS analysis 1.3. Eligos2 analysis: Eligos2 identifies the position of modification based on the differences in 'error at specific base'[33] between the native HIV-1 RNA and IVT RNA. Similar to Tombo-MSC analysis, we generated odd ratios of error at specific base at each position for both native RNA and IVT subread data using 5,411 reads of full-length IVTs as a canonical control (Supplementary Fig. 5). The modification signals (odd ratios) of native RNA were refined by subtracting the baseline noise (1,450 IVT subreads).

Dwell time was extracted for per-read and per-position levels using the Tombo Python application programming interface. For Tombo-MSC '$d$ values' ('estimated fraction of significantly modified reads'), see Extended Data Fig. 2a.

**Determining the signal instability at the first and the last 40 nucleotides of DRS reads.** Nanopore DRS fails to read the 5′ end (the first 10–12 nucleotides) of RNAs due to the instability of the ends of RNA during the DRS runs[27,28,87]. In our data analysis, we also found that the electric signals of both ends of DRS reads (although successfully read by DRS) can still be unreliable (Supplementary Fig. 4a). To clearly address this, we evaluated the stability of local DRS signals by comparing long (F1 and F2) and short IVT RNA (F3–F9) reads with identical nucleotide sequences (Supplementary Fig. 4b). A systemic evaluation of DRS signals per position from each end (5′ or 3′) of the reads showed that DRS signals of the first 10–40 nucleotides and the last 10–40 nucleotides are not reliable, showing significant difference compared with the identical sequences in the middle of long RNA reads ($P < 0.01$; Mann–Whitney $U$ tests comparing every pair of 10 base bins between long and short IVT reads; Supplementary Fig. 4b(ii)). For an accurate detection of RNA modifications, we excluded the DRS data of the first and the last 40 nucleotides.

**Nanocompore and xPore analyses.** To detect modified nucleotides, we also used Nanocompore (evaluating current intensity and dwell-time differences)[34] and xPore (evaluating current intensity differences within a read)[35] comparing raw DRS signals between native HIV-1 RNA and IVT RNA (Fig. 1 and Extended Data Fig. 3). Tools that compare raw DRS signals of two comparing samples, including Nanocompore, xPore and Tombo-Level Sample Compare (Tombo-LSC), do not require the noise removal using IVT subreads.

**Nanom6A and m6Anet analyses.** For a read-level detection of m⁶As, we employed Nanom6A[40] and m6Anet[49], which are pretrained tools. Similar to the Tombo-MSC and Eligos2 analyses described above, we generated m⁶A modification ratios for both native RNA and IVT subread

data and performed a baseline noise removal to refine the m⁶A data (Supplementary Fig. 6).

**Determining common modification sites among Tombo, Eligos2 and Nanocompore results.** The sensitivities of DRS signal features, including current intensities, dwell time and base-calling error rates, can vary depending on the types and positions of the modifications[36]. Among all the tested in our study, Tombo, Eligos2 and Nanocompore generated the most reproducible results (Extended Data Fig. 3). Given approximately 80–200 modifications of various kinds may exist per genome based on previous mass spectrometry studies[5,12], we selected and compared the 149 strongest peaks of modification signals from Tombo-MSC ($d$ value >0.05), 167 from Eligos2 (odd ratios >2.4) and 156 from Nanocompore results (logit log-odd-ratio (LOR) score >0.73) (Supplementary Fig. 7a). To determine most probable modification sites among the myriad of modification signals in these analyses, we cross-compared these datasets and identified common peaks of these analyses (Supplementary Fig. 7b–c).

**The probability that 14 out of 25 common sites coincide with DRACH sites.** We found that 14 out of 25 common modification sites (from Tombo and Eligos 2 analysis) on or one base away from the centre of the DRACH site (m⁶A sequence motifs; Supplementary Fig. 7c). To test whether this frequency is simply a random event (null hypothesis), we generated 25 random sites on the HIV-1 genome and calculated their chances to locate on or one base away from the centre of the DRACH sites on the HIV-1 genome. To be considered 'on a DRACH' (or 'success'), a random event must occur within six bases (either upstream or downstream) of the centre of a DRACH site. The sixth base was chosen given the five-base resolution of Nanopore signals and a one-base margin as defined in Supplementary Fig. 7b(iii). The probability of a random site to be 'on a DRACH' (success) is approximately 0.313 = 2,867/9,173: (2,867; number of nucleotides within 6 bases of the centre of DRACH sites on the HIV-1 genome)/(9,173; number of nucleotides of the HIV-1 genome). Given that the probability distribution of 25 random events (from 0 success to 25 successes) is a binomial distribution, we calculated that the probability to have 14 or higher successes is approximately 0.00893 (see the probability distribution in Fig. 1d). This is sufficiently low to reject the null hypothesis; the chance that 14 out of 25 common sites coincide with DRACH sites is highly unlikely to be random.

The probabilities that m⁵C, ac⁴C and m⁶A-reader-binding sites to coincide with the 25 common sites were also calculated as described above using the m⁵C- and ac⁴C-detected areas and m⁶A-reader binding areas (Supplementary Table 4).

## In vitro demethylation of m⁶A on HIV-1 RNA
Recombinant ALKBH5 (active motif) was used for in vitro treatment of HIV-1 virion RNA as described[88]. The reaction mixture contained KCl (100 mM), $MgCl_2$ (2 mM), RnaseOUT (Invitrogen), L-ascorbic acid (2 mM), α-ketoglutarate (300 μM), $(NH_4)_2Fe(SO_4)_2 \cdot 6H_2O$ (150 μM) and 50 mM of HEPES buffer (pH 6.5). The mixture was incubated for 1.5 h at room temperature and then stopped by the addition of 5 mM EDTA.

## m⁶A dot immunoblotting
The extracted HIV-1 virion RNA was directly used for dot-blot assays, as previously described[85]. Briefly, 50 ng of virion RNA, diluted in 1 mM EDTA (total 100 μl), were mixed with 60 μl of 2× saline sodium citrate (SSC) buffer (Invitrogen) and 40 μl of 37% formaldehyde (Invitrogen). The mixture was incubated at 65 °C for 30 min. The nitrocellulose membrane (Bio-Rad) and nylon membrane (Roche) were both soaked with 10× SSC for 5 min before loading the RNA samples. Samples were loaded equally on nitrocellulose and nylon membrane followed by washing with 10× SSC buffer. The nylon membrane was washed with 1× TBST (25 mM Tris, 0.15 M NaCl and 0.05% Tween 20) and stained with

methylene blue while shaking for 2–5 s and washed with ddH$_2$O. The nitrocellulose membrane was UV cross-linked and then blocked with 5% milk in 1× TBST 1 h. m⁶A levels were detected by using an m⁶A-specific antibody (Abcam, cat. no. ab208577,1:1,000). Images were analysed by ImageJ software (v.1.53), and the relative RNA m⁶A levels were normalized to methylene blue staining.

## LC–MS/MS sample preparation

The oligo mixture, including a biotin-labelled target-specific DNA oligomer (1:100) and oligos covering other sites (1:30), was annealed to HIV-1 virion RNA at 65 °C for 5 min and then cooled down to room temperature, as described previously[42]. Samples were digested by nuclease S1 (Invitrogen) for 2 h at room temperature followed by phenol:chloroform purification as previously described[42]. The biotin-labelled target DNA:RNA duplex was recovered by using Dynabeads MyOne Streptavidin C1 (Thermo Fisher Scientific) following the manufacturer's instructions. For RNase T1 digestion, about 200 ng of denatured (95 °C for 2 min and snap cooling at 4 °C) HIV-1 RNA (obtained by modified RNase protection assay) was digested with 50 units of RNase T1 (Worthington) at 37 °C for 2 h and dried in a SpeedVac system (Thermo Fisher Scientific).

## LC–MS/MS

The LC–MS/MS analysis was performed using a BEH C18 column (1.7 µm, 0.3 mm × 150 mm, Waters) with Ultimate 3000 ultra high performance liquid chromatography (Thermo Scientific) coupled to the Synapt G2-S (Waters) mass spectrometer as described previously[42]. The gradient chromatography was performed at 5 µl min⁻¹ flow rate using mobile phase A (8 mM TEA and 200 mM HFIP, pH 7.8 in water) and mobile phase B (8 mM TEA and 200 mM HFIP in 50% methanol) at 60 °C. The gradient consists of an initial hold at 3% B for sample loading, followed by ramping to 55% B in 70 min, 99% in 2 min with 5 min hold before re-equilibration (30 min) at 3% B for initial conditions. The resolved digestion products in the chromatographic eluent were detected in negative ion mode through electrospray ionization on a Synapt G2-S (quadrupole time-of-flight) mass spectrometer operating in sensitivity mode (V-mode). The electrospray ionization conditions included 2.2 kV at capillary, 30 V at sample cone while maintaining source and desolvation temperatures at 120 °C and 400 °C, and gas flow rates at 3 l h⁻¹ and 600 l h⁻¹, respectively. A scan range of 545–2,000 $m/z$ (0.5 s) and 250–2,000 $m/z$ (1 s) was used for first (MS) and second (MS/MS) stage data acquisition. The top three most abundant ions in the first stage were selected for fragmentation for MS/MS using $m/z$ dependent collision energy profile (20–23 V at $m/z$ 545; 51–57 V at $m/z$ 2,000) before excluding them for 60 s using the dynamic exclusion feature.

## LC–MS/MS data processing

The $m/z$ values of the RNase T1 digestion products (and their fragment ions) of a 40-base-long HIV-1 RNA sequence were predicted using Mongo Oligo mass calculator. Manual identification and assignment of m⁶A modification was made by scoring for ~14 Da mass shift of the theoretically expected oligonucleotides (following cleavage at the 3′ end of guanosine) in the modified RNA compared with the unmodified version. A set of controls was used to assign the m⁶A modification at positions 8,975 and 8,989, respectively.

## Site-directed mutagenesis

gBlocks (Integrated DNA Technologies) with single or combination mutations of m⁶A sites were introduced to the HIV-1 vector pNL4-3 for the mutant plasmids. For each mutant plasmid, 500 ng pNL4-3 and 100 ng gBlocks were digested with NcoI-HF and BamHI-HF and ligated for 30 min at room temperature using T4 quick ligation (NEB). The sequences of the mutant plasmids were confirmed by Sanger sequencing.

The mutations were designed based on the following rationales to minimize changes in protein function or RNA structure due to the mutation. The A8079G mutation is situated in the overlapping region of *rev* and *env* genes, designed to be a silent mutation for Rev but inducing the substitution of glutamine with glycine at position 771 in Env. We selected this mutation because this amino acid substitution was shown to only moderately reduce HIV-1 fitness in vitro[89]. A8975C and A8989T are located in the 3′ UTR (U3) of the HIV-1 RNAs. These mutations were chosen to preserve the RNA structures predicted by a minimal free energy structure prediction tool[90]. Although the structures used to design these mutations remain to be validated, A8079G, A8975C and A8989T mutant viruses replicate normally, exhibiting insignificant or only marginal differences in various features of their RNAs—including m⁶A methylation, alternative splicing, 3′ poly(A) tail, and translation of viral RNAs—suggesting an insubstantial impact of the mutations themselves.

## Digital quantitative PCR for total HIV-1 RNA

Viral RNA production was measured by RT–PCR and DRS analysis. An equal amount of total cellular RNA for each sample was used for RT and complementary DNA generation. The QuantStudio 3D Digital PCR System (Applied Biosystems) was used with appropriate consumables provided by the manufacturer, including the QuantStudio 3D Digital PCR Master Mix v.2 and TaqMan 5′-6 FAM or VIC probe (Applied Biosystems). The primers used are listed in Supplementary Table 9.

## Western blot analysis

Collected cells were lysed in RIPA 1× buffer (Abcam) with a protease inhibitor cocktail (Sigma-Aldrich) and incubated for 30 min on ice. The cell lysates were centrifuged at 12,000g for 15 min at 4 °C and the supernatant was transferred to a fresh tube and mixed with an equal volume of Laemmli 2× buffer (Bio-Rad). The mixture was then incubated at 95 °C for 10 min. The proteins were separated on sodium dodecyl sulfate–polyacrylamide gels and then transferred to a nitrocellulose membrane (Bio-Rad). The membranes were washed in 1× PBS, Tween 20 (PBST) (10 mM sodium phosphate, 0.15 M NaCl and 0.05% Tween 20) and blocked in 5% milk in 1× PBST for 1 h. Primary and secondary antibodies were diluted at 1:1,000 and 1:5,000, respectively, each in 5% milk in 1× PBST. The signals were visualized by chemiluminescence. The primary antibodies used were HIV-1 p24 (NIH AIDS Reagent Program, cat. no. ARP-6458,1:1,000), gp41 (NIH AIDS Reagent Program, cat. no. ARP-11391,1:500), Vif (NIH AIDS Reagent Program, cat. no. ARP-6459,1:500), GAPDH (Abcam, cat. no. ab8245,1:1,000) and anti-mouse (Promega, cat. no. W4021,1:5,000) for the secondary antibody.

## Measuring HIV-1 infectivity using GFP reporter cells

An equal amount of virus stock (pg) was used to infect GHOST R3/X4/R5 cells (ARP-3943, NIH HIV reagent program) in 6-well plates[91]. After 48 h post-infection, the cells were washed with PBS and fixed. The green fluorescent protein (GFP) expressions for all samples were acquired by the Attune NxT flow cytometer (Thermo Fisher Scientific) and analysed by the FlowJo software (BD Biosciences).

## Single cycle infection of CEM-SS cells

A total of $2 × 10^6$ CEM-SS cells (ARP-776, NIH HIV reagent program) were infected with WT or triple mutant virus at 2 multiplicity of infection (MOI) in 1 ml Roswell Park Memorial Institute (RPMI) 1640 with 1% penicillin/streptomycin (P/S) and 10% foetal bovine serum (FBS). Cells were incubated with viruses for 1 h, swirling every 20 min, and then transferred to T25 flask with 10 ml RPMI 1640 (1% P/S and 10% FBS). At 24 h post-infection, the cells were washed with PBS and the culture medium was exchanged with RPMI 1640 medium with drugs (1% P/S, 10% FBS, 100 nM T20 and 100 nM IDV). At 96 h post-infection, the single cycle infected cells were collected and total cellular RNAs were extracted with TRI Reagent (Sigma-Aldrich, T9424), following the manufacturer's instructions.

## Jurkat cell infection

A total of $6 \times 10^6$ Jurkat cells (ARP-177, NIH HIV reagent program) were infected with WT or triple mutant virus at 1 MOI (first experiment) or 2 MOI (second and third experiments) in 2 ml RPMI 1640 (1% P/S and 10% FBS) for 1 h, swirling every 20 min. Cells were then transferred to T75 flask, adding RPMI 1640 (1% P/S and 10% FBS) up to 30 ml. At 24 h post-infection, the cells were washed with PBS and the culture medium was exchanged with fresh RPMI 1640 (1% P/S and 10% FBS). At 96 h post-infection, total cellular RNAs were extracted with TRI Reagent (Sigma-Aldrich, T9424), following the manufacturer's instructions.

## Machine learning: determining m⁶A modifications per-read per-position basis

The goal is to build machine learning models that determine whether an HIV RNA molecule has m⁶A modifications on a per-read and per-position basis. The classification is based on the read-level $P$ value output from Tombo-MSC. The source code is available at ref. 92.

**Read-level $P$ value patterns analysis.** Tombo-MSC uses IVT data to adjust the expected current signal levels and generates per-read, per-position $P$ values for current signal difference of target RNA (native HIV-1 RNA) reads. We found that Tombo-MSC's per-read $P$ values were highly reproducible when a sufficient number of IVT canonical control reads were used (for example, $r^2 > 0.999$ with >20,000 IVT reads; Extended Data Fig. 2c) and when tested in our repeated experiments using four sets of virion RNAs that were prepared independently of each other (Extended Data Fig. 2d). We also found a consistent pattern of per-read $P$ value distribution near the three m⁶A sites (positions 8,079, 8,975 and 8,989) on native HIV-1 RNA, showing a shift of the $d$ values (or median $P$ value) to upstream of the m⁶A sites at positions $N_{-4}$ to $N_1$ ($N_0$, m⁶A site) (Fig. 5a(ii) and Extended Data Fig. 9e). The patterns were consistent between native HIV-1 RNAs and m⁶A-control RNAs. Here we developed new read-level binary classification methods m⁶A based on the read-level $P$ values at positions $N_{-4}$ to $N_1$ (m⁶Arp models) for the three predominant m⁶A sites. Additional positions (including $N_2$ to $N_5$) contributed only negligibly to the model accuracy.

**Preparing control RNAs.** We built three separate models to detect modifications at positions 8,079, 8,975 and 8,989. We generated approximately 1 kb long positive control RNAs using three synthetic RNA oligos, each harbouring m⁶A at 8,978, 8,975 and 8,989, respectively (Supplementary Table 2 and Extended Data Fig. 9a–d). The 8,079, 8,975 and 8,989 models used DRS reads of these RNAs (8978m⁶A+, 8975m⁶A+ and 8989m⁶A+ datasets) as positive-labelled training data. In parallel, IVT RNA reads that cover the same 1 kb region were used as negative-labelled training data. We also tested full-length IVT RNA and F2 IVT RNA reads as negative-labelled training data. These negative-labelled training data showed only negligible difference in the performance of the m⁶Arp models.

Synthetic RNA oligos were custom synthesized by Horizon Discovery (Extended Data Fig. 9a). Positive-control RNAs were generated by ligating carrier IVT RNA to synthetic RNA oligos (Extended Data Fig. 9b); the ligated RNA aligns to approximately 1 kb of the 3' end of HIV-1 RNA. The 3' end of carrier IVT RNA and the 5' end of phosphorylated synthetic RNA were joined by ligating these two RNAs. The ligated RNA was then subjected to poly (A) tailing using *Escherichia coli* poly(A) polymerase (NEB). The carrier IVT RNA were generated as described in 'Preparation of IVT RNA controls' section above using DNA templates generated by PCR using primers shown in Supplementary Table 3.

**Selecting Fisher options.** Models trained with Tombo-MSC $P$ values generated with Fisher 0, 1, 2 and 3 showed varying levels of the AUROC, FPR and FNR (Extended Data Fig. 9f). Of these, the 'Fisher = 0' option showed the best AUROC values for all three models (models 8079,

8975 and 8989) ranging from 0.95 to 0.97, with 8.40–10.80% FPR and 8.90–12.40% FNR (Supplementary Table 7).

**Model selection.** All our machine learning models were linear support-vector classifiers, implemented using the scikit-learn Python package (v.0.23.2) (ref. 93). Models used default scikit-learn settings, except where noted. We selected support-vector classifiers in consideration of their potentially high generalizability and resistance to over-fitting due to their small number of parameters ($n + 1$, where $n$ is the number of features). The magnitude of a coefficient in the model is a measure of the relevance of the corresponding feature to the classification problem. Coefficients of our trained models are reported in Supplementary Table 7. The positive and negative classes were weighted to achieve a balanced comparison. Other than this weighting, models used default parameters as listed on the scikit-learn documentation page. As an alternative to the support-vector classifier described here, an unsupervised learning model based on the DBSCAN algorithm was also assessed. It demonstrated a similar FPR and FNR to the support-vector classifier (data not shown).

**Five-fold cross-validation for the accuracy of the model.** The model accuracy was first assessed using fivefold cross-validation[94]. In brief, to assess the accuracy of a model with dataset $T$, the dataset is partitioned at random into five equal-size subsets, $T_1$, $T_2$, $T_3$, $T_4$ and $T_5$. Five new models are trained, each on its own four-fifths of the data. For example, Model $M_1$ is trained using the combined data of $T_2$, $T_3$, $T_4$ and $T_5$, then its FNR and FPR are computed using the reserved partition $T_1$. The FNRs and FPRs from all five folds are averaged to estimate the model's performance on future unseen data.

**Linearity of quantification analysis.** To evaluate the ability of each model to enumerate m⁶A positive and negative reads at the test site represented in their training data, we tested the linearity of quantification of the three models. For the three testing sites (positions 8,079, 8,975 and 8,989), we generated five sets of mixed data each having positive to negative reads ratios of 0:100, 25:75, 50:50, 75:25 and 100:0 (Extended Data Fig. 9g) and assessed the ability of the three new models to quantitate m⁶A modifications. The quantitation results were directly proportional to the fraction of m⁶A positive control reads in premixed controls with expected FPR and FNR at both ends of mix ratios (Extended Data Fig. 9g). After adjusting for each model's FPR and FNR, the output showed nearly complete matches to the expected values.

## Nucleotide sequence conservation near the major m⁶A sites

The HIV-1 B subtype sequences corresponding to A8079, A8110, A8975 and A8989 of the NL4-3 strain (RNA) were extracted from the HIV sequence database (https://www.hiv.lanl.gov/). The sequence logo plots were generated using the ggseqlogo R package (v.0.1).

## HIV-1 RNA splicing analysis

Reads were aligned using minimap2 (v.2.24) against the HIV genome AF324493.2 from the NCBI in spliced mapping mode using a $k$-mer size of 14. Unmapped reads, alignments with a quality lower than 30, and alignments were discarded using SAMtools (v.1.6). Reads were screened for potential splicing donor (SD) and splicing acceptor (SA) sites by identifying exon start/end positions from BED files. SD and SA sites[46] are shown in Figs. 1c, 4a and 6a. Reads presenting the same combination of splice junction and exotic sequences were grouped together and counted. If the potential splicing sites do not match to the SD/SA sites[46], the reads were annotated as unclassified. Annotation was performed according to the convention established in ref. 46 or for potential spliced isoforms, to the open reading frame encountered in the read. The relative levels of spliced viral isoforms were quantified using absolute counts compared with the WT. Splicing donor and splicing acceptor usage were calculated using absolute counts after

screening of known SD/SA sites from BED files. The reads were classified as spliced when their splice junction include either D1 or D1c splicing donor site. If they have an A7 splicing acceptor site, they were further classified into CS. Then, the rest were classified into PS. The poly(A) tail length of each classified group was estimated using the Nanopolish (v.0.14.0) polya module with default parameters.

### IVT read depth analysis

Six sets of IVT data were generated by randomly choosing 500, 1,000, 5,000, 10,000, 20,000, 30,000 and 40,000 reads from a pool of a total 50,000 half-length IVT reads. Each of these IVT sets was tested as a canonical control for Tombo analysis (Fishers 0). A total of 384 reads of HIV-1 virion RNA were used as a sample dataset. The averaged $P$ values for each position were tested with non-parametric pairwise comparison using the ggstatsplot (v.0.9.1) R package. A $P$ value <0.05 was considered significant.

### Statistical analysis

Statistical analysis was performed using GraphPad Prism9 (v.9.5.0) or R package (v.4.0.2), and detailed statistical tests used are indicated. All averaged data include error bars that denote s.d., with single data points shown. A $P$ value <0.05 was considered significant. We provide triplicated (or quadruplicated) experimental data of biologically independent samples. No statistical methods were used to predetermine sample sizes, but the sample size of $n = 3$ or $n = 4$ routinely provides sufficient statistical power (when present) in our study utilizing accurate and highly reproducible assays and molecular biology experiments. These number of samples are commonly used in molecular biology publications to provide statistical conclusions, as well as to address the rigour and reproducibility. Data collection and analysis were not performed blind to the conditions of the experiments. Experimental groups were determined on the basis of the experimental hypothesis (for example, the impact of mutations or RNA isoforms) and all the experimental data and sequencing data that pass the data exclusion criteria were used without any additional selection. Data collection and analysis were not performed blind to the conditions of the experiments.

### Reporting summary

Further information on research design is available in the Nature Portfolio Reporting Summary linked to this article.

## Data availability

All data supporting the findings of this study are available within the paper and its Supplementary Information. The Nanopore sequencing data used in this study were deposited into the European Nucleotide Archive with an accession number PRJEB61077.

## Code availability

The source code for m6Arp modelling and analyses is available at https://github.com/ksanggu/m6Arp.

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

## Acknowledgements

We thank Ohio Supercomputer Center (Ohio Supercomputer Center, Columbus, OH: http://osc.edu/ark:/19495/f5s1ph73) for their service. Jurkat, CEM-SS and GHOST cell lines were provided by the National Institutes of Health AIDS reagent programme (currently BEI resources, https://www.beiresources.org/). This research was funded by National Institutes of Health, grant numbers HG010318 (S.K.), HG010108 (S.K.), AI169659 (L.W.), AI170070 (L.W.) and GM058843 (P.L.); US Department of Defense, grant number HT9425-23-1-0582 (S.K.); and US Department of Energy, grant numbers 248445 (M.A.S.), DE-SC0023307 (M.A.S.). G.-E.L. is a recipient of the graduate fellowship from the C. Glenn Barber Fund Trust. The funders had no role in study design, data collection and analysis, decision to publish or preparation of the manuscript.

## Author contributions

S.K., A.B. and G.-E.L. conceptualized the study. S.K., A.B., G.-E.L., A.R., B.A. and L.W. designed experiments. A.B., G.-E.L., S.G., A.R., S.C., N.T., H.Y. and Sh.K. performed experiments. A.B., G.-E.L., S.G., A.R., A.A.M., C.K., O.V., B.A. and S.K. analysed data. S.K., A.B. and G.-E.L. wrote the manuscript. S.G., A.R., A.A.M., M.B.S., B.A. and L.W. reviewed and/or edited the manuscript before submission. S.K. coordinated and supervised the study.

## Competing interests

The authors declare no competing interests.

## Additional information

**Extended data** is available for this paper at https://doi.org/10.1038/s41564-024-01638-5.

**Correspondence and requests for materials** should be addressed to Sanggu Kim.

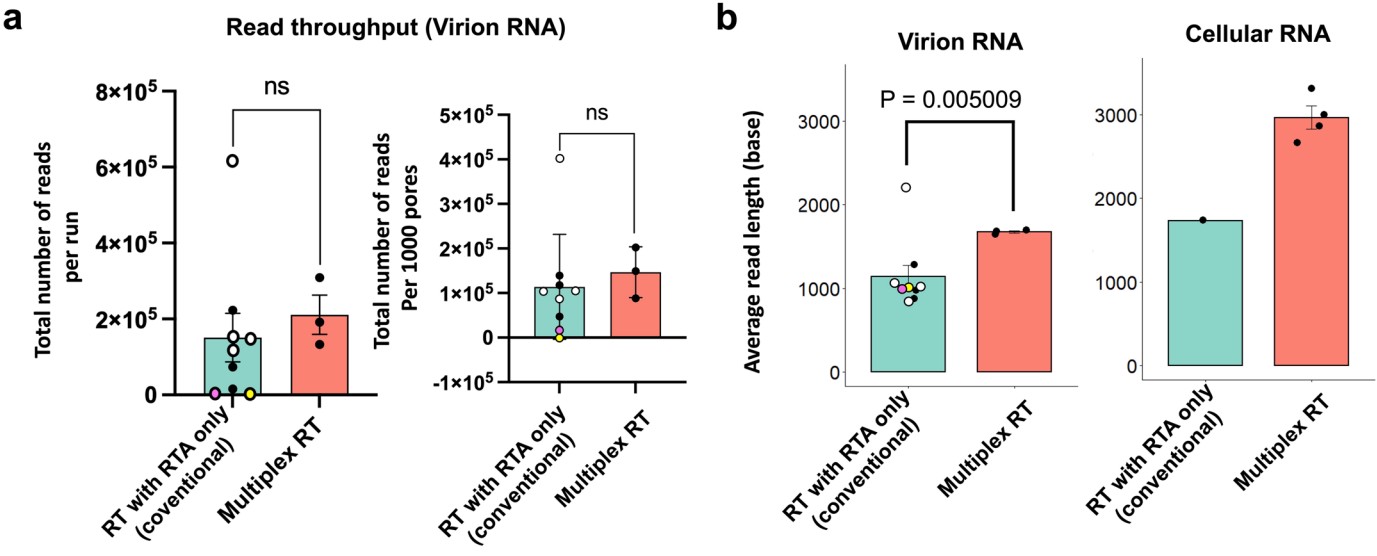

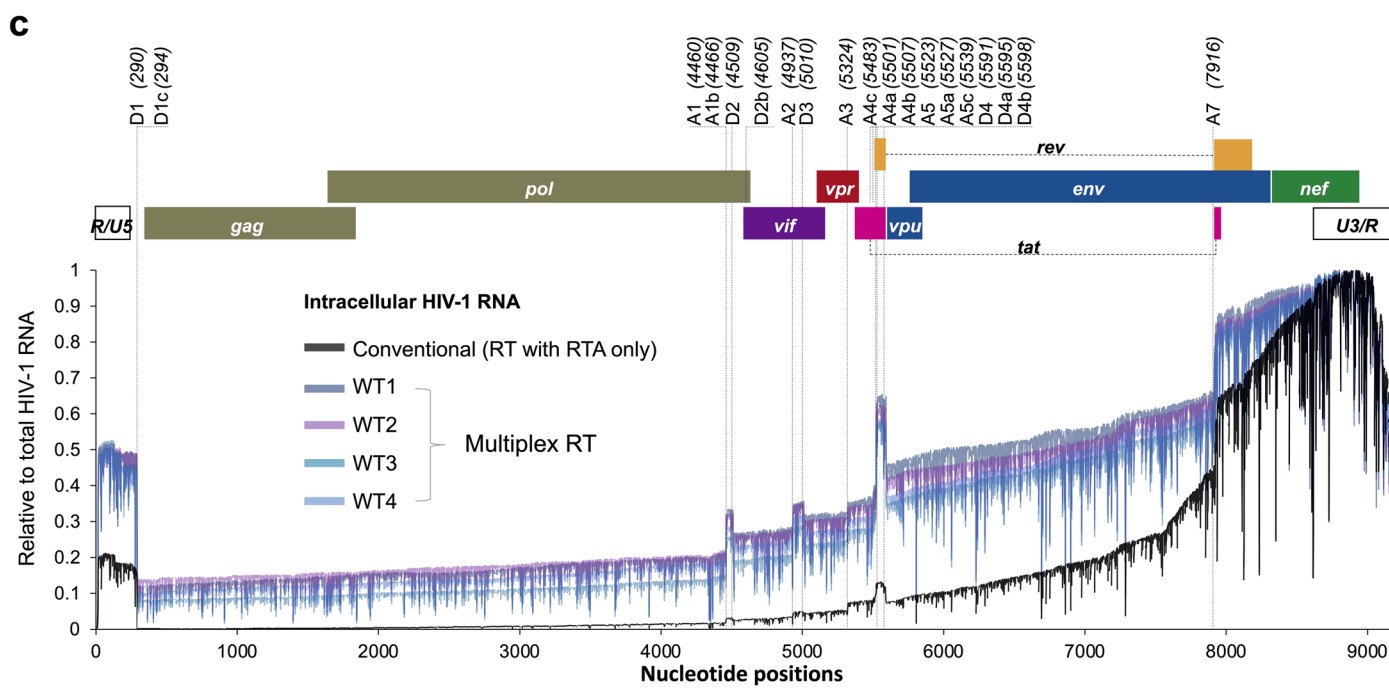

**Extended Data Fig. 1 | Multiplex RT improves DRS of full-length HIV-1 RNA.**
**(a-b)** The DRS read throughput (**a**) and average read lengths (**b**) were compared between DRS using new Multiplex RT method (orange box; n = 3 distinct samples) and DRS following the conventional ONT protocol (green boxes; n = 9 distinct samples). The multiplex RT significantly improved the average read length of virion RNAs compared to the conventional ONT protocol (p = 0.005; left panel; two-tailed T test), while maintaining the read throughput. Data are presented as mean values +/− standard errors. The conventional methods failed to generate more than 13 near-full-length HIV-1 virion RNA reads (> 0.01% of HIV-1 reads) in 8 out of 9 MinION runs. The multiplex RT methods showed the recovery ratios of more than 0.5% (457–898 reads per run) for virion RNAs (Supplementary Table 5). For intracellular HIV-1 RNAs, the full-length recovery rates with Multiplexed

RT reached to 34.9%, 31.5%, and 54.1% for unspliced (US), partially spliced (PS), and completely spliced (CS) RNAs, respectively (see Fig. 4a; n = 4 for Multiplex RT; n = 1 for conventional). The open circles in the green boxes are 4 repeated experiments using the DRS standard protocols; the colored circles represent DRS runs using modified RT conditions, including the conditions using TIGRT (pink circle) and marathon RT (yellow circle) that replaced SSIV RT of the conventional protocol. **(c)** Mapping of intracellular HIV-1 RNA. All 4 DRS runs using multiplex RT showed an improved and highly reproducible mapping onto the reference. Read depths (y-axis) of WT and mutant RNA reads are shown over the HIV-1 genome (NL4-3 strain). The positions of the splicing donor (D1-D4) and acceptor (A1-A7) sites are shown on the top.

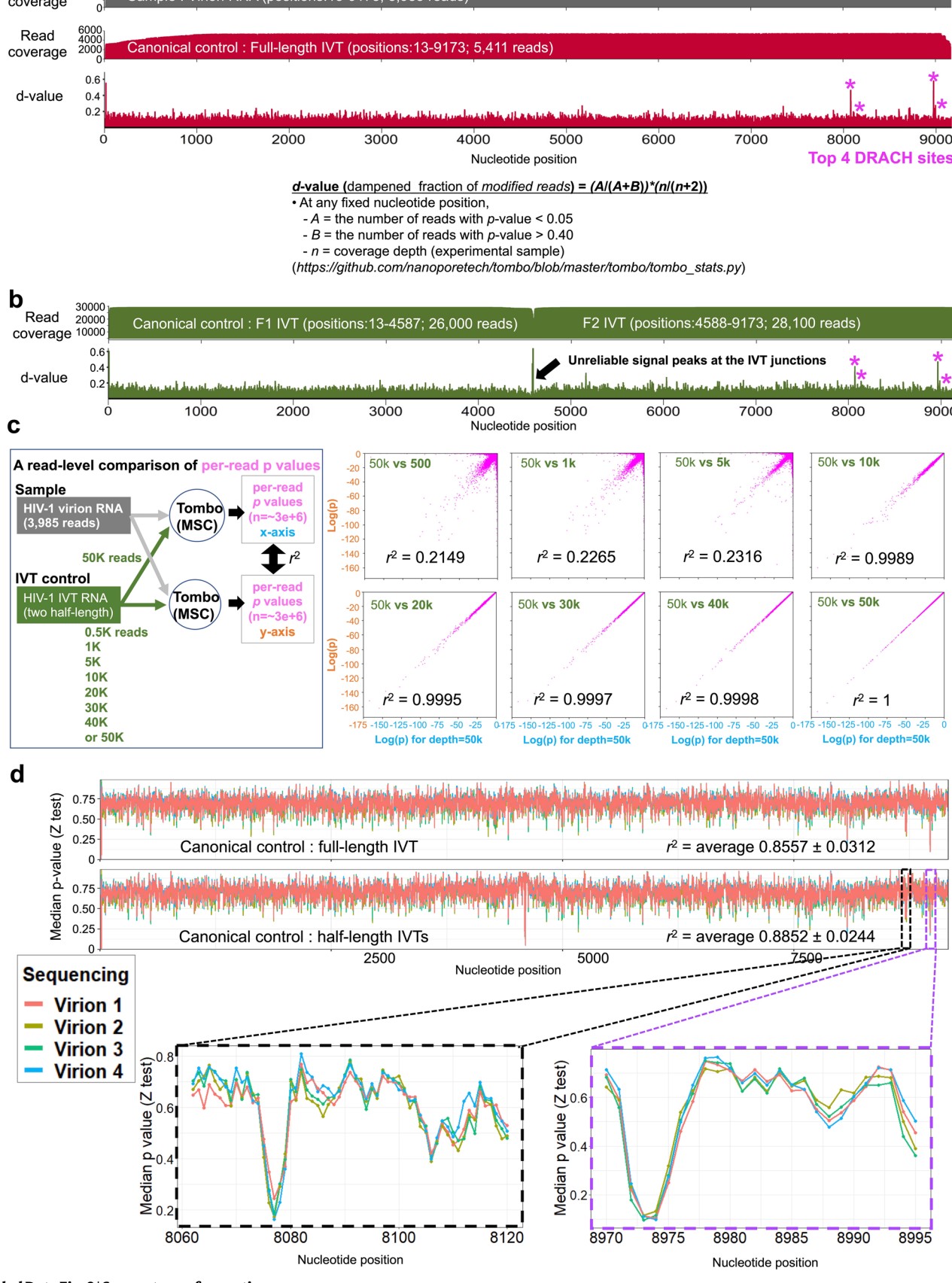

$d$-value (dampened fraction of *modified reads*) = $(A/(A+B))*(n/(n+2))$
- At any fixed nucleotide position,
  - $A$ = the number of reads with $p$-value < 0.05
  - $B$ = the number of reads with $p$-value > 0.40
  - $n$ = coverage depth (experimental sample)
  (*https://github.com/nanoporetech/tombo/blob/master/tombo/tombo_stats.py*)

**Extended Data Fig. 2 | See next page for caption.**

**Extended Data Fig. 2 | Highly reproducible Tombo analysis using HIV-1 IVT canonical controls.** (**a-b**) Tombo-'model sample compare' (MSC) analysis for full-length virion RNA reads (grey; total 3,985 reads of >8 Kb). We used 2 different sets of HIV-1 IVT canonical controls: full-length IVT RNAs (5,411 reads) (a) and half-length IVT RNAs (26,000 reads of F1 and 28,100 reads of F2) (b). Tombo-MSC analysis using these F1 and F2 reads show a similar read depth coverage (approximately 25K reads) for both F1 and F2 regions. The resulting 'd-value' plots show the 4 most prominent modification signals (purple asterisks) near the 3' end side of the genome. (**c**) Consistent generation of per-read $p$-values. We tested the effects of the read depth of the IVT canonical control on Tombo-MSC analysis. Tombo-MSC $p$-values for virion RNA were generated using 0.5 K, 1K, 5K, 10K, 20K, 30K, 40K, or 50K reads of F1 and F2 IVT canonical reads (the green box in the schematic view). Per-read per-position $p$-values of these datasets were directly compared to $p$-values of the identical position of the identical reads of the control datasets generated using 50K IVT (the grey box in the schematic view). Per-read $p$-values generated with >20K reads were highly reproducible ($r^2 > 0.999$) when compared to those of 50K IVT dataset. (**d**) Tombo-MSC analysis of 4 datasets of WT virion RNA DRS (Virions 1 to 4). Median per-read $p$-values generated with Tombo-MSC using 5,411 reads of full-length IVT RNA (Top panel) and 25K reads of half-length IVT RNAs (lower panel) were compared. Both datasets show highly reproducible $p$-values for the 4 virion datasets separately run on MinIONs. For these runs, 4 different virion RNA samples were prepared by separate transfection of HEK293T cells with pNL4-3 plasmids.

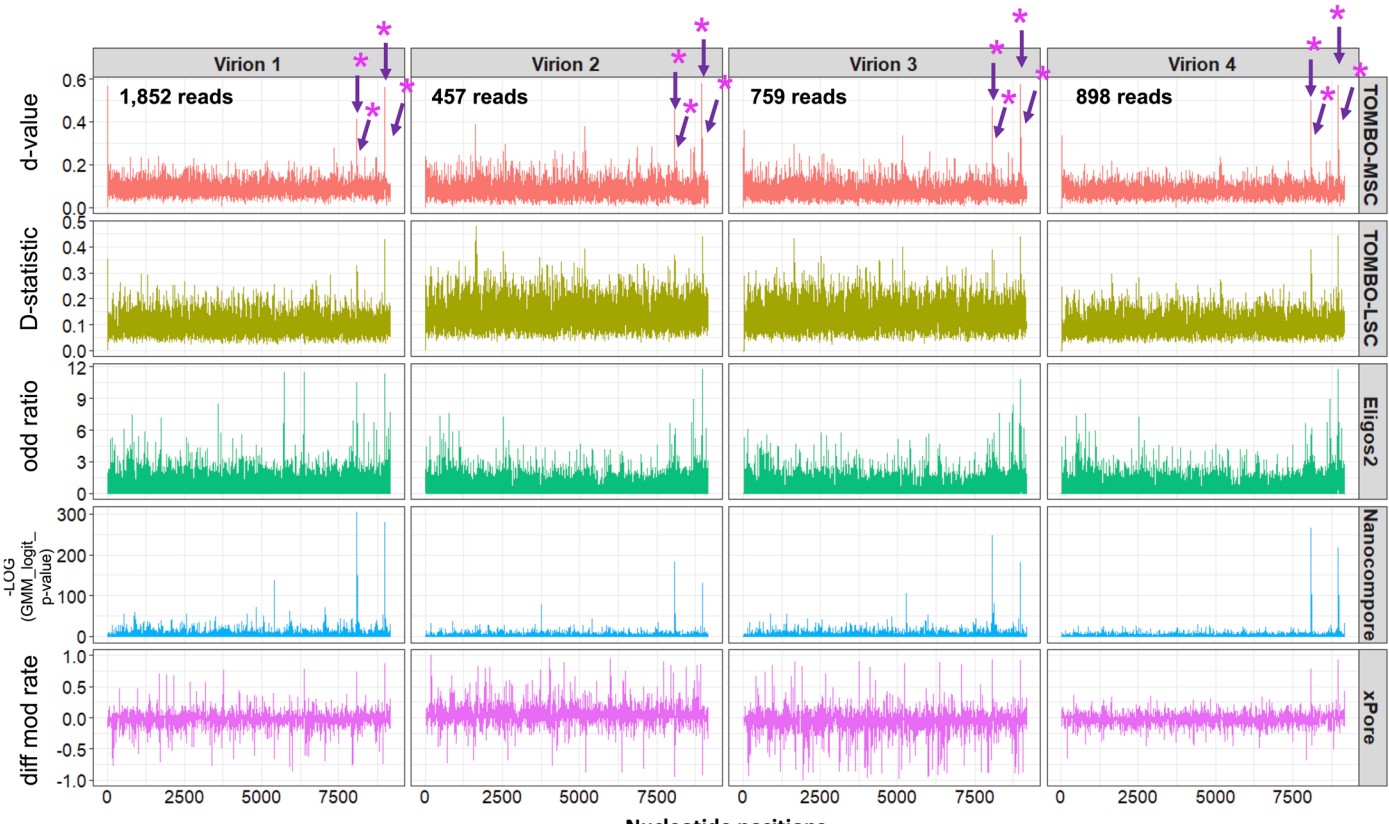

**Extended Data Fig. 3 | Reproducible detection of the 4 most prominent peaks.** 4 sets of HIV-1 virion RNA DRS runs were tested with Tombo-MSC (top panels), Tombo-level-sample-compare (Tombo-LSC; second from the top), Eligos2 (third from the top), Nanocompore (fourth from the top), and xPore (bottom panels). All tools identified the four m⁶A sites (purple asterisks) among the most prominent modification signals. Tombo-MSC, Eligos2 and Nanocompore results showed relatively more reproducible than Tombo-LSC and xPore.

# Comparison of HIV m⁶A RNA modification sites

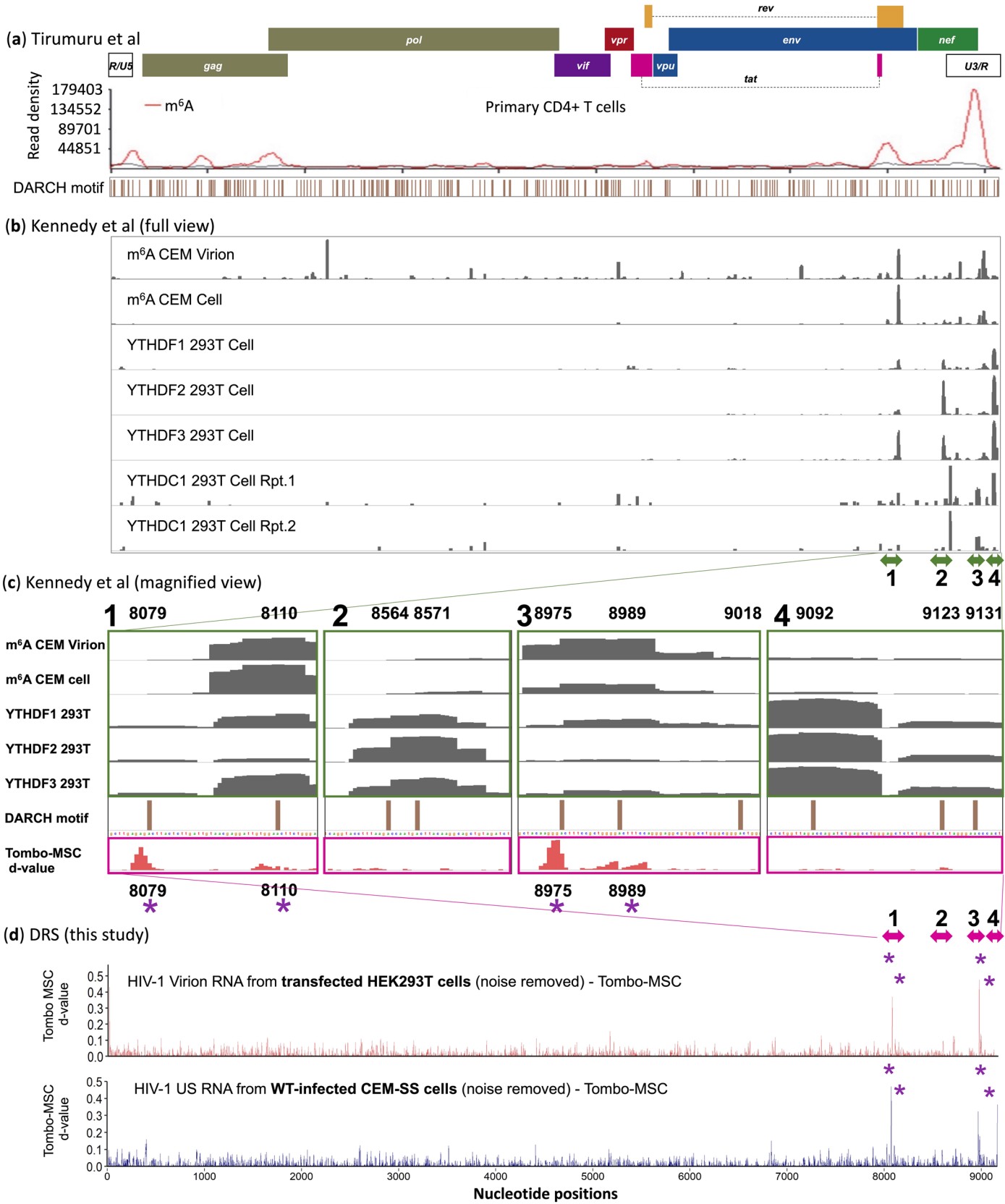

**Extended Data Fig. 4 | See next page for caption.**

**Extended Data Fig. 4 | Comparison of m6A sites between Nanopore DRS and Short-Read Sequencing data.** (**a**) m⁶A-seq analysis of primary CD4+ T Cells infected with HIV-1 NL4-3 strain from Tirumuru et al.[8] RNA fragments containing m⁶A methylation were aligned to the HIV-1 genome. (**b**) m⁶A sites and m⁶A-reader binding sites mapped by photo-crosslinking-assisted m⁶A sequencing (PA- m⁶A-seq) and photoactivatable ribonucleoside-enhanced crosslinking and immunoprecipitation (PAR-CLIP), respectively, from Kennedy et al.[6] and Tsai et al.[10] PA- m⁶A-seq Analysis of Virion RNA (Top Panel) and Cellular RNA (Second Panel from the Top) from HIV-1 NL4-3-Infected CD4+CEM-SS T Cells, as well as PAR-CLIP analysis of m⁶A reader binding sites, including YTHDF1-3 (middle three panels) and YTHDC1(bottom two panels) are shown. (**c**) A magnified View of the four major m6A sites predicted by Kennedy et al. Notably, areas 1 and 3 coincide with our Nanopore DRS m⁶A peaks (purple asterisks in the bottom panel). (**d**) Potential Modification Sites in Nanopore DRS Data. In this section, we present potential modification sites detected in Nanopore DRS data for HEK293T cells transfected with pNL4-3 (upper panel) and those for WT-infected CD4+ T cells (CEM-SS) 96-hour post infection (lower panel; result of single cycle infection, see methods for details).

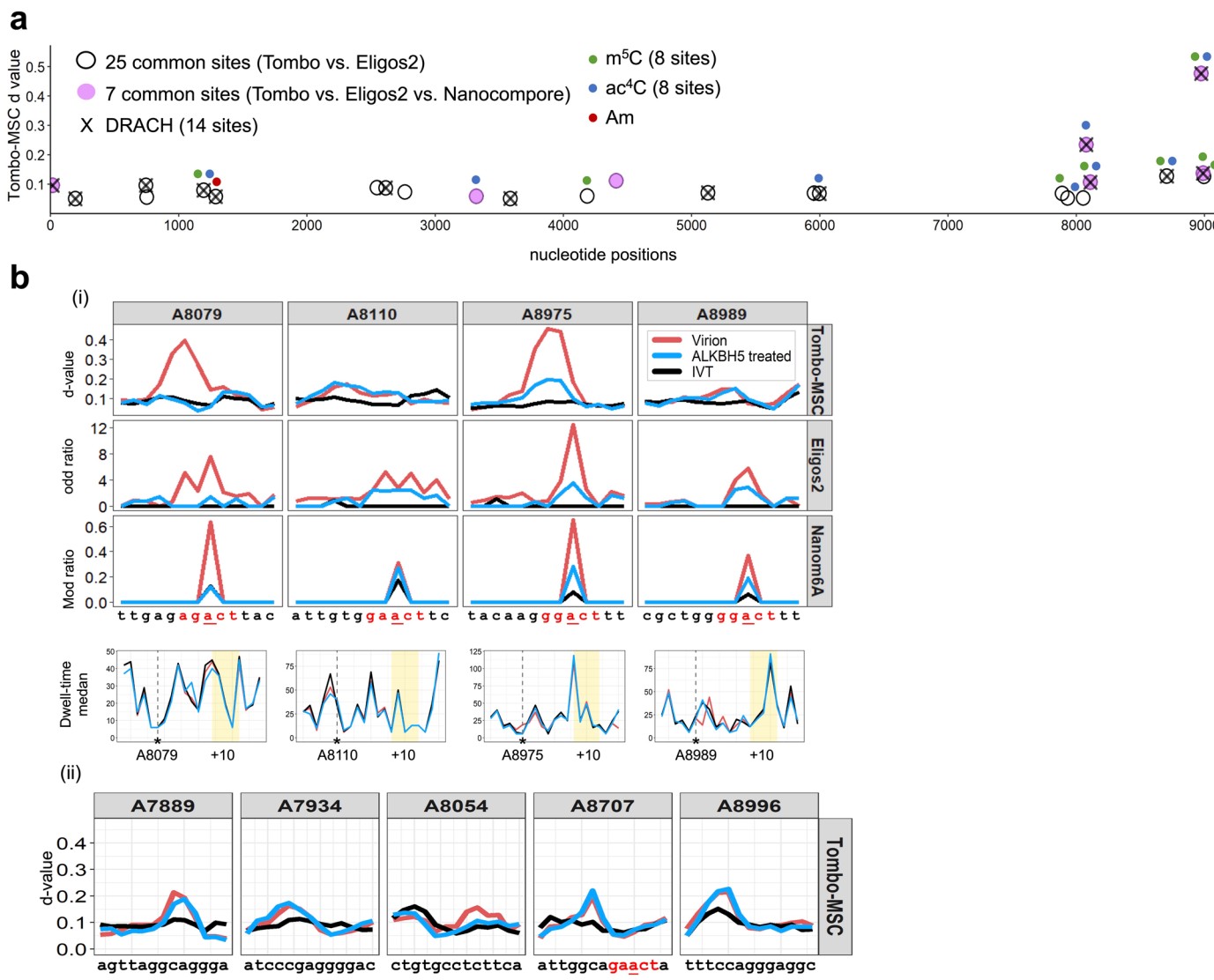

**Extended Data Fig. 5 | Evaluation of DRS-detected modification sites. (a)** The top panel shows the 25 common sites between Tombo and Eligos2 data (circles) and 7 sites common in all three datasets (purple circles) (see Methods and Supplementary Fig. 7a). The 25 common sites show a significant correlation with DRACH sites (denoted by X) and previously m6A-mapped areas and m6A-reader binding sites (see Supplementary Table 4). Other published modification sites, including 5-methylcytosine (m5C; green dots above the circles) N4-acetylcytidine (ac4C; blue dots above the circles) showed no significant correlation (see Supplementary Table 4 -i-). There was only one event where the 25 common sites overlap with previously published 2'-O-methylation sites (Am; red dot above the circles; it overlaps with one of the 14 common DRACH sites). **(b)** The modification signals of ALKBH5-treated RNA (blue lines) and PBS-treated virion RNA (red lines) are shown. The black line denotes the IVT-subset control. (i) The DRS signals of 13 nucleotides surrounding the four DRACH sites (A8079, A8110, A8975, and A8989) are compared. Signals from Tombo-MSC (d-values; top), Eligos2 (second from the top), nanom6A (third from the top), and dwell time (fourth from the top) are shown. Dwell time differences were measured at both the putative m6A site and its 10 base downstream. (ii) The signal reduction for non-DRACH sites, including A7889, A7934, A8054, A8707 and A8996 was relatively mild or undetectable.

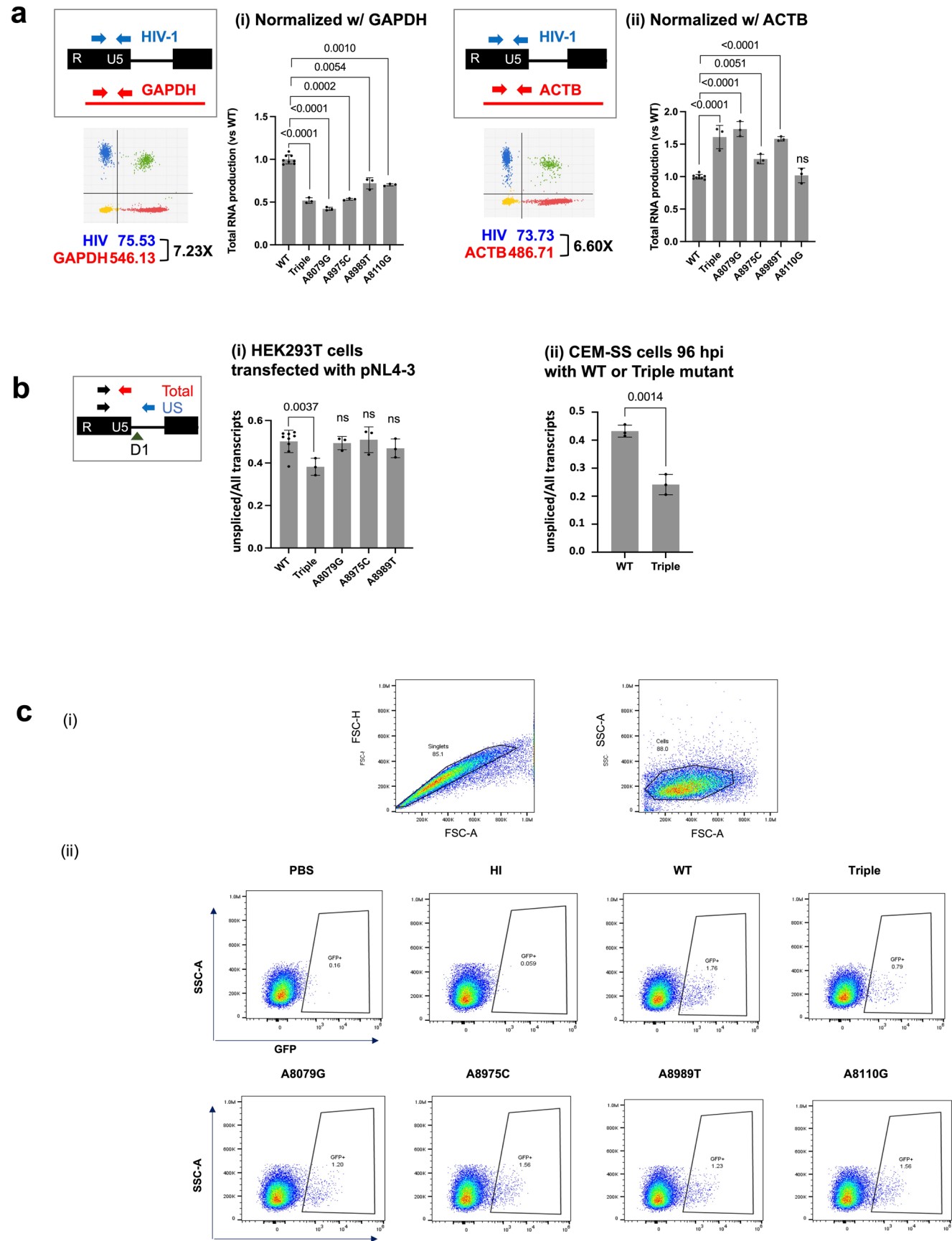

**Extended Data Fig. 6 | See next page for caption.**

**Extended Data Fig. 6 | Knocking out all three dominant m6As, but not the single m⁶A, affects HIV-1 fitness. (a)** Digital PCR was used to measure total HIV-1 RNA production normalized by glyceraldehyde-3-phosphate dehydrogenase (GAPDH) (i) or actin beta (ACTB) RNAs (ii). The relative ratios of total HIV-1 RNA and GAPDH RNA (i) or ACTB RNA (ii) were simultaneously measured by digital PCR, with total HIV RNA measured targeting the 5′ U5 region. These showed mixed results depending on the controls used, but Nanopore DRS data (Fig. 4d–i) showed no difference in Triple mutant (Triple). Data are presented as mean values +/− standard deviation. (two-tailed T test; WT: n = 3 for each comparing set (3 experiments, total n = 9); Triple, A8079G, A8975C, and A8989T: n = 3; biologically independent samples) **(b)** Approximately half of intracellular HIV-1 RNA is unspliced (US) RNA. The US and total HIV-1 RNA were measured by digital PCR.

HEK293T cells were analyzed 72 hour post-transfection with pNL4.3 plasmids (i). Data are presented as mean values +/− standard deviation. (two-tailed T test; WT: n = 3 for each comparing set (3 experiments, total n = 9); Triple, A8079G, A8975C, and A8989T: n = 3; biologically independent samples) Jurkat T cells were analyzed 96 hour post infection (hpi) with MOI of 1-2 of HIV-1$_{NL4.3}$ (ii). Data are presented as mean values +/− standard deviation. (two-tailed T test; WT: n = 3 triple mutant: n = 3; biologically independent samples) **(c)** Flow cytometry analysis of GHOST cell infection. (i) Flow cytometry gating strategies showing the single cell-gating (left panel) and a removal of outliers (right panel). (ii) Gating of GFP+ cells. Gates were determined based on the negative control (PBS treated) and the positive control (WT-infected cell) data. Infection assays were triplicated.

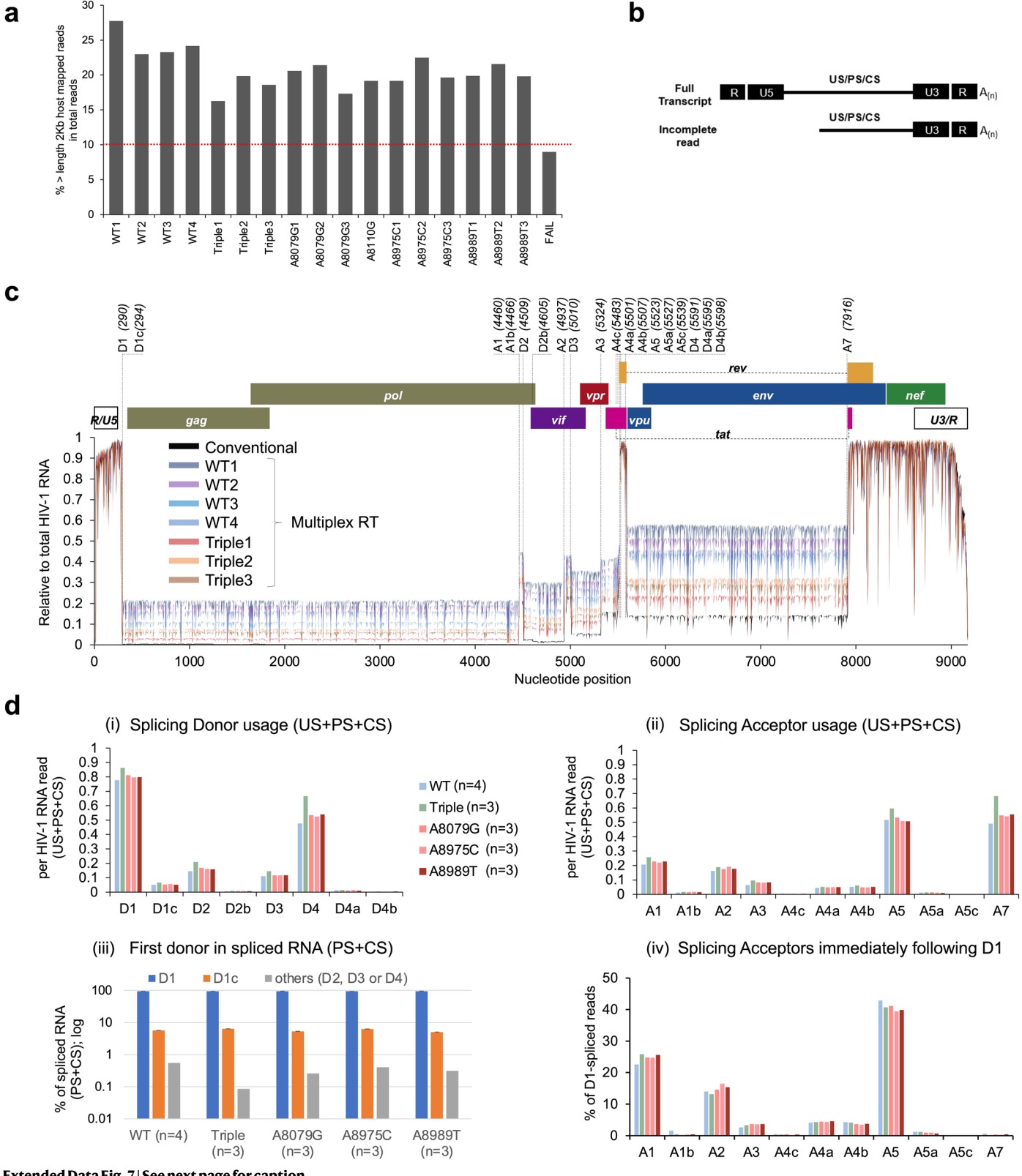

**Extended Data Fig. 7 | See next page for caption.**

**Extended Data Fig. 7 | Analysis of HIV-1 RNA alternative splicing. (a)** DRS cellular RNA runs occasionally showed poor read length distributions (see FAIL in a & b). When the fraction of > 2 Kb RNAs is less than 10% of the total reads, these samples were considered unsuitable and excluded from HIV-1 splicing analysis. The length distributions of cellular RNA are shown for 4 WT, 3 triple mutants, 10 single mutants (A8079G, A8110G, A8975C and A8989T), and FAIL. **(b)** Selection of full-length HIV-1 intracellular RNAs. Any HIV-1 RNA reads that lack the U5 sequence are removed from the analyses; only full-length CS, PS, and US reads were used for the analysis. **(c)** The relative fraction of intracellular HIV-1 RNAs (full-length CS, PS, and US) mapped onto the reference genome (pNL4-3). A total of 196 exon combinations, including the major 53 viral RNA isoforms, utilizing various combinations of splicing donors (D1-D4) and acceptors (A1-A7) were

identified. WT1-4 and Triple 1-3 Data were produced via multiplex RT method. Conventional DRS results following ONT's standard DRS protocol (using SSIV and RTA for RT) are shown for comparison. **(d)** splicing donor and acceptor site usage. (i-ii) bar-graphs showing the relative usage rates per HIV-1 RNA (US+PS+CS combined; y-axis) for Splicing Donor usage (i) and Splicing Acceptor usage (ii). (iii) First donor sites. Nearly all (93.5%-94.6%) spliced RNA uses D1 donor; 5.3% to 6.3% use D1c; and less than 0.5% use other donors (D2, D3 or D4) for the first splicing. (iv) The acceptor usage rates following the D1 donor usage (% of D1, y-axis) during the first splicing event. WT (n = 4 distinct samples), triple mutants (n = 3 distinct samples) and single mutants (A8079G, A8975G, A8989T; n = 3 samples) are shown.

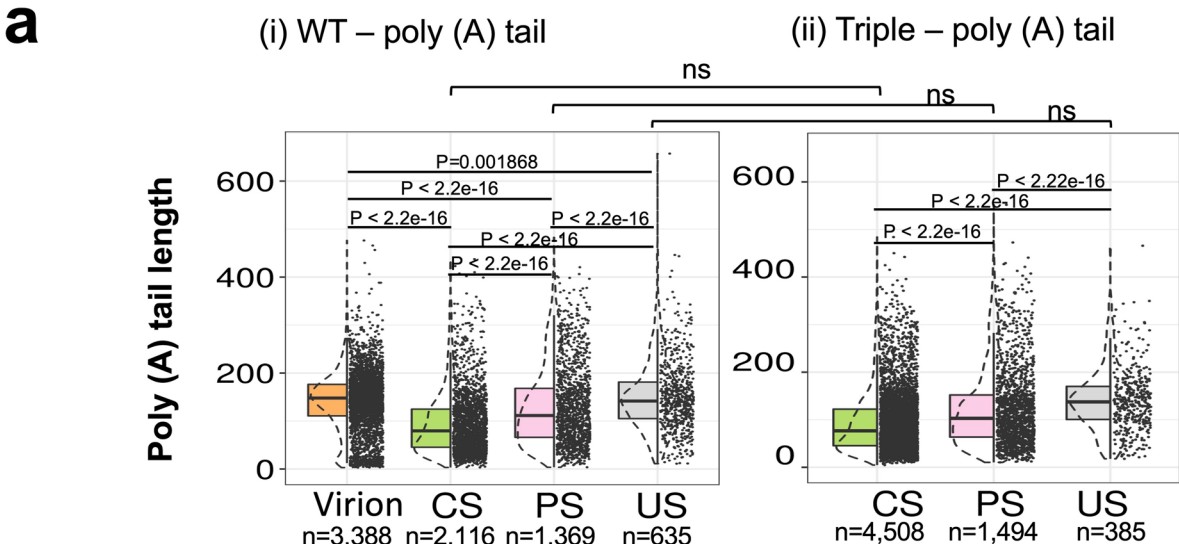

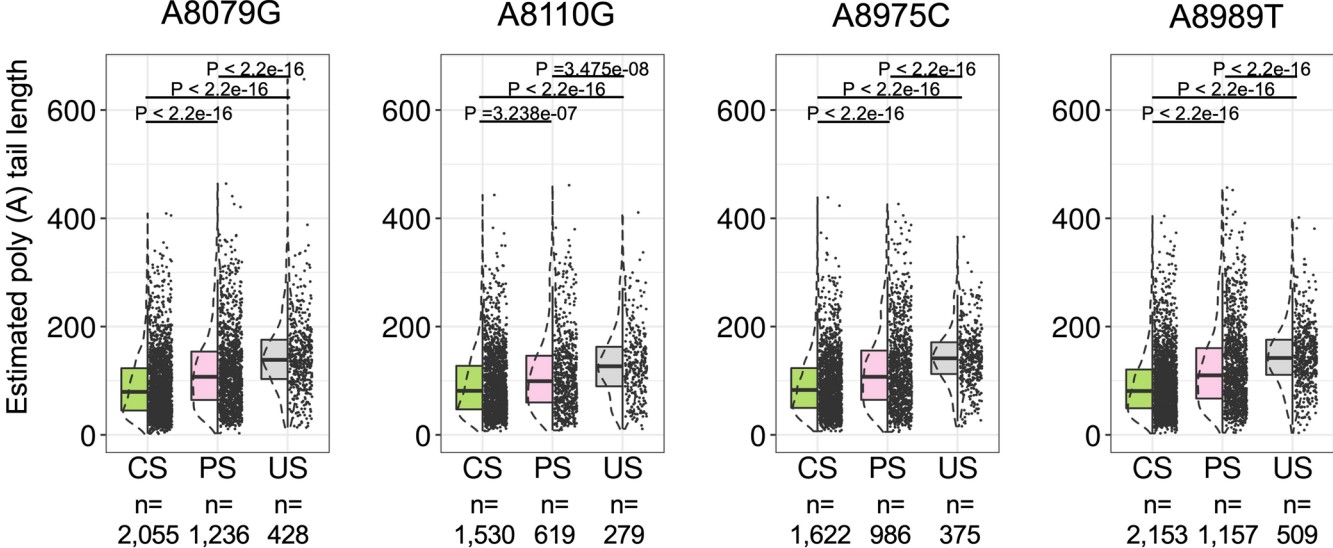

**Extended Data Fig. 8 | Analysis of HIV-1 RNA 3′ poly (A) tails. (a)** The lengths of 3′ polyadenylation, poly (A) tail, varied significantly among CS, PS, US and virion RNAs, but there was no significant difference between WT (i) and triple mutants (ii) (two-tailed kolmogorov-smirnov test: box, first to last quartiles; whiskers, 1.5X interquartile range; center line, median; points, individual data values; violin, distribution of density). **(b)** Poly (A) length distribution of single mutants. They also showed significant differences among CS, PS and US RNA (two-tailed kolmogorov-smirnov test: box, first to last quartiles; whiskers, 1.5X interquartile range; center line, median; points, individual data values; violin, distribution of density).

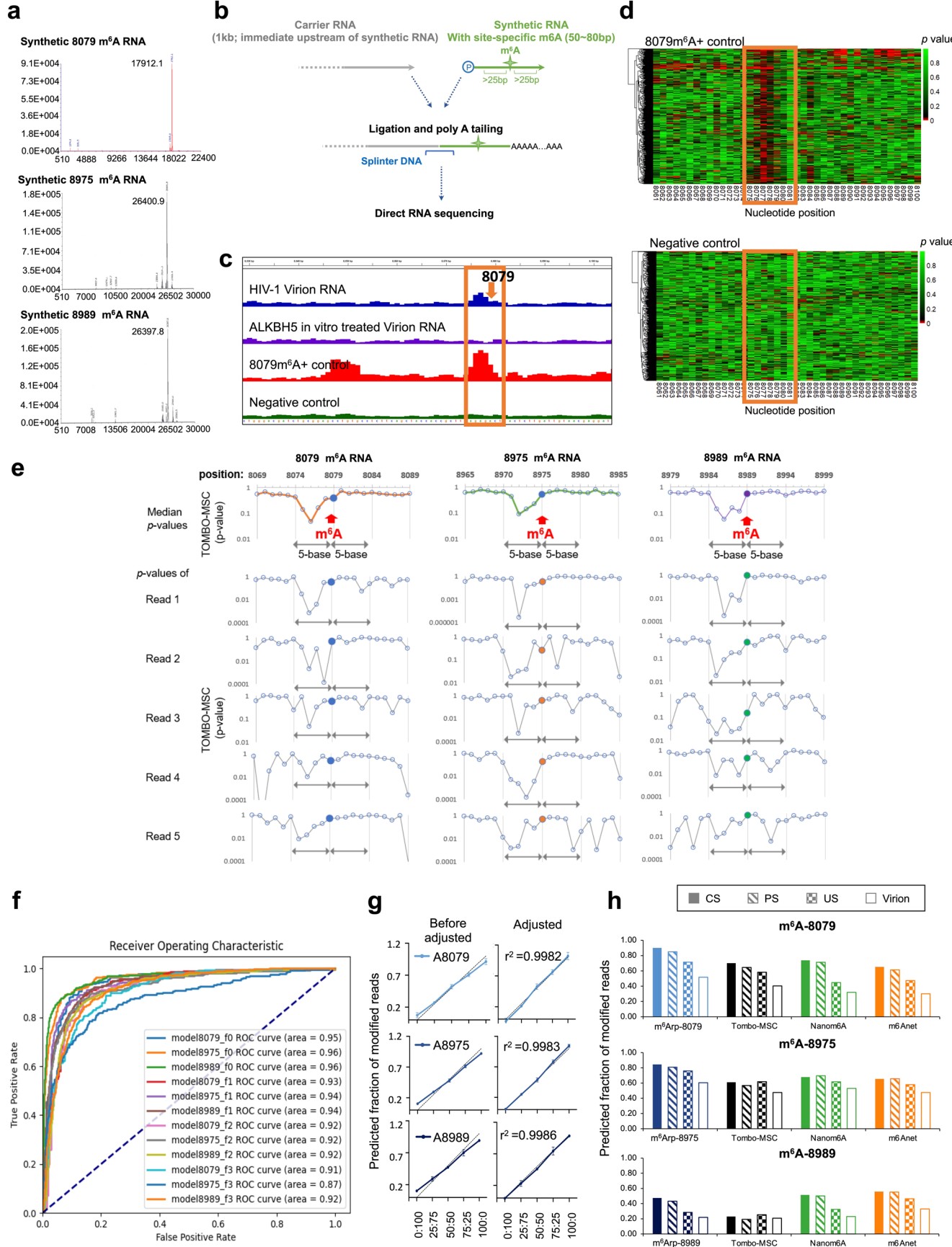

**Extended Data Fig. 9 | See next page for caption.**

**Extended Data Fig. 9 | Development of binary classification models (m⁶Arp) for accurate detection of three dominant m⁶As at the read level.** (**a-b**) Three synthetic RNAs with an m⁶A modifications at positions 8079, 8975, or 8989 were ligated to carrier RNA to generate positive control data. (a) Mass spectrometry data show nearly complete m⁶A modification in synthetic RNA controls (Horizon Discovery Ltd.). (**b**) A schematic view of generating positive control RNAs (see Methods). (**c**) Tombo-MSC d-values near the position 8079 are shown as an example of DRS data for control RNAs. (**d**) The heatmap views show per-read $p$-value distributions for 8079m⁶A+ control (upper panel) and negative control (lower panel). Each row represents a different RNA molecule. (**e**) A shift of $p$-values to upstream of the modification site. The top panels show median $p$-values near the 8079 (left), 8975 (middle) and 8989 (right) sites with notable $p$-value peaks spanning −4 to +1 positions ($N_{-4}, N_{-3}, N_{-2}, N_{-1}, A_0, N_1$; where $A_0 = $ m⁶A) of the m⁶A site. The patterns were consistent with previous cellular transcript data[48] Per-read $p$-value patterns for five chosen reads (Read 1 to 5). Each read showed substantial variations and irregularities, but nevertheless, all reads demonstrated robust differences in the $p$-value patterns compared to those of the negative controls (see Supplementary Table 8 for all available data in the data repository). (**f**) Optimizing m⁶Arp models. Tombo-MSC per-read $p$-values were prepared with fisher = 0, 1, 2 or 3 options. Fisher = 0 (no data fusion) option (model8079_f0, model8975_f0, and model8989_f0) showed the best area under the receiver operating characteristics curve (AUROC) (see Supplementary Table 7). (**g**) Our models also showed a strong linearity of quantification ($R^2 > 0.9982$), after adjustments for FNR and FPR ($r^2 = 0.9982$ to $0.9987$; see Methods). (**h**) All read-level quantification tools, including m⁶Arp-models (light blue), Tombo d-value (dark blue), Nanom6A (green) and m6ANet (orange), showed relatively low m⁶A stoichiometry in unspliced RNA (virion and US RNA) compared to those in spliced (CS and PS) RNA for all the 3 sites, including A8079 (top panels), A8975 (middle panels), and A8989 (bottom panels).

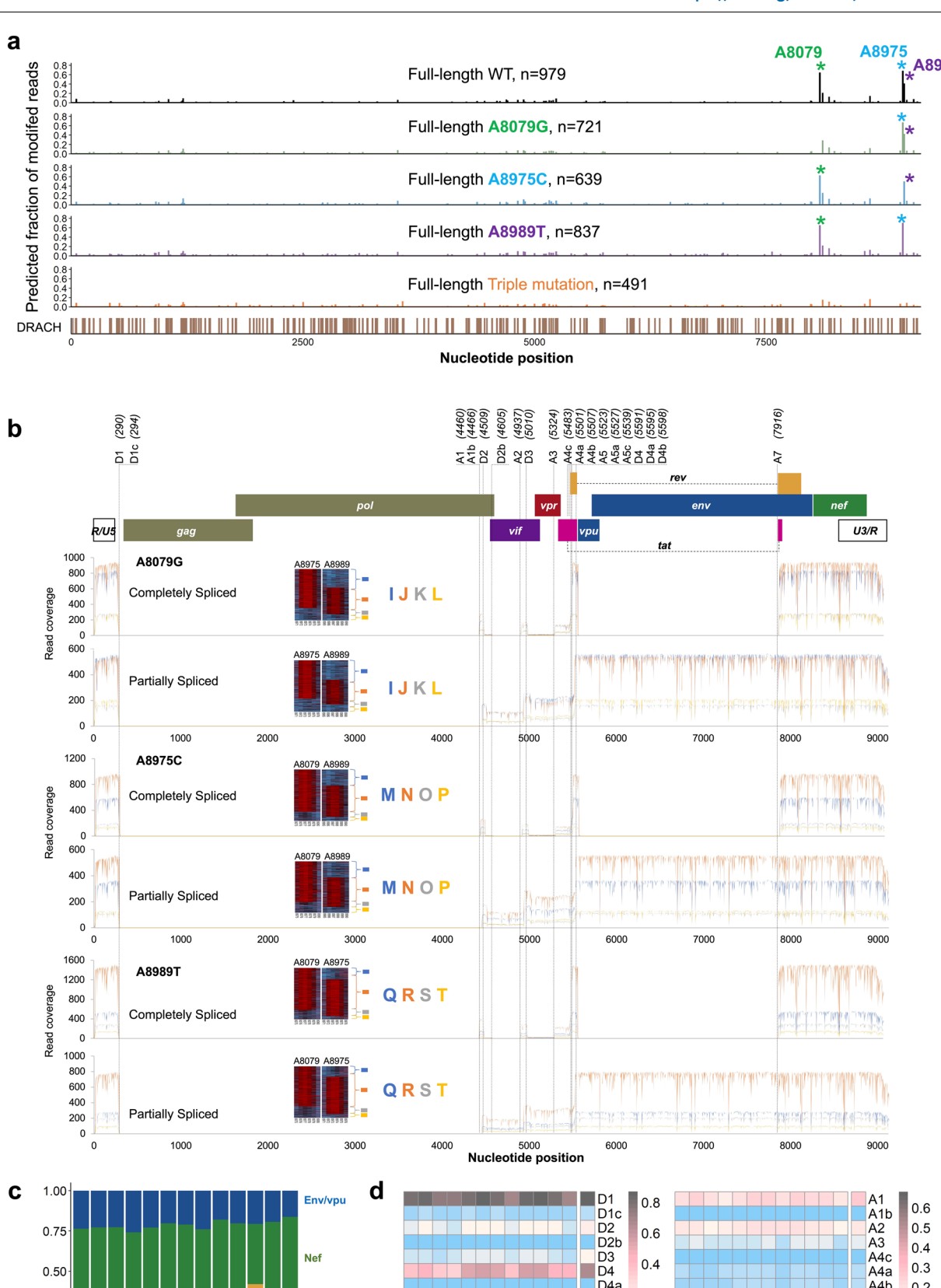

**Extended Data Fig. 10 | See next page for caption.**

**Extended Data Fig. 10 | Single-molecule-level analysis reveals the functional redundancy of the m6As.** (**a**) Nanom6A analysis of full-length DRS reads from WT, single mutant, and triple mutant HIV-1 RNAs showed no notable signal changes in the DRACH sites in the full genome compared to the WT landscape. The dominant m⁶As at A8079, A8975, and A8989 are indicated by green, blue, and purple asterisks, respectively. Mutations effectively removed m⁶A signals at the target site. The DRACH sites are shown in the bottom panel. (**b**) Splicing patterns of RNA subspecies with distinct m⁶A ensembles for A8079G (top two panels), A8975C (middle two panels), and A8989T (bottom two panels). Four RNA subspecies (blue, orange, grey, and yellow, with distinct m⁶A ensembles) of completely spliced (CS) and those of partially spliced (PS) RNA were mapped onto the HIV-1 reference sequence (NL4-3 strain) to show their splicing patterns. All RNA subspecies showed indistinguishable or moderate differences in the usage of splicing donors and acceptors. (**c-d**) All subspecies from the three mutants (including subspecies I, J, K, and L, from A8079G; subspecies M, N, O, and P from A8975C; and subspecies Q, R, S, and T from A8989T) showed moderate differences in the production of protein-specific mRNAs (c) and donor and acceptor usage rates (d). Triple mutant data are shown for comparison (arrowheads).

# Reporting Summary

## Statistics

For all statistical analyses, confirm that the following items are present in the figure legend, table legend, main text, or Methods section.

| n/a | Confirmed | |
|---|---|---|
| ☐ | ☒ | The exact sample size ($n$) for each experimental group/condition, given as a discrete number and unit of measurement |
| ☐ | ☒ | A statement on whether measurements were taken from distinct samples or whether the same sample was measured repeatedly |
| ☐ | ☒ | The statistical test(s) used AND whether they are one- or two-sided *Only common tests should be described solely by name; describe more complex techniques in the Methods section.* |
| ☒ | ☐ | A description of all covariates tested |
| ☐ | ☒ | A description of any assumptions or corrections, such as tests of normality and adjustment for multiple comparisons |
| ☐ | ☒ | A full description of the statistical parameters including central tendency (e.g. means) or other basic estimates (e.g. regression coefficient) AND variation (e.g. standard deviation) or associated estimates of uncertainty (e.g. confidence intervals) |
| ☐ | ☒ | For null hypothesis testing, the test statistic (e.g. $F$, $t$, $r$) with confidence intervals, effect sizes, degrees of freedom and $P$ value noted *Give P values as exact values whenever suitable.* |
| ☒ | ☐ | For Bayesian analysis, information on the choice of priors and Markov chain Monte Carlo settings |
| ☒ | ☐ | For hierarchical and complex designs, identification of the appropriate level for tests and full reporting of outcomes |
| ☐ | ☒ | Estimates of effect sizes (e.g. Cohen's $d$, Pearson's $r$), indicating how they were calculated |

*Our web collection on statistics for biologists contains articles on many of the points above.*

## Software and code

Policy information about availability of computer code

| | |
|---|---|
| Data collection | QuantStudio 3D Digital PCR System was used for RT-PCR assays; Attune™ NxT flow cytometer (Thermo Fisher Scientific) for flow cytometry data collection; MinKNOW GUI (v3 or later; Nanopore technology) was used for sequencing data collection. |
| Data analysis | Multi-fast5 reads were base-called by guppy (version 3.2.8 or higher) and converted to single-read fast5s using the Oxford Nanopore Technologies API, ont_fast5 (v3.3.0). Fastqs were aligned with minimap2 (v2.24), and processed with SAMtools (v.1.6), NanoFilt (v2.7.1), and bedtools (v.2.25.0). The HIV-1 B subtype sequences for sequence conservation analysis were extracted from the HIV sequence database (https://www.hiv.lanl.gov/) and visualized using ggseqlogo R package (v.0.1). Tombo (v1.5.1), Eligos2 (v2.0.0), Nanocompore (v1.0.4), xPore (v2.1), Nanom6A (v2.0), and m6Anet (v-1.1.1) were used for RNA modification site calling. HIV-1 RNA splicing and poly(A) tail length analysis were conducted using minimap2 (v2.24), SAMtools (v.1.6), and Nanopolish (v.0.14.0). Blots were analyzed by ImageJ (v.1.53). Statistical analysis performed using GraphPad Prism9(v.9.5.0) or R package (v.4.0.2). |

For manuscripts utilizing custom algorithms or software that are central to the research but not yet described in published literature, software must be made available to editors and reviewers. We strongly encourage code deposition in a community repository (e.g. GitHub). See the Nature Portfolio guidelines for submitting code & software for further information.

## Data

Policy information about availability of data

All manuscripts must include a data availability statement. This statement should provide the following information, where applicable:

- Accession codes, unique identifiers, or web links for publicly available datasets
- A description of any restrictions on data availability
- For clinical datasets or third party data, please ensure that the statement adheres to our policy

All data supporting the findings of this study are available within the paper and its Supplementary Information. The Nanopore sequencing data used in this study were deposited into the European Nucleotide Archive (ENA) with an accession number PRJEB61077. The processed sequence data for each figure are available in Supplementary Information. The HIV-1 B subtype sequences corresponding to A8079, A8110, A8975 and A8989 of the NL4-3 strain (RNA) were extracted from the HIV sequence database (https://www.hiv.lanl.gov/).

## Human research participants

Policy information about studies involving human research participants and Sex and Gender in Research.

| Reporting on sex and gender | NA |
| --- | --- |
| Population characteristics | NA |
| Recruitment | NA |
| Ethics oversight | NA |

Note that full information on the approval of the study protocol must also be provided in the manuscript.

# Field-specific reporting

Please select the one below that is the best fit for your research. If you are not sure, read the appropriate sections before making your selection.

☒ Life sciences   ☐ Behavioural & social sciences   ☐ Ecological, evolutionary & environmental sciences

For a reference copy of the document with all sections, see nature.com/documents/nr-reporting-summary-flat.pdf

# Life sciences study design

All studies must disclose on these points even when the disclosure is negative.

| Sample size | We provide triplicated (or quadruplicated) experimental data of biologically independent samples. No statistical methods were used to pre-determine sample sizes, but the sample size of n=3 or n=4 routinely provide sufficient statistical power (when present) in our study utilizing accurate and highly reproducible assays and molecular biology experiments. These number of samples are commonly used in molecular biology publications to provide statistical conclusions, as well as to address the rigor and reproducibility. |
| --- | --- |
| Data exclusions | Low quality sequence reads were excluded by default QC threshold by MinKNOW (Nanopore technology). DRS cellular RNA runs occasionally showed poor read length distributions (Extended Data Fig.7a); when the fraction of > 2 Kb RNAs is less than 10% of the total reads, these samples were considered unsuitable and excluded from HIV-1 splicing analysis. For HIV-1 alternative splicing analysis, only the full-length reads were used (Fig. 4a). |
| Replication | Our data were highly reproducible in our repeated experiments using RNA samples that are independently prepared (n=3 or 4). |
| Randomization | Experimental groups were determined based on the experimental hypothesis (e.g. the impact of mutations or RNA isoforms) and all the experimental data and sequencing data that pass the data exclusion criteria were used without any additional selection. |
| Blinding | All the sequencing data that pass the data exclusion criteria were used without any additional selection. This study does not involve human studies. Data collection and analysis were not performed blind to the conditions of the experiments. |

# Reporting for specific materials, systems and methods

We require information from authors about some types of materials, experimental systems and methods used in many studies. Here, indicate whether each material, system or method listed is relevant to your study. If you are not sure if a list item applies to your research, read the appropriate section before selecting a response.

## Materials & experimental systems

| n/a | Involved in the study |
|---|---|
| ☐ | ☒ Antibodies |
| ☐ | ☒ Eukaryotic cell lines |
| ☒ | ☐ Palaeontology and archaeology |
| ☒ | ☐ Animals and other organisms |
| ☒ | ☐ Clinical data |
| ☒ | ☐ Dual use research of concern |

## Methods

| n/a | Involved in the study |
|---|---|
| ☒ | ☐ ChIP-seq |
| ☐ | ☒ Flow cytometry |
| ☒ | ☐ MRI-based neuroimaging |

# Antibodies

| Antibodies used | Antibodies used for western blots are;<br>Anti-N6-methyladenosine (m6A) antibody (Abcam; Cat#ab208577) 1:1000 dilution<br>Anti-Gag(p24) (NIH AIDS Reagent; Cat#ARP-6458) 1:1000 dilution<br>Anti-Vif (NIH AIDS Reagent;cat# ARP-6459) 1:500 dilution<br>Anti-gp41(NIH AIDS reagent; cat# ARP-11391)1:500 dilution<br>Anti-Mouse HRP (Promega; cat#W4021)1:5000 dilution<br>Anti-GAPDH (Abcam; cat#ab8245) 1:1000 dilution |
|---|---|
| Validation | All antibodies were validated by respective vendors and obtained from the NIH AIDS Reagent program. These antibodies are commonly used and have been validated in multiple publications.<br>Anti-N6-methyladenosine (m6A) antibody (Abcam; Cat#ab208577) suitable for Northwestern, IP, Southern Blot. https://www.abcam.com/products/primary-antibodies/n6-methyladenosine-m6a-antibody-17-3-4-1-ab208577.html<br>Anti-Gag(p24) (NIH AIDS Reagent; Cat#ARP-6458),Anti-Vif (NIH AIDS Reagent;cat# ARP-6459) and Anti-gp41(NIH AIDS reagent; cat# ARP-11391) were obtained from NIH AIDS reagent Program and validated.<br>Anti-Mouse HRP (Promega; cat#W4021) suitable for WB. https://www.promega.com/products/protein-detection/primary-and-secondary-antibodies/anti_mouse-igg-h-and-l-hrp-conjugate/?catNum=W4021<br>Anti-GAPDH (Abcam; cat#ab8245) suitable for WB, ICC/IF. https://www.abcam.com/products/primary-antibodies/gapdh-antibody-6c5-loading-control-ab8245.html |

# Eukaryotic cell lines

Policy information about cell lines and Sex and Gender in Research

| Cell line source(s) | HEK293T cells were purchased from ATCC; GHOST CXCR4+CCR5+ cells, Jurkat, and CEM-ss cell lines were from the NIH-supported HIV reagent program. |
|---|---|
| Authentication | HEK293T cells, GHOST CXCR4+CCR5+ cells, Jurkat, and CEM-ss cell lines are commonly used commercial cell lines distributed by ATCC and NIH-supported HIV reagent program, respectively; these cell lines were not further authenticated. |
| Mycoplasma contamination | HEK293T cells, GHOST CXCR4+CCR5+ cells, Jurkat, and CEM-ss cell lines used in this study were not tested for mycoplasma contamination. |
| Commonly misidentified lines<br>(See ICLAC register) | No commonly misidentified cell lines were used in the study |

# Flow Cytometry

## Plots

Confirm that:

☒ The axis labels state the marker and fluorochrome used (e.g. CD4-FITC).

☒ The axis scales are clearly visible. Include numbers along axes only for bottom left plot of group (a 'group' is an analysis of identical markers).

☒ All plots are contour plots with outliers or pseudocolor plots.

☒ A numerical value for number of cells or percentage (with statistics) is provided.

## Methodology

| Sample preparation | An equal amount of virus stock (pg) was used to infect GHOST cells in 6-well plates. After 48 h post-infection, the cells were washed with PBS three times and fixed with 3.7% formaldehyde for 10 minutes. |
|---|---|
| Instrument | The GFP expressions for all samples were acquired by the Attune NxT flow cytometer (Thermo fisher scientific) |
| Software | GFP expressions were analyzed by the FlowJo software (v 10.8.0) (BD biosciences) |

Cell population abundance | The cell population abundance is shown in the relevant figures. Sorting was based on GFP.

Gating strategy | After dead cell removal based on the FSC/SSC gating, GFP+ cells were determined based on the gating two separate and distinct populations were visible in the relevant figures.

☒ Tick this box to confirm that a figure exemplifying the gating strategy is provided in the Supplementary Information.

