## [Peer Review File · Nature Microbiology]

Peer Review Information

Journal: Nature Microbiology

Manuscript Title: Single-molecule epitranscriptomic analysis of full-length HIV-1 RNAs reveals functional roles of site-specific m6As

Corresponding author name(s): Sanggu Kim

Reviewer Comments & Decisions:

Decision Letter, initial version:

Message: 27th April 2023

Dear Sanggu,

Thank you for your patience while your manuscript "Single-RNA-level analysis of full-length HIV-1 RNAs reveals functional redundancy of m6As" was under peer-review at Nature Microbiology. It has now been seen by 2 out of 3 referees, whose expertise and comments you will find at the end of this email. In order to not further delay the decision and since the remaining reviewer cannot submit the report soon, we have decided to move forward with 2 review reports in hand. Once we receive the last report, we will forward this to you via e-mail as well. Although the 2 reviewers find your work of some potential interest, they have raised a number of concerns that will need to be addressed before we can consider publication of the work in Nature Microbiology.

In particular, reviewer #1 asks to perform additional control experiments to confirm effects of the identified m6A sites on splicing, as well as to confirm your data by using orthogonal approaches such as SAC-seq, GLORI, or similar. Reviewer #2 suggests testing the effects of m6A modifications on a Gag negative virus construct and to more elaborately discuss your findings compared to previous work and discuss other factors that could be affected by the identified modifications.

Should further experimental data allow you to address these criticisms, we would be happy to look at a revised manuscript.

Please include a data availability statement as a separate section after Methods but before references, under the heading "Data Availability". This section should inform readers about the availability of the data used to support the conclusions of your study. This information includes accession codes to public repositories (data banks for protein, DNA or RNA sequences, microarray, proteomics data etc...), references to source data published alongside the paper, unique identifiers such as URLs to data repository entries, or data set DOIs, and any other statement about data availability. At a minimum, you should include the following statement: "The data that support the findings of this study are available from the corresponding author upon request", mentioning any restrictions on availability. If DOIs are provided, we also strongly encourage including these in the Reference list (authors, title, publisher (repository name), identifier, year). For more guidance on how to write this section please see: <http://www.nature.com/authors/policies/data/data-availability-statements-data-citations.pdf>

- * If you have not done so already we suggest that you begin to revise your manuscript so that it conforms to our Article format instructions at <http://www.nature.com/nmicrobiol/info/final-submission>. Refer also to any guidelines provided in this letter.

When submitting the revised version of your manuscript, please pay close attention to our [href="https://www.nature.com/nature-portfolio/editorial-policies/image-integrity">Digital Image Integrity Guidelines](https://www.nature.com/nature-portfolio/editorial-policies/image-integrity). and to the following points below:

- that unprocessed scans are clearly labelled and match the gels and western blots presented in figures.

2- that control panels for gels and western blots are appropriately described as loading on sample processing controls
- all images in the paper are checked for duplication of panels and for splicing of gel lanes.

Note: This url links to your confidential homepage and associated information about manuscripts you may have submitted or be reviewing for us. If you wish to forward this e-mail to co-authors, please delete this link to your homepage first.

Nature Microbiology is committed to improving transparency in authorship. As part of our efforts in this direction, we are now requesting that all authors identified as 'corresponding author' on published papers create and link their Open Researcher and Contributor Identifier (ORCID) with their account on the Manuscript Tracking System (MTS), prior to acceptance. This applies to primary research papers only. ORCID helps the scientific community achieve unambiguous attribution of all scholarly contributions. You can create and link your ORCID from the home page of the MTS by clicking on 'Modify my Springer Nature account'. For more information please visit please visit www.springernature.com/orcid.

If you wish to submit a suitably revised manuscript we would hope to receive it within 6 months. If you cannot send it within this time, please let us know. We will be happy to consider your revision, even if a similar study has been accepted for publication at Nature Microbiology or published elsewhere (up to a maximum of 6 months).

Yours sincerely,

Reviewer Expertise:

Referee #1: Epitranscriptome

Referee #2: HIV, RNA

Referee #3: HIV

Reviewer Comments:

Reviewer #1 (Remarks to the Author):

In this manuscript, Baek et al carefully characterize the m6A landscape on the HIV

3transcriptome at the single molecule level. They identify predominantly 3 methylated residues, and demonstrate that point-mutating these residues collectively - but not individually - results in missplicing of the HIV transcriptome, which is of critical consequences for HIV propagation. The authors speculate that (1) this points at genetic redundancy, and (2) that such redundancy may serve as a bet-hedging strategy.

There are three key elements we find intriguing in this study: (1) the relative simplicity of the methylation landscape in HIV, contrasting with some previous studies, and (2) the functional consequences on splicing associated with disruption of the methylation site. To date there are very few studies connecting individual m6A sites with a function, and there is an important need for such studies. Moreover, in this case, the association between m6A and splicing is quite intriguing, given that m6A has primarily been linked to stability in mammalian systems, whereas the existence of a causal link to splicing has been controversial, (3) this study sets out to measure the interdependency in m6A across adjacent positions in the same transcript at the single molecule resolution. This is important, and has rarely been done to date due to the technical difficulties associated with this kind of analysis.

Thus, overall I am supportive of the publication of this manuscript.

Nonetheless, I do have two key concerns:

The paucity in methylation sites is a major conclusion of this study. Yet, each of the analytic tools employed by the authors yields ~150 predicted sites. The vast majority of these sites were filtered because they weren't reproducibly detected by all three approaches. While this approach may maximize the specificity of detection, it certainly won't maximize sensitivity. Considering a theoretical scenario wherein there are 150 m6A sites, and one of the three employed approaches perfectly identifies all of them yet the other two fail to identify any of them, the approach employed by the authors would fail to detect any m6A site. Also from inspection of Sup. Fig. 5A, it is clear that when only assessing the intersection between two approaches, additional sites are found, many at m6A motifs. To make a more conclusive statement with respect to the paucity in m6A sites, we feel that an orthogonal approach would be required, such as some of the recently published ones (SAC-seq, GLORI, ...).

The genetic disruption of methylation sites - individually and combined - is a great experiment. Nonetheless, the point-mutation experiments could, in principle, also impact aspects other than the methylated site (e.g. an RNA-binding protein binding site). To more conclusively be able to establish that it is indeed the absence of m6A that underlies the phenotype rather than an unrelated consequence of disrupting the sequence, in our view additional mutants disrupting m6A (e.g. by disrupting the 'C' downstream of the methylated adenosine) would be required. In addition, control mutants, impacting the same general region (but not impacting m6A) would help in establishing the specificity of the phenotype. This question is all the more pertinent as (1) no mechanism is proposed for how presence of m6A impacts splicing, and (2) in mammalian cells in general, m6A is by and large not thought to impact splicing, as shown by different studies that acutely inhibited METTL3 and saw only very limited impact on splicing. Needless to say, working out the mechanism may be beyond the scope of this study, but given the absence of such a mechanism in my view additional evidence would be required here for supporting this

4proposed causal relationship.

The authors argue for a model in which m6A at the different sites act redundantly, suggestive of a bet-hedging strategy. Yet, in figure 4d, panel v, it is evident that also in the single mutants there is increased donor and acceptor usage. So rather than pointing at redundancy, isn't this suggestive of an additive model?

Minor comments

"Single-RNA-level analysis" - this terms appears in the title, abstract and text, and yet it remains unclear to us what the authors are referring to. Are they alluding to the single-molecule measurements? Or to isoform-resolved mapping? This terminology is unclear and should be modified.

I had a difficult time following the section on HIV splicing. To understand it I had to go back and read some previous reviews on the topic. Some additional background in this paper on the topic would be beneficial.

Reviewer #2 (Remarks to the Author):

In this manuscript, Back and Lee et al. use direct RNA sequencing (DRS) to identify several m6A sites in the 3' end of the HIV-1 RNA that alter splicing and the production of viral proteins.

Strengths:

The manuscript is very clearly written, the work is technically outstanding, and the results are of general interest.

There are several innovative advances related to nanopore sequencing, including the use of oligo pools during reverse transcription to promote full length RNA sequencing, the employment of multiple signal analysis tools to identify high confidence sites of modification, and the development of site specific m6a model perform binary classification of individual sequencing reads. These advances are very well described in the manuscript and supplementary notes.

The results are very well controlled, and the authors have used a variety of different methods to confirm the DRS results, including mass-spectrometry, enzymatic demethylation, and synthetic oligos. They use mutagenesis and standard HIV-1 virology assays to identify alterations in infectivity, which they ascribe to splicing defects.

Weaknesses:

There are few weaknesses in this very impressive study. There are some recommendations for improvements, the most important of which are points #1, #7 and #8.

--

5

Specific comments:

1. The title of the manuscript claims functional redundancy of the m6A modifications. The results of the virology experiments in Fig 3. would support this conclusion, but the DRS analysis of splicing defects of the individual mutations hints that each individual m6A site alters the splicing landscape in a different way, and that the effects may be additive. For some reason, the authors chose to perform only a single experiment / analysis, so it is difficult to draw strong conclusions. As these results are very interesting, I would recommend that the authors 'firm up' this data with additional experimental replicates (especially as these data are included in a main figure).
2. How were the single mutations designed? Are they silent? Could they alter splicing indirectly via changes in RNA structure?
3. In Fig 1e legend, the author mention the location of the modifications in relation to the A7 splice acceptors or G-quadruplexes in the U3, but this is not discussed. Can the authors include these facts in their discussion?
4. Fig 2b. The results of the ALKBH5 experiments are most convincing for A8079 and A8975, and not so much for A8989 (A1110 is discarded anyway). Can the authors confirm the removal of A8989 by LC-MS/MS? Also, can LC-MS/MS be used to measure stoichiometry at this position?
5. Fig 4d shows that there is very little change in the use of the canonical D1 splice donor, rather an increase in D1c usage. Is this something that the authors could comment on? Is it important? How might this occur?
6. The colour scheme in Fig 4d(v) could be improved e.g. using a blue-white-red colour scheme to highlight the differences. These data should be plotted log₂ to ensure that the representation is similar for a decrease as for an increase.
7. The manuscript would benefit from an improved (longer) discussion of the results in the context of m6A and viral infections.
 - 7a. In particular, recent studies have suggested that m6A is deposited on RNAs by default, and are excluded from splice-site proximal regions by exon junction complex ([https://www.cell.com/molecular-cell/pdf/S1097-2765\(22\)01205-9.pdf](https://www.cell.com/molecular-cell/pdf/S1097-2765(22)01205-9.pdf), <https://www.nature.com/articles/s41467-022-35643-1> and others). How do these insights compare to the data obtained for HIV-1?
 - 7b. It is interesting that unspliced RNA is less methylated (which is the opposite expectation, based on the recent studies mentioned above). The authors mention that they may be demethylated by Gag-FTO complex prior to virion packaging. This brings to mind several comments. (i) Gag is known to recognise the packaging signal in the 5'UTR - is this not too far from 3' methylations, and how do 3' modifications alter splicing behaviour in the 5' end, especially if we assume splicing is co-transcriptional. (ii) what would happen if the authors were to transfect and analyse methylation patterns in a Gag(-) construct? I would

6encourage the authors to perform this experiment, as they are in a very good position clearly demonstrate a role for Gag in altering gene expression patterns to favour production of the full length RNA i.e. to show that these modifications are globally regulated during replication cycle.

7c. There is tension in the data. Removal of the m6A site by mutation leads to loss of US RNA. But the US RNA does not have m6A? So m6A required to enhance the production of HIV-1 US RNA, but it is subsequently demethylated? Is this interpretation correct, and how does it relate #7b above? These points could be more directly and explicitly discussed in the manuscript.

7d. How would methylations in the 3' alter translation of viral RNAs - can the authors speculate?

8. Raw data (excel tables) should be provided for the each of the figures, so that they can be reproduced independently. This is especially important for the results of the modification pipelines. Nature Micro should insist on this point.

9. I did not appreciate the difference between Fig 4d and Extended Data Fig 7d. In Extended Data Fig 7d the mutations do not affect relative splice acceptor usage, but in Fig 4d(v) describes many changes. Can the authors clarify this point?

Other minor:

- Title: 'Single-RNA-level analysis of full-length HIV-1 RNAs' - would 'Single-molecule analysis' read better?

- Fig 1 could be updated to show the locations all DRACH motifs and spliced-junctions for discussion.

- p1, l34 of summary: 'More densely installed in viral mRNAs than in genomic RNAs' genomic RNA is technically an mRNA. Better to use 'More densely installed in spliced viral mRNA than in genomic RNAs'

- Supplementary note, p1, l39. Correct 1973 to 9173

- l546. Which reference? Supply genbank id or similar

- l563. Which aligner is used?

Author Rebuttal to Initial comments

Editorial comments:

727th April 2023

Thank you for your patience while your manuscript "Single-RNA-level analysis of full-length HIV-1 RNAs reveals functional redundancy of m6As" was under peer-review at Nature Microbiology. It has now been seen by 2 out of 3 referees, whose expertise and comments you will find at the end of this email. In order to not further delay the decision and since the remaining reviewer cannot submit the report soon, we have decided to move forward with 2 review reports in hand. Once we receive the last report, we will forward this to you via e-mail as well. Although the 2 reviewers find your work of some potential interest, they have raised a number of concerns that will need to be addressed before we can consider publication of the work in Nature Microbiology.

In particular, reviewer #1 asks to perform additional control experiments to confirm effects of the identified m6A sites on splicing, as well as to confirm your data by using orthogonal approaches such as SAC-seq, GLORI, or similar.

Reviewer #2 suggests testing the effects of m6A modifications on a Gag negative virus construct and to more elaborately discuss your findings compared to previous work and discuss other factors that could be affected by the identified modifications.

Should further experimental data allow you to address these criticisms, we would be happy to look at a revised manuscript.

12th May 2023

I'm writing you since we have received the remaining review report by reviewer #3. The full report can be found below. In particular, the reviewer mentions that the claims on bet-hedging should be toned down as this is not directly shown. In addition, the reviewer requests to look at viral transcripts in infected CD4+ T cells or a T cell line for more biological relevance.

The remaining comments are clear and straightforward.

I hope you find these comments helpful and we are looking forward to receiving your revised manuscript.

Author Response to Editorial Comments:

We are grateful for the opportunity to revise our manuscript and appreciate all the valuable comments provided by the reviewers. In response to the specific issues raised by the reviewers' comments, we have conducted additional Nanopore direct RNA sequencing (DRS) for single

8mutants, performed a direct comparison with m6A-seq datasets, and demonstrated our major findings using T cells. Furthermore, we have toned down and clarified our descriptions of bet-hedging, as suggested by the reviewers, and included a more comprehensive discussion regarding the potential mechanisms underlying our findings. We believe that we have diligently addressed all the comments raised by the three reviewers and improved our manuscript. The modified sections in the revised manuscript are highlighted in yellow. Our sequence data and codes are publicly available.

Code availability (see Lines 723-725 in the revised manuscript)

The source code for m⁶Arp modeling and analyses is available at <https://github.com/ksanggu/m6Arp>

Data availability (see Lines 718-721 in the revised manuscript)

All data supporting the findings of this study are available within the paper and its Supplementary Information. The Nanopore sequencing data used in this study were deposited into the European Nucleotide Archive (ENA) with an accession number PRJEB61077.

Reviewer #1 (Remarks to the Author):

In this manuscript, Baek et al carefully characterize the m6A landscape on the HIV transcriptome at the single molecule level. They identify predominantly 3 methylated residues, and demonstrate that point-mutating these residues collectively - but not individually - results in missplicing of the HIV transcriptome, which is of critical consequences for HIV propagation. The authors speculate that (1) this points at genetic redundancy, and (2) that such redundancy may serve as a bet-hedging strategy.

There are three key elements we find intriguing in this study: (1) the relative simplicity of the methylation landscape in HIV, contrasting with some previous studies, and (2) the functional consequences on splicing associated with disruption of the methylation site. To date there are very few studies connecting individual m6A sites with a function, and there is an important need for such studies. Moreover, in this case, the association between m6A and splicing is quite intriguing, given that m6A has primarily been linked to stability in mammalian systems, whereas the existence of a causal link to splicing has been controversial, (3) this study sets out to measure the interdependency in m6A across adjacent positions in the same transcript at the single molecule resolution. This is important, and has rarely been done to date due to the technical difficulties associated with this kind of analysis.

Thus, overall I am supportive of the publication of this manuscript.

Reviewer 1 comment #1

Nonetheless, I do have two key concerns: **(#1a)** The paucity in methylation sites is a major conclusion of this study. Yet, each of the analytic tools employed by the authors yields ~150 predicted sites. The vast majority of these sites were filtered because they weren't reproducibly detected by all three approaches. While this approach may maximize the specificity of detection, it certainly won't maximize sensitivity. Considering a theoretical scenario wherein there are 150 m6A sites, and one of the three employed approaches perfectly identifies all of them yet the other two fail to identify any of them, the approach employed by the authors would fail to detect any m6A site. Also from inspection of Sup. Fig. 5A, it is clear that when only assessing the intersection between two approaches, additional sites are found, many at m6A motifs. **(#1b)** To make a more conclusive statement with respect to the paucity in m6A sites, we feel that an orthogonal approach would be required, such as some of the recently published ones (SAC-seq, GLORI, ...).

Author Response to Reviewer 1 comment #1

10The reviewer raised two issues (**#1a** and **#1b**) that would significantly improve the clarity of this manuscript. We address each of these separately as follows:

Author Response to Reviewer 1 comment #1a, regarding the determination of RNA modifications:

The primary focus of this study is the detection and analysis of “site-specific” modifications in HIV-1 RNA. Our focus is not on identifying all the chemical modifications, as previous mass-spectrometry studies have estimated there may be between 80 and 200 various modifications per HIV-1 genome (Courtney et al., 2019; McIntyre et al., 2018). The locations and the functions of site-specific modifications have remained elusive.

To detect site-specific modifications, we employed population-level DRS analyses, as demonstrated in Figure 1. This type of analysis effectively filters out non-site-specific or randomly distributed modification signals, making them undetectable or insignificant (see Fig.1c below). Our data reveal only three major site-specific modifications on HIV-1 RNA, all of which happen to be m6A.

We acknowledge that focusing on modification sites common to the three different analyses may not yield the highest sensitivity, but the primary goal of this population-level analysis is not maximizing sensitivity. Our main goal was to detect and demonstrate the consistency and significance of these three site-specific m6As, which consistently emerged as the most predominant signals in all our analyses (Figure 1 and Supplemental Note 1).

To clarify these issues, the manuscript is modified as follows:

Figure 1:

Figure 1c is modified to include a diagram describing signal peaks of “site-specific” vs. “non-site-specific” modifications in population-level analysis, as follows:

Results 1

PREVIOUS: (Lines 93-100)

“To identify modification sites on a whole-genome scale, we used a two-step signal-refinement process that involved the use of in vitro transcribed (IVT) HIV-1 RNAs as a non-modified RNA control (see Supplementary Note 1 for details). With our optimized conditions, the Tombo

analysis¹ generated highly reproducible per-read modification signals (*p* values) in our repeated experiments (Extended Data Fig. 2). The results from Tombo and other tested software tools, including Eligos2², Nanocompore³, and xPore⁴, consistently revealed a small number of prominent modification signals on the 3' side of the HIV-1 genome (Fig. 1c and Extended Data Fig. 4).”

NEW: (Lines 92-103)

“Previous mass-spectrometry studies have estimated approximately 80-200 modifications of various kinds exist per HIV-1 RNA genome^{4, 14}. The locations and the functions of site-specific modifications, however, remain unclear due to the challenges to identify their precise locations. To identify site-specific modification sites on a whole-genome scale, we used a two-step signal-refinement process that involved the use of *in vitro* transcribed (IVT) HIV-1 RNAs as a non-modified RNA control (see Supplementary Note 1 for details). With our optimized conditions, the Tombo analysis⁴¹ generated highly reproducible per-read modification signals (*p* values) in our repeated experiments (Extended Data Fig. 2). The results from Tombo and other tested software tools, including Eligos2⁴², Nanocompore⁴³, and xPore⁴⁴, consistently revealed a small number of prominent modification signals on the 3' side of the HIV-1 genome (Fig. 1c and Extended Data Fig. 4). These prominent signals likely point to site-specific modifications; the signals from non-site-specific modifications are diluted in these population-based analyses (Fig. 1c).”

Results 2

PREVIOUS: (Lines 101-105)

“To identify ~~the most likely modification sites~~, we compared the modification signals generated by three different tools – Tombo, Eligos2, and Nanocompore¹⁻³, each analyzing different aspects of DRS signals, such as ionic current levels, base-calling error rates, and dwell time⁷. We selected the top 149, 167, and 156 modification signals, respectively, from these tools and cross-compared them (Supplementary Note 1.6).”

NEW: (Lines 104-109)

“To identify ~~the most consistent and significant site-specific modifications~~, we compared the modification signals generated by three different tools – Tombo, Eligos2, and Nanocompore⁴¹⁻⁴³, each analyzing different aspects of DRS signals, such as ionic current levels, base-calling error rates, and dwell time⁴⁵. Given these tools can detect various kinds of chemical modifications, we selected the top 149, 167, and 156 modification signals, respectively, from these tools and cross-compared them (Supplementary Note 1.6).”

Results 3

PREVIOUS: (Lines 122-123)

12~~“Given these modifications are several-fold more frequent or at least as common as m^6As on the HIV-1 genome^{6,8}, the results suggest they may be less site-specific than m^6As .”~~

NEW: (Lines 128-134)

“Given mass spectrometry estimates a large number of chemical modifications on the HIV-1 genome – particularly, Nm, 5mC and ac4C are several-fold more frequent or at least as common as m^6As ^{6, 14}, – it is notable that there are only 3-4 high stoichiometry modification sites, while all other modifications are either undetectable or barely above the detection threshold in these population-level analyses (Fig. 1c and d). These results suggest that Nm, 5mC and ac4C modifications are generally less site-specific than m^6As .”

Author Response to Reviewer 1 comment #1b, regarding the need of orthogonal short-read sequencing:

Short-read sequencing for m6A sites on HIV-1 RNA has been previously reported by Li Wu (Tirumuru et al., 2016), a co-author of this manuscript, as well as by three other research groups (Kennedy et al., 2016; Lu et al., 2020; Pereira-Montecinos et al., 2022). The resolutions in these studies range from approximately 40 to 200 nt.

We demonstrate that the three predominant m6A sites (A8079, A8975 and A8989) selected in this study align well with the low-resolution m6A peaks in these studies (Extended Data Fig.11). We excluded one publication (Lu et al., 2020) as it does not provide the 3' UTR.

To address this, we modified our manuscript as follows:

Extended Data Fig.11Legend of Extended Data Fig.11

New: (Lines 903-916)

“Extended Data Figure 11. Comparison of Nanopore DRS with Short-Read Sequencing for m⁶A Methylation Site Detection. (a) m⁶A-seq analysis of primary CD4⁺ T Cells infected with HIV-1 NL4-3 strain from Tirumuru et al.⁹ RNA fragments containing m⁶A methylation were aligned to the HIV-1 genome. (b) m⁶A sites and m⁶A-reader binding sites mapped by photo-crosslinking-assisted m⁶A sequencing (PA- m⁶A-seq) and photoactivatable ribonucleoside-enhanced crosslinking and immunoprecipitation (PAR-CLIP), respectively, from Kennedy et al.⁶ and Tsai et al.¹⁷ PA- m⁶A-seq Analysis of Virion RNA (Top Panel) and Cellular RNA (Second Panel from the Top) from HIV-1 NL4-3-Infected CD4⁺CEM-SS T Cells, as well as PAR-CLIP analysis of m⁶A reader binding sites, including YTHDF1-3 (middle three panels)⁶ and YTHDC1(bottom two panels)¹⁷ are shown. (c) A magnified View of the four major m⁶A sites predicted by Kennedy et al.⁶ Notably, areas 1 and 3 coincide with our Nanopore DRS m⁶A peaks (purple asterisks in the bottom panel). (d) Potential Modification Sites in Nanopore DRS Data. In this section, we present potential modification sites detected in Nanopore DRS data for HEK293T cells transfected with pNL4-3 (upper panel) and those for WT-infected CD4⁺ T cells (CEM-SS) 96 hour post infection (lower panel; result of single cycle infection, see methods for details).”

Results

PREVIOUS: (Lines 114-115)

“These sites also coincided with the major m⁶A peaks in previous short-read sequencing studies^{8,9}.”

NEW: (Lines 118-120)

“These sites also coincided with the major m⁶A peaks in previous short-read sequencing studies (Extended Data Fig. 11)^{6, 9”}

Reviewer 1 comment 2

The genetic disruption of methylation sites - individually and combined - is a great experiment. (#2a) Nonetheless, the point-mutation experiments could, in principle, also impact aspects other than the methylated site (e.g. an RNA-binding protein binding site). To more conclusively be able to establish that it is indeed the absence of m⁶A that underlies the phenotype rather than an unrelated consequence of disrupting the sequence, in our view additional mutants disrupting m⁶A (e.g. by disrupting the ‘C’ downstream of the methylated adenosine) would be required. In

15addition, control mutants, impacting the same general region (but not impacting m6A) would help in establishing the specificity of the phenotype. (#2b) This question is all the more pertinent as (1) no mechanism is proposed for how presence of m6A impacts splicing, and (2) in mammalian cells in general, m6A is by and large not thought to impact splicing, as shown by different studies that acutely inhibited METTL3 and saw only very limited impact on splicing. Needless to say, working out the mechanism may be beyond the scope of this study, but given the absence of such a mechanism in my view additional evidence would be required here for supporting this proposed causal relationship.

Author Response to Reviewer 1 comment #2a, regarding potential phenotypic impact of nucleotide mutation itself:

Addressing the direct impact of nucleotide changes on the complex 9 Kb HIV-1 RNA structure and RNA-protein interactions is highly challenging. The predicted local structures near the mutation sites (A8079G, A8975C and A8989T) vary among different studies (Lavender et al., 2015; Sükösd et al., 2015; Watts et al., 2009; Wilkinson et al., 2008). While many minimal free energy structure prediction tools (Reuter and Mathews, 2010) showed no notable changes in local structures with the mutations (data not shown), the validity of these structures remains uncertain.

Rather than directly assessing structural changes or RNA-protein interactions, we evaluated phenotypic differences in mutant viruses. Additional DRS experiments were performed to triplicate single mutant DRS data, enabling a statistically evaluation of single-mutant RNA phenotypes (see the updated data in Figures 4, 5, 6, Extended Data Figures 8 and 10). The results indicate that all single mutants exhibited no significant differences compared to WT viruses in most tests. While some mutants did show statistical significance in total HIV-1 RNA levels (Figures 4d-i) and m6A stoichiometry (Figure 6d), but the differences were marginal.

We agree that additional mutagenesis studies, such as altering the "C" in the DRACH motif or introducing benign mutations that do not impact m6A, would be useful control experiments. However, conducting such experiments for HIV-1 RNA is exceedingly challenging. Unlike other smaller or simpler RNAs, HIV-1 RNA encodes highly condensed and multi-layered information within its RNA genome. Predicting the consequences of such control mutations is difficult, as the control mutations themselves could introduce numerous confounding effects.

To address these issues, we added the following statement in the Discussion:

Discussion

New: (Lines 364-373)

"The A8079G mutation is in the overlapping region of rev and env genes and is silent for Rev but induces the substitution of Glutamine with Glycine at position 771 in Env. This change

induces only a moderate reduction in HIV-1 fitness in vitro¹¹¹. A8975C and A8989T are located in the 3' UTR of the HIV-1 RNAs. As expected, we found no significant reduction in viral protein production and infectivity between the single mutants and the wild type. Additionally, single mutants did not exhibit any notable differences in various features of viral RNA, including m⁶A methylation, alternative splicing, 3' polyadenylation, and translation of viral RNAs. Given the significance of RNA-RNA or RNA-protein interactions in regulating these features^{74-77, 112}, it appears that these mutations are well-tolerated and unlikely have had significant impacts on the RNA-RNA or RNA-protein interactions that underlie the maturation and function of viral RNAs."

Author Response to Reviewer 1 comment #2b regarding potential mechanisms by which m6As affect RNA splicing:

While the connection between m6As and RNA splicing can be complex in whole transcriptome-level analyses, it is much clearer in studies focusing on specific genes or viruses. Drawing from our current understanding of m6A's roles in cellular RNAs and HIV-1 RNA biology, we discuss two potential mechanisms: (i) a direct regulation via host regulatory elements and (ii) an indirect regulation via HIV-1 self-regulatory mechanisms. See below for details:

Discussion

NEW: (Lines 346-363)

"The connection between m⁶As and RNA splicing is also intricate and likely context-dependent. The impact of m⁶As on individual genes seems to be heterogeneous in whole-transcriptome studies⁹²⁻⁹⁴. The timing of m⁶A deposition (occurring before or after splicing) is key to understanding the connection, but it appears to be complex for cellular RNAs^{64-66, 80, 83, 93-95}. Notably, gene-specific or virus-specific investigations have established clearer links between m⁶As and alternative splicing, involving m⁶A writers⁹⁶⁻⁹⁹, readers (YTHDC1)^{16, 17, 100}, and erasers^{15, 101, 102}, as well as interactions with splicing regulatory elements^{20, 21, 23, 24, 92, 103}. In HIV-1, the impact of YTHDC1 knockdown on alternative splicing remained partially controversial in two recent studies^{16, 17}. G-quadruplexes (G4s), co-localized with the three major m⁶A sites, might also influence alternative splicing¹⁰⁴⁻¹⁰⁸. m⁶A-G4 co-localization has been reported for other viruses¹⁰⁹, but its role in viral RNA regulation remains unclear.

Besides the direct involvement through cellular splicing regulators, the three m⁶As might indirectly affect alternative splicing considering that the production of partially spliced and unspliced HIV-1 RNAs depends on the production of viral regulatory protein (Tat and Rev)^{51, 79}. m⁶A-mediated translational enhancement of these regulatory proteins, particularly rev, could significantly affect the production of HIV-1 unspliced RNA^{51, 79, 110}. In-depth mechanistic

investigations into the roles of the site-specific m⁶As are warranted to better understand the balance between viral self-regulation and host-regulation.”

Reviewer 1 comment 3

The authors argue for a model in which m6A at the different sites act redundantly, suggestive of a bet-hedging strategy. Yet, in figure 4d, panel v, it is evident that also in the single mutants there is increased donor and acceptor usage. So rather than pointing at redundancy, isn't this suggestive of an additive model?

Author Response to Reviewer 1 comment #3

We agree that alternative splicing in the single mutants exhibit some differences compared to the wild type (WT), although most of these differences were statistically insignificant in the updated Figure 4 and Extended Data Fig. 7b with new triplicate single-mutant DRS data. It is notable that only the triple mutant demonstrates a significant increase in the usage of the D1 donor site, which is required for the initiation of all the splicing events (Fig. 4d-ii, iii & v).

Considering that functional redundancy and additive (or synergistic) effects are not mutually exclusive concepts, we initially evaluated potential additive effects of the three m6As, but it may not have been clearly described (Fig. 6 and Extended Data Fig. 10; see the modified results section below). Briefly, we found no clear evidence of (i) compensatory deposition of new m6As at other DRACH sites (Extended Data Fig. 10a) nor (ii) selection or enrichment of RNAs that harbor multiple m6As (Fig. 6b,c) in respond to the loss of a dominant m6A. Furthermore, our read-level splicing analysis revealed that (iii) any HIV-1 RNA molecules with at least one of the major m6As, regardless of the position, can go through similar splicing (Fig.6) and that (iv) the functional properties of these major m6As do not change in single mutants (Extended Data Fig. 10b-d). Lastly, we also found that (v), despite the substantial loss of total m6As, single mutants exhibit only a marginal reduction in the fraction of RNAs that have at least one of the three m6As (Fig. 6d).

Interpreting these findings in an evolutionary context, our data suggest that the functional redundancy of m6As best aligns with the concept of bet-hedging among the three core modes of evolutionary responses (adaptive tracking, phenotypic plasticity, and bet-hedging)(*Simons, 2011*). We propose that HIV-1 maintains multiple redundant m6As to reduce the risk of losing them during viral replication.

We also acknowledge the need for additional confirmatory experiments and agree to reduce “the focus on the word bet-hedging” as suggested in **Reviewer #3 comment 1** below.

We modified the manuscript to more explicitly and clearly address these issues, as follows:

18Results

PREVIOUS: (Lines 251-270)

“Given the functional redundancy of these m⁶As, we then asked why HIV-1 maintains excessive m⁶As on its RNAs. First, having multiple m⁶As may be necessary to sustain normal levels of viral replication. In such a scenario, our single mutations would show either a selection for RNAs with multiple m⁶As or an increase in m⁶A stoichiometry at intact DRACH sites to compensate for the loss of a dominant m⁶A. However, we found indistinguishable or only moderate differences in m⁶A stoichiometry (Fig. 6b,c) and m⁶A landscape (Extended Data Fig. 10a) between the single mutants and the WT HIV-1. Like WT RNAs, all RNA subspecies of single mutants, regardless of m⁶A status, showed no significant bias toward the utilization of certain splicing donors or acceptors (Extended Data Fig. 10b-d).

Given that there is no clear evidence of adaptive selection or changes in m⁶A functions, a more favorable scenario may be that HIV-1 spreads the risk of losing m⁶A functions⁶⁴. HIV-1 may tolerate these multiple redundant m⁶As to minimize the risk of unpredictable random mutagenesis (10⁻⁵ to 10⁻³ mutations/bp/cycle for HIV-1⁶²) that may knockout a major m⁶As, analogous to bet-hedging in evolutionary biology reducing risks against unpredictably fluctuating environments⁶⁴. Interestingly, all the single mutants maintained at least one of the three m⁶As in most of their CS (91.5-95.8%) and PS (89.1%-95.0%) RNAs, levels comparable to the WT HIV-1 (Fig. 6d). Despite causing substantial loss of m⁶A at the population level, single mutations had only a marginal effect on overall viral fitness. The loss of all three, however, eroded the potential of RNA communities to sustain their control over splicing and translation, and adversely affected the various stages of the HIV-1 life cycle (Fig. 3).”

NEW: (Lines 266-292)

“Given the functional redundancy of these m⁶As, we then asked why HIV-1 maintains excessive m⁶As on its RNAs. To explore potential additive or synergistic effects of the three major m⁶As, we hypothesized that having multiple of these m⁶As on the majority of individual RNA molecules (‘subspecies A-D’ in Fig. 5c) is vital for HIV-1 to maintain normal levels of viral replication. Given that all single mutants exhibited no significant reduction in most of their replication stages (Fig. 3), we investigated whether (i) the single mutants deposit new m⁶As at other DRACH sites in response to the loss of a dominant m⁶A and/or (ii) selectively enrich multiple-m⁶A-containing RNAs in their RNA pool. However, we found indistinguishable or only moderate differences in the m⁶A stoichiometry at each of the three major m⁶A sites (Fig. 6b, c) and m⁶A landscape (Extended Data Fig. 10a) between the single mutants and the WT HIV-1. These results suggest no significant additive effects of the three m⁶As on viral replication, except for a moderate increase in alternative splicing in the single mutants.

Our read-level splicing analysis also showed that all RNA subspecies of single mutants exhibited no significant bias toward the utilization of specific splicing donors or acceptors (Extended Data Fig. 10b-d), similar to what was observed in WT (Fig. 6a-v), suggesting no functional changes of the three m⁶As in single mutants.

In the context of RNA-population-level evolutionary responses, our data suggest that the functional redundancy of m⁶As on the HIV-1 RNA best aligns with the bet-hedging mode among the three core modes of evolutionary response, including adaptive tracking, plasticity in phenotype (or function), and bet-hedging⁶⁰. HIV-1 may tolerate these multiple redundant m⁶As to minimize the risk of unpredictable random mutagenesis (10⁻⁵ to 10⁻³ mutations/bp/cycle for HIV-1⁶¹) that may knockout a major m⁶As. Interestingly, all the single mutants maintained at least one of the three m⁶As in most of their CS (87.7-94.7%) and PS (81.9-93.9%) RNAs, levels comparable to the WT HIV-1 (Fig. 6d). Despite causing substantial loss of m⁶A at the population level, single mutations had only a marginal effect on overall viral fitness. The loss of all three, however, eroded the potential of RNA communities to sustain their control over splicing and translation, and adversely affected the various stages of the HIV-1 life cycle (Fig. 3)”

Reviewer 1 Minor comment 1

“Single-RNA-level analysis” - this terms appears in the title, abstract and text, and yet it remains unclear to us what the authors are referring to. Are they alluding to the single-molecule measurements? Or to isoform-resolved mapping? This terminology is unclear and should be modified.

Author Response to Reviewer 1 Minor comments 1

“Single-RNA-level analysis” was used to indicate “the single-molecule measurements”, which enables “isoform-resolved mapping” and other read-level analyses in this study. To clarify these, we modified our manuscript as follows:

Title

PREVIOUS: (Line 1)

“Single-RNA-level analysis of full-length HIV-1 RNAs reveals functional redundancy of m⁶As ”

NEW: (Line 1)

“Single-molecule analysis of full-length HIV-1 RNAs reveals functional redundancy of m⁶As ”

Abstract

PREVIOUS: (Lines 38-41)

“Our single-RNA-level study demonstrates that HIV-1 tolerates functionally redundant m⁶As to provide stability and resilience to viral replication while minimizing the risk of losing m⁶As in unpredictably changing conditions – potentially a novel RNA-level evolutionary strategy for HIV-1.”

NEW: (Lines 38-41)

*“Our **single-molecule-level** study demonstrates that HIV-1 tolerates functionally redundant m⁶As to provide stability and resilience to viral replication while minimizing **the risk of losing m⁶As due to random mutations** – potentially a novel RNA-level evolutionary strategy for HIV-1”*

Extended Data Fig.10

PREVIOUS: (Line 761)

“Extended Data Fig. 10: Single-RNA-level analysis reveals the functional redundancy of the m⁶As.”

NEW: (Line 888)

*“Extended Data Fig. 10: **Single-molecule-level** analysis reveals the functional redundancy of the m⁶As.”*

Reviewer 1 Minor comment 2

I had a difficult time following the section on HIV splicing. To understand it I had to go back and read some previous reviews on the topic. Some additional background in this paper on the topic would be beneficial.

Author Response to Reviewer 1 Minor comments 2

We used Fig.4b to briefly introduce to HIV-1 alternative splicing, and modified the Results section as follows.

Results

NEW: (Lines 182-185)

“All HIV-1 RNAs are produced as a full-length initially and remain US (genomic RNA for virion packaging or mRNA for gag/gag-pol) or spliced into CS (major mRNA for nef, rev, or tat) and PS (major mRNA for vif, vpr, or env/vpu) (Fig.4b). RNA modifications have been suggested to affect HIV splicing^{4, 16, 17.”}

Reviewer 2 comment 1

1. The title of the manuscript claims functional redundancy of the m6A modifications. The results of the virology experiments in Fig 3. would support this conclusion, but the DRS analysis of splicing defects of the individual mutations hints that each individual m6A site alters the splicing landscape in a different way, and that the effects may be additive. For some reason, the authors chose to perform only a single experiment / analysis, so it is difficult to draw strong conclusions. As these results are very interesting, I would recommend that the authors 'firm up' this data with additional experimental replicates (especially as these data are included in a main figure).

Author Response to Reviewer 2 comment 1

We performed additional DRS on single-mutant producing cells and updated figures with triplicated DRS data and appropriate statistics (Figures 4, 5, and 6; Extended Data Figures 8 and 10). Our response to this comment (regarding the potential additive effect) is essentially identical to “**Author Response to Reviewer 1 comment #3**” above.

Following figures were updated with triplicated single mutant DRS data:

Figure 4a: (updated with triplicated single mutant data)

Figure 4d: (updated with triplicated single mutant data)

22

Fig. 4d legend

PREVIOUS: (Lines 362-363)

“($p=0.0106$; two-tailed T test; WT: $n=4$ distinct samples; Triple: $n=3$ distinct samples; error bar: standard deviation)”

NEW: (Lines 464-465)

“($p=0.0106$; two-tailed T test; error bar: standard deviation; WT: $n=4$; Triple, **A8079G, A8975C, and A8989T**: $n=3$; all independently prepared samples)”

Figure 4e: (updated with triplicated single mutant data)

Figure 6b-d. (updated with triplicated single mutant data)

Figure 6 legend

NEW: (Lines 510-511)

“(two-tailed *T* test; error bar: standard deviation; WT: *n*=4; A8079G, A8975C, and A8989T: *n*=3; all independently prepared samples)”

Extended Data Figure 7d. (updated with triplicated single mutant data; new figures i-iii are added)

d

The legend of Extended Data Fig. 7d:

PREVIOUS: (Lines 710-711)

“(d) Relative usage rates of splicing acceptors (%; y-axis) are shown for WT (*n*=4 distinct samples), triple mutants (*n*=3 distinct samples) and single mutants (A8079G, A8975G, A8989T, and A8110G).”

New (Lines 833-839)

“(d) splicing donor and acceptor site usage. (i-ii) bar-graphs showing the relative usage rates per HIV-1 RNA (US+PS+CS combined; y-axis) for Splicing Donor usage (i) and Splicing Acceptor usage (ii). (iii) First donor sites. Nearly all (93.5%–94.6%) spliced RNA uses D1 donor; 5.3% to 6.3% use D1c; and less than 0.5% use other donors (D2, D3 or D4) for the first splicing. (iv) The acceptor usage rates following the D1 donor usage (% of D1, y-axis) during the first

splicing event. WT (n=4 distinct samples), triple mutants (n= 3 distinct samples) and single mutants (A8079G, A8975G, A8989T, and A8110G; n=3 samples) are shown.”

Extended Data Figure 10. (updated with triplicated single mutant data)26

ing,
:
s,

unless

indicated otherwise in a credit line to the material. If material is not included in the article's Creative Commons license and your intended use is not permitted by statutory regulation or exceeds the permitted use, you will need to obtain permission directly from the copyright holder. To view a copy of this license, visit <http://creativecommons.org/licenses/by/4.0/>.

Results : the following section is modified to more explicitly and clearly address these issues.

PREVIOUS: (Lines 251-270)

~~“Given the functional redundancy of these m^6 As, we then asked why HIV-1 maintains excessive m^6 As on its RNAs. First, having multiple m^6 As may be necessary to sustain normal levels of viral replication. In such a scenario, our single mutations would show either a selection for RNAs with multiple m^6 As or an increase in m^6 A stoichiometry at intact DRACH sites to compensate for the loss of a dominant m^6 A. However, we found indistinguishable or only moderate differences in m^6 A stoichiometry (Fig. 6b,c) and m^6 A landscape (Extended Data Fig. 10a) between the single mutants and the WT HIV-1. Like WT RNAs, all RNA subspecies of single mutants, regardless of m^6 A status, showed no significant bias toward the utilization of certain splicing donors or acceptors (Extended Data Fig. 10b-d).~~

~~Given that there is no clear evidence of adaptive selection or changes in m^6 A functions, a more favorable scenario may be that HIV-1 spreads the risk of losing m^6 A functions⁶⁴. HIV-1 may tolerate these multiple redundant m^6 As to minimize the risk of unpredictable random mutagenesis (10^{-5} to 10^{-3} mutations/bp/cycle for HIV-1⁶²) that may knockout a major m^6 As, analogous to bet-hedging in evolutionary biology reducing risks against unpredictably fluctuating environments⁶⁴. Interestingly, all the single mutants maintained at least one of the three m^6 As in most of their CS (91.5-95.8%) and PS (89.1%-95.0%) RNAs, levels comparable to the WT HIV-1 (Fig. 6d). Despite causing substantial loss of m^6 A at the population level, single mutations had only a marginal effect on overall viral fitness. The loss of all three, however, eroded the potential of RNA communities to sustain their control over splicing and translation, and adversely affected the various stages of the HIV-1 life cycle (Fig. 3).”~~

NEW: (Lines 266-292)

~~“Given the functional redundancy of these m^6 As, we then asked why HIV-1 maintains excessive m^6 As on its RNAs. To explore potential additive or synergistic effects of the three major m^6 As, we hypothesized that having multiple of these m^6 As on the majority of individual RNA molecules ('subspecies A-D' in Fig. 5c) is vital for HIV-1 to maintain normal levels of viral replication. Given that all single mutants exhibited no significant reduction in most of their replication stages (Fig. 3), we investigated whether (i) the single mutants deposit new m^6 As at other DRACH sites in response to the loss of a dominant m^6 A and/or (ii) selectively enrich multiple- m^6 A-containing RNAs in their RNA pool. However, we found indistinguishable or only moderate differences in the m^6 A stoichiometry at each of the three major m^6 A sites (Fig. 6b, c) and m^6 A landscape (Extended Data Fig. 10a) between the single mutants and the WT HIV-1. These results suggest no significant additive effects of the three m^6 As on viral replication, except for a moderate increase in alternative splicing in the single mutants.~~

Our read-level splicing analysis also showed that all RNA subspecies of single mutants exhibited no significant bias toward the utilization of specific splicing donors or acceptors (Extended Data Fig. 10b-d), similar to what was observed in WT (Fig. 6a-v), suggesting no functional changes of the three m⁶As in single mutants.

In the context of RNA-population-level evolutionary responses, our data suggest that the functional redundancy of m⁶As on the HIV-1 RNA best aligns with the bet-hedging mode among the three core modes of evolutionary response, including adaptive tracking, plasticity in phenotype (or function), and bet-hedging⁶⁰. HIV-1 may tolerate these multiple redundant m⁶As to minimize the risk of unpredictable random mutagenesis (10⁻⁵ to 10⁻³ mutations/bp/cycle for HIV-1⁶¹) that may knockout a major m⁶As. Interestingly, all the single mutants maintained at least one of the three m⁶As in most of their CS (87.7-94.7%) and PS (81.9-93.9%) RNAs, levels comparable to the WT HIV-1 (Fig. 6d). Despite causing substantial loss of m⁶A at the population level, single mutations had only a marginal effect on overall viral fitness. The loss of all three, however, eroded the potential of RNA communities to sustain their control over splicing and translation, and adversely affected the various stages of the HIV-1 life cycle (Fig. 3)”

Reviewer 2 comment 2

2. How were the single mutations designed? Are they silent? Could they alter splicing indirectly via changes in RNA structure?

Author Response to Reviewer 2 comment 2

We appreciate the comment. Our response to this comment is essentially identical to “**Author Response to Reviewer 1 comment #2a**” above.

To address these issues, we modified our manuscript as follows:

Discussion

New: (Lines 364-373)

“The A8079G mutation is in the overlapping region of rev and env genes and is silent for Rev but induces the substitution of Glutamine with Glycine at position 771 in Env. This change induces only a moderate reduction in HIV-1 fitness in vitro¹¹¹. A8975C and A8989T are located in the 3' UTR of the HIV-1 RNAs. As expected, we found no significant reduction in viral protein production and infectivity between the single mutants and the wild type. Additionally, single mutants did not exhibit any notable differences in various features of viral RNA, including m⁶A methylation, alternative splicing, 3' polyadenylation, and translation of viral RNAs. Given the significance of RNA-RNA or RNA-protein interactions in regulating these features^{74-77, 112}, it

appears that these mutations are well-tolerated and unlikely have had significant impacts on the RNA-RNA or RNA-protein interactions that underlie the maturation and function of viral RNAs.”

Reviewer 2 comment 3

3. In Fig 1e legend, the author mention the location of the modifications in relation to the A7 splice acceptors or G-quadruplexes in the U3, but this is not discussed. Can the authors include these facts in their discussion?

Author Response to Reviewer 2 comment 3

In the discussion section, the following section is added to expand on the potential roles of common RNA features near the three m⁶As in relation to the site-specific deposition and functions of these m⁶As.

Discussion

NEW: (Lines 312-315)

“HIV-1 also exhibits predominant m⁶A deposition downstream of the last splicing site (A7) and adjacent to the stop codons of tat and nef genes, mirroring the typical m⁶A patterns on cellular mRNAs⁶⁴⁻⁶⁹, but interestingly HIV-1 deposits m⁶As almost exclusively at A8079, A8975, and A8989 sites for all its RNA isotypes.”

NEW: (Lines 354-356)

“G-quadruplexes (G4s), co-localized with the three major m⁶A sites, might also influence alternative splicing¹⁰⁴⁻¹⁰⁸. m⁶A-G4 co-localization has been reported for other viruses¹⁰⁹, but its role in viral RNA regulation remains unclear.”

Reviewer 2 comment 4

4. Fig 2b. The results of the ALKBH5 experiments are most convincing for A8079 and A8975, and not so much for A8989 (A1110 is discarded anyway). Can the authors confirm the removal of A8989 by LC-MS/MS? Also, can LC-MS/MS be used to measure stoichiometry at this position?

Author Response to Reviewer 2 comment 4

The presence of m⁶A at A8989 was confirmed by oligonucleotide LC-MS/MS (Fig. 2c-iii-), as well as site-directed mutagenesis (Fig. 2b-ii-) and DRS of synthetic oligos with m⁶As at A8989 (Fig. 5a-ii-). However, confirming the removal of A8989 or quantitative analysis of A8989 m⁶As using oligonucleotide LC-MS/MS is far too difficult to handle in this manuscript. This technique has proven effective for small RNAs, such as tRNAs and rRNAs (D'Ascenzo et al., 2022), but

29analyzing larger, full-length RNAs like the 9 Kb HIV-1 RNA has posed challenges – due largely to the ambiguity of RNase T1-digested fragments (RNase T1 cuts every G) from large RNA molecules and also the limited detection sensitivity for a small amount of input RNA. We developed a novel method involving the enrichment and purification of a 50-base long target fragment of HIV-1 RNA containing both A8975 and A8989 sites, but this process requires a substantial amount (10 µg) of virion RNA (9 Kb) to enrich a sufficient amount of target RNA for LC-MS/MS. The current sensitivity achieved using these RNA was sufficient for detecting the presence of m6As at these two sites. However, quantifying m6A modifications at A8989 would require a significantly larger amount of viral RNA and extensive assay optimization. While this may be possible, confirming the removal of m6A at A8989 or measuring the stoichiometry at this position using LC-MS/MS is beyond the primary scope of this study. ALKBH5 has shown differential enzymatic activities depending on the local structure of the substrate (Xu et al., 2014).

We added the following statement in the results section to provide additional confirmation data:

Results 1

NEW: (Lines 139-140)

“ALKBH5 has shown differential enzymatic activities depending on the local structure of the substrate⁴⁷.”

Results 2

PREVIOUS: (Lines 135-137)

~~*“As the significant modification signals on A8079 site were consistent in all our tests, we selected A8079, A8975, and A8989 to investigate their site-specific roles. m⁶A at A8110, however, remains unclear.”*~~

NEW: (Lines 145-153)

“As significant m⁶A modification signals were consistently observed at the A8079 site in all our tests, we selected A8079, A8975, and A8989 for further investigation of their site-specific roles. DRS of synthetic oligonucleotides with m⁶A at A8079, A8975, or A8989 further supports the presence of m⁶As at these three sites (see Fig. 5a-ii and Extended Data Fig. 9). Despite multiple attempts, we could not confirm m⁶A methylation at A8110 by LC-MS/MS due to the insufficient enrichment of RNA fragments containing A8110. Read-level quantification assays, including m6Anet and Nanom6A (Supplementary Note Fig. SN6), suggest that m⁶A at A8110 is a relatively low-stoichiometry methylation, if it is present at all.”

Reviewer 2 comment 5

305. Fig 4d shows that there is very little change in the use of the canonical D1 splice donor, rather an increase in D1c usage. Is this something that the authors could comment on? Is it important? How might this occur?

Author Response to Reviewer 2 comment 5

The usage rate of D1 (approximately 93.5%-94.6%) is >16-fold higher than that of D1c (approximately 5.3% to 6.3%). The differences in D1c usage rates were more sensitively displayed in Fig. 4d (values relative to WT). To more clearly present the donor and acceptor usages, we modified Fig. 4d to a log₂-scale heatmap (as suggested in **Reviewer 2 comment 6** below) and provide new bar graphs in Extended Data Fig.7d (i - iii) as shown below:

Extended Data Figure 7d.

Extended Data Fig. 7d legend:

PREVIOUS: (Lines 710-711)

“(d) Relative usage rates of splicing acceptors (%; y-axis) are shown for WT (n=4 distinct samples), triple mutants (n= 3 distinct samples) and single mutants (A8079G, A8975G, A8989T, and A8110G).”

New (Lines 833-839)

“(d) splicing donor and acceptor site usage. (i-ii) bar-graphs showing the relative usage rates per HIV-1 RNA (US+PS+CS combined; y-axis) for Splicing Donor usage (i) and Splicing Acceptor usage (ii). (iii) First donor sites. Nearly all (93.5%-94.6%) spliced RNA uses D1 donor; 5.3% to 6.3% use D1c; and less than 0.5% use other donors (D2, D3 or D4) for the first splicing.

31(iv) The acceptor usage rates following the D1 donor usage (% of D1, y-axis) during the first splicing event. WT (n=4 distinct samples), triple mutants (n= 3 distinct samples) and single mutants (A8079G, A8975G, A8989T; n=3 samples) are shown.”

Figure 4d(v).

Before:

Now:

Figure 4 legend:

PREVIOUS: (Lines 365-366)

“Donor and acceptor usage rates are generally higher in triple-mutant-producing cells than those in WT-producing cells (v).”

Now : (Lines 467-469)

“Donor and acceptor usage rates (shown in log₂ scale heatmap; * p<0.05, ** p<0.01, student t-test) are generally higher in triple-mutant-producing cells than those in WT-producing cells (v).”

Reviewer 2 comment 6

6. The colour scheme in Fig 4d(v) could be improved e.g. using a blue-white-red colour scheme to highlight the differences. These data should be plotted log₂ to ensure that the representation is similar for a decrease as for an increase.

Author Response to Reviewer 2 comment 6

The color scheme is changed as suggested (see new Figure 4 in “Author Response to Reviewer 2 comment 5” above).

Reviewer 2 comment 7

327. The manuscript would benefit from an improved (longer) discussion of the results in the context of m⁶A and viral infections.

Author Response to Reviewer 2 comment 7.

We have expanded our discussion, incorporating additional details and insights into the complex biology of RNA modifications and HIV-1 replication. See below our responses to specific comments 7a-d.

Reviewer 2 comment 7a.

7a. In particular, recent studies have suggested that m⁶A is deposited on RNAs by default, and are excluded from splice-site proximal regions by exon junction complex ([https://www.cell.com/molecular-cell/pdf/S1097-2765\(22\)01205-9.pdf](https://www.cell.com/molecular-cell/pdf/S1097-2765(22)01205-9.pdf), <https://www.nature.com/articles/s41467-022-35643-1> and others). How do these insights compare to the data obtained for HIV-1?

Author Response to Reviewer 2 comment 7a.

Our HIV-1-specific mapping data are largely consistent with recent reports demonstrating EJC-mediated “targeted-suppression” mechanisms controlling of m⁶A deposition. However, our HIV-1 data, some of which are distinct from cellular RNA patterns, also suggest the existence of additional, potentially HIV-1-specific mechanisms. See below for details:

Discussion

NEW: (Lines 305-320)

“HIV-1 displays a unique pattern of m⁶A deposition, focusing primarily on the three DRACH sites (A8079, A8975, and A8989) out of a total 242 DRACH sites on its RNA. This observation is intriguing when compared to general m⁶A deposition patterns on cellular RNAs, which are largely influenced by steric hindrance (or “targeted suppression”) of RNA-binding-protein (RBP) complexes, such as the exon junction complexes (EJC)⁶⁴⁻⁶⁹. In cellular mRNAs, m⁶A sites are thereby restricted to specific transcript regions, away from exon junctions, favoring 3' untranslated regions (UTRs)⁶⁴⁻⁶⁹. Any DRACH site may undergo methylation in the absence of RBP binding. HIV-1 also exhibits predominant m⁶A deposition downstream of the last splicing site (A7) and adjacent to the stop codons of tat and nef genes, mirroring the typical m⁶A patterns on cellular mRNAs⁶⁴⁻⁶⁹, but interestingly HIV-1 deposits m⁶As almost exclusively at A8079, A8975, and A8989 sites for all its RNA isotypes. Even after losing the major m⁶As due to mutation(s), HIV-1 showed no notable changes in m⁶A site selection nor any increase in m⁶A at any of the remaining DRACH sites. Moreover, unlike cellular unspliced RNAs that do not show m⁶A enrichment in 3' UTRs (due likely to the lack of EJC binding)⁷⁰⁻⁷³, HIV-1 unspliced RNA exhibits no differences in the site-specificity of m⁶As compared to spliced HIV-1 mRNAs. This

unique m⁶A pattern on HIV-1 RNA implies additional and site-specific mechanisms regulating m⁶A deposition.”

Reviewer 2 comment 7b.

7b. It is interesting that (#7b-1) unspliced RNA is less methylated (which is the opposite expectation, based on the recent studies mentioned above). (#7b-2) The authors mention that they may be demethylated by Gag-FTO complex prior to virion packaging. This brings to mind several comments. (i) (#7b-3) Gag is known to recognise the packaging signal in the 5'UTR - is this not too far from 3' methylations, and (#7b-4) how do 3' modifications alter splicing behaviour in the 5' end, especially if we assume splicing is co-transcriptional. (ii) (#7b-5) what would happen if the authors were to transfect and analyse methylation patterns in a Gag(-) construct? I would encourage the authors to perform this experiment, as they are in a very good position clearly demonstrate a role for Gag in altering gene expression patterns to favour production of the full length RNA i.e. to show that these modifications are globally regulated during replication cycle.

Author Response to Reviewer 2 comment #7b-1, regarding “*unspliced RNA is less methylated (which is the opposite expectation, based on the recent studies mentioned above)*”:

We acknowledge that there are several unique features in the m⁶A patterns on HIV-1 RNA that differ from typical cellular RNAs. For example, (i) unspliced HIV-1 RNAs exhibit 3' enrichment of m⁶As, whereas cellular unspliced RNAs (e.g., nascent RNA or long-non-coding RNAs like LINE-1) do not display m⁶A enrichment in the 3' UTR due to the absence of EJC binding(He et al., 2023; Murakami and Jaffrey, 2022; Uzonyi et al., 2023; Yang et al., 2022), and (ii) unspliced HIV-1 genomic RNA is less methylated than spliced HIV-1 mRNAs. This second observation is also interesting, considering previous reports showing an increased m⁶A deposition in the absence of steric hindrance of EJC(Uzonyi et al., 2023).

We propose that these unique HIV-1-specific patterns indicate the necessity of additional, HIV-1-specific mechanisms beyond EJC-mediated steric hindrance to explain the site-specific m⁶A deposition in HIV-1. Our response to this comment is identical to “**Author Response to Reviewer 2 comment 7a**” above.

Author Response to Reviewer 2 comment #7b-2, regarding “*The authors mention that they may be demethylated by Gag-FTO complex prior to virion packaging.*”:

We propose two potential scenarios to explain the differential m⁶A patterns between HIV-1 unspliced (genomic) RNAs and spliced mRNAs: (a) as a consequence of early fate decision (genomic RNAs and mRNAs) and (b) demethylation by Gag-FTO.

34We provide a new paragraph to address to this comment:

Discussion

NEW: Lines 321-332 (continued from the previous paragraph)

“In particular, the differential m⁶A depositions between HIV-1 virion RNAs and spliced mRNAs may reflect differential interactions between cellular/viral RBPs and different HIV-1 RNA isoforms⁷⁴⁻⁷⁷. Recent studies suggest that the fate of HIV-1 RNA, whether it becomes mRNA or genomic RNA, might be determined early during transcription by factors such as transcription start site variations (e.g., guanosine number at the 5' terminal) and structural differences in the 5' UTR^{58, 59, 78}. These factors may lead HIV precursor RNAs to enter mutually exclusive RNA-fate pathways^{74, 79}. Given these findings, it's possible that differential m⁶A deposition on HIV-1 mRNA and genomic RNA is influenced by distinct transcriptional context and speed⁸⁰⁻⁸³, as well as distinct RNA-RBP interactions⁶⁴⁻⁶⁷ in these different fate pathways. Another potential scenario involves a selective demethylation of m⁶A modifications on genomic RNA by a Gag-FTO complex, enhancing virion packaging¹⁵. This may occur independently of the early fate decision discussed above. However, further investigation is required to fully understand the exact mechanisms.”

Author Response to Reviewer 2 comment #7b-3, regarding “how would Gag facilitate the removal of m6As at the 3'end”:

Interestingly, Gag has displayed a substantial binding preference toward the 3' side of viral RNA, besides its well-known binding to the 5' UTR, in previous CLIP studies for genome-wide Gag-binding sites (Kessl et al., 2016; Kutluay et al., 2014). It is, however, premature to directly correlate this Gag-3'RNA binding with m6A demethylation. Gag-FTO-mediated demethylation is one of many possible scenarios (as discussed in **Author Response to Reviewer 2 comment #7b-2** above).

Author Response to Reviewer 2 comment #7b-4, regarding “How do 3' end m6As alter splicing behavior on the 5' side?”:

Our response is essentially identical to “**Author Response Reviewer 1 comment #2b**” above, addressing potential mechanisms how the 3' site-specific m6As control HIV-1 splicing.

To effectively address these #7b-4, we moved and expanded following section to the Discussion section as follows:

PREVIOUS: Lines 235-240 in the Results section

~~“Given the genetic and structural differences between HIV-1 mRNAs and genomic RNAs⁵⁷⁻⁵⁸, the differential levels of the three m⁶As between these two types of RNAs may be controlled co-transcriptionally^{59, 60}. A recent report also suggests that m⁶As on the genomic RNA are selectively demethylated by a Gag-FTO complex prior to virion packaging¹⁵. While the exact modes of action remaining unclear, the fine-tuning of these m⁶As by HIV-1 emphasizes the critical importance of these m⁶As in viral replication.”~~

NEW: (Lines 346-363)

“The connection between m⁶As and RNA splicing is also intricate and likely context-dependent. The impact of m⁶As on individual genes seems to be heterogeneous in whole-transcriptome studies⁹²⁻⁹⁴. The timing of m⁶A deposition (occurring before or after splicing) is key to understanding the connection, but it appears to be complex for cellular RNAs^{64-66, 80, 83, 93-95}. Notably, gene-specific or virus-specific investigations have established clearer links between m⁶As and alternative splicing, involving m⁶A writers⁹⁶⁻⁹⁹, readers (YTHDC1)^{16, 17, 100}, and erasers^{15, 101, 102}, as well as interactions with splicing regulatory elements^{20, 21, 23, 24, 92, 103}. In HIV-1, the impact of YTHDC1 knockdown on alternative splicing remained partially controversial in two recent studies^{16, 17}. G-quadruplexes (G4s), co-localized with the three major m⁶A sites, might also influence alternative splicing¹⁰⁴⁻¹⁰⁸. m⁶A-G4 co-localization has been reported for other viruses¹⁰⁹, but its role in viral RNA regulation remains unclear.

Besides the direct involvement through cellular splicing regulators, the three m⁶As might indirectly affect alternative splicing considering that the production of partially spliced and unspliced HIV-1 RNAs depends on the production of viral regulatory protein (Tat and Rev)^{51, 79}. m⁶A-mediated translational enhancement of these regulatory proteins, particularly rev, could significantly affect the production of HIV-1 unspliced RNA^{51, 79, 110}. In-depth mechanistic investigations into the roles of the site-specific m⁶As are warranted to better understand the balance between viral self-regulation and host-regulation.”

Author Response to Reviewer 2 comment #7b-5, regarding “methylation patterns in a Gag(-) construct”.

The Soto Rifo group, who demonstrated Gag-FTO-mediated demethylation, has shown that Gag(-) viruses significantly reduce m⁶A levels on their genomic RNAs. However, it's important to note that the influence of Gag is one of many possible scenarios, and investigating into each of these mechanisms would require an intensive study on viral and cellular biology that extend far beyond the scope of this manuscript. Independent mechanism studies should be followed up to thoroughly investigate these events. We address this issue in Discussion as one of possible scenarios, as follows:

36Discussion

NEW: Lines 329-332

“Another potential scenario involves a selective demethylation of m⁶A modifications on genomic RNA by a Gag-FTO complex, enhancing virion packaging¹⁵. This may occur independently of the early fate decision discussed above. However, further investigation is required to fully understand the exact mechanisms.”

Reviewer 2 comment 7c.

7c. There is tension in the data. Removal of the m⁶A site by mutation leads to loss of US RNA. But the US RNA does not have m⁶A? So m⁶A required to enhance the production of HIV-1 US RNA, but it is subsequently demethylated? Is this interpretation correct, and how does it relate #7b above? These points could be more directly and explicitly discussed in the manuscript.

Author Response to Reviewer 2 comment 7c.

We agree that these observations may appear inconsistent. To clarify this issue drawing upon our most recent understanding of m⁶A and HIV-1 Biology, we have separated the discussion into following two issues: (a) differential m⁶A deposition between unspliced and spliced HIV-1 RNA and (b) the impact of m⁶As on the production of unspliced RNAs.

For (a), we propose two potential mechanisms: -i- differential methylation via the early fate decision; -ii- genomic-RNA-specific m⁶A demethylation (e.g. by Gag-FTO complex) (see **Author Response to Reviewer 2 comment #7b-1** and **#7b-2** above).

For (b), we provide current insights and potential scenarios to explain how m⁶As can affect HIV-1 RNA splicing and unspliced RNA production, as follows:

Discussion

NEW: (Lines 346-363)

“The connection between m⁶As and RNA splicing is also intricate and likely context-dependent. The impact of m⁶As on individual genes seems to be heterogeneous in whole-transcriptome studies⁹²⁻⁹⁴. The timing of m⁶A deposition (occurring before or after splicing) is key to understanding the connection, but it appears to be complex for cellular RNAs^{64-66, 80, 83, 93-95}. Notably, gene-specific or virus-specific investigations have established clearer links between m⁶As and alternative splicing, involving m⁶A writers⁹⁶⁻⁹⁹, readers (YTHDC1)^{16, 17, 100}, and erasers^{15, 101, 102}, as well as interactions with splicing regulatory elements^{20, 21, 23, 24, 92, 103}. In HIV-1, the impact of YTHDC1 knockdown on alternative splicing remained partially controversial in two recent studies^{16, 17}. G-quadruplexes (G4s), co-localized with the three major m⁶A sites,

might also influence alternative splicing¹⁰⁴⁻¹⁰⁸. m⁶A-G4 co-localization has been reported for other viruses¹⁰⁹, but its role in viral RNA regulation remains unclear.

Besides the direct involvement through cellular splicing regulators, the three m⁶As might indirectly affect alternative splicing considering that the production of partially spliced and unspliced HIV-1 RNAs depends on the production of viral regulatory protein (Tat and Rev)^{51, 79}. m⁶A-mediated translational enhancement of these regulatory proteins, particularly rev, could significantly affect the production of HIV-1 unspliced RNA^{51, 79, 110}. In-depth mechanistic investigations into the roles of the site-specific m⁶As are warranted to better understand the balance between viral self-regulation and host-regulation.”

Reviewer 2 comment 7d.

7d. How would methylations in the 3' alter translation of viral RNAs - can the authors speculate?

Author Response to Reviewer 2 comment 7d.

m⁶As located in the 3' end UTR have been demonstrated to promote RNA translation by binding to cytoplasmic m⁶A readers, forming a closed mRNA loop structure that connects the 5' and 3' ends (Shi et al., 2017; Wang et al., 2015). However, the involvement of m⁶A readers in this process has been a topic of controversy (Lasman et al., 2020; Shi et al., 2017; Wang et al., 2015; Zaccara and Jaffrey, 2020). It's worth noting that while m⁶As in the UTR generally enhance translation, those in coding regions have been shown to negatively affect translation (Choi et al., 2016; Qi et al., 2016; Slobodin et al., 2017). We provide the following paragraph to discuss the potential roles of the m⁶As in HIV-1 RNA stability and translation:

Discussion

NEW: Lines 333-345

“The three site-specific m⁶As also exhibit functional features partially distinct from m⁶As in cellular RNAs⁶⁷. Recent studies have revealed that cytoplasmic m⁶A readers, YTHDF1-3, share their binding sites on cellular RNAs and facilitate RNA degradation^{84, 85}. HIV-1 also exhibited shared binding sites among YTHDF1-3⁶, but unlike these reports, we did not find any notable impact of the three m⁶As on RNA stability; the total HIV-1 RNA copies remained similar in both WT- and Triple-mutant-producing cells. Instead, triple mutation affected the translation efficiency of viral mRNAs, resulting in significantly lower viral protein levels. m⁶As in UTRs have been reported to stimulate translation for cellular mRNAs⁸⁶⁻⁹⁰, although the precise mechanisms remain somewhat controversial^{84-86, 89}. m⁶As in the coding regions, however, have been shown to negatively affect translation^{81, 90, 91}. The impact of m⁶As on HIV-1 RNA stability has yielded mixed results in previous cellular perturbation and qPCR-based studies^{6, 9, 17}. Our study

contributes new insights into this area by directly examining the knockouts of HIV-1-specific m⁶As and quantifies native RNA molecules individually using Nanopore DRS.”

Reviewer 2 comment 8.

8. Raw data (excel tables) should be provided for the each of the figures, so that they can be reproduced independently. This is especially important for the results of the modification pipelines. Nature Micro should insist on this point.

Author Response to Reviewer 2 comment 8.

Raw data for each figure are provided in **Supplement table 6**.

Code availability (see Lines 723-725)

The source code for m⁶Arp modeling and analyses is available at <https://github.com/ksanggu/m6Arp>

Data availability (see Lines 718-721)

All data supporting the findings of this study are available within the paper and its Supplementary Information. The Nanopore sequencing data used in this study were deposited into the European Nucleotide Archive (ENA) with an accession number PRJEB61077.

Reviewer 2 comment 9.

9. I did not appreciate the difference between Fig 4d and Extended Data Fig 7d. In Extended Data Fig 7d the mutations do not affect relative splice acceptor usage, but in Fig 4d(v) describes many changes. Can the authors clarify this point?

Author Response to Reviewer 2 comment 9.

Fig 4d(-v-) presents general donor and acceptor site usage rates relative to those of WT. Whereas, Extended Data Fig 7d (currently, Extended Data Fig 7d-iii-) presents the acceptor usage rates immediately following the D1 donor usage, showing which acceptors were used during the first splicing using D1.

To clarify this, the heatmap views of Fig. 4d-v- are replotted into bar-graphs (see new Extended Data Fig. 7) as follows:

Extended Data Figure 7d.

Extended Data Fig. 7d legend:

PREVIOUS: (Lines 710-712)

“(d) Relative usage rates of splicing acceptors (% , y-axis) are shown for WT (n=4 distinct samples), triple mutants (n= 3 distinct samples) and single mutants (A8079G, A8975G, A8989T, and A8110G).”

New (Lines 833-839)

“(d) splicing donor and acceptor site usage. (i-ii) bar-graphs showing the relative usage rates per HIV-1 RNA (US+PS+CS combined; y-axis) for Splicing Donor usage (i) and Splicing Acceptor usage (ii). (iii) First donor sites. Nearly all (93.5%-94.6%) spliced RNA uses D1 donor; 5.3% to 6.3% use D1c; and less than 0.5% use other donors (D2, D3 or D4) for the first splicing. (iv) The acceptor usage rates following the D1 donor usage (% of D1, y-axis) during the first splicing event. WT (n=4 distinct samples), triple mutants (n= 3 distinct samples) and single mutants (A8079G, A8975G, A8989T; n=3 samples) are shown.”

Reviewer 2 minor comment 1.

- Title: 'Single-RNA-level analysis of full-length HIV-1 RNAs' - would 'Single-molecule analysis' read better?

Author Response to Reviewer 2 minor comment 1.

We modified our title according to the suggestion.

Title

PREVIOUS: (Line 1)

“Single-RNA-level analysis of full-length HIV-1 RNAs reveals functional redundancy of m6As”

Now: (Line 1)

“**Single-molecule** analysis of full-length HIV-1 RNAs reveals functional redundancy of m6As”

Reviewer 2 minor comment 2.

- Fig 1 could be updated to show the locations all DRACH motifs and spliced-junctions for discussion.

Author Response to Reviewer 2 minor comment 2.

We updated Figure 1 with DRACH motifs and splice-junction positions.

Reviewer 2 minor comment 3.

- p1, l34 of summary: 'More densely installed in viral mRNAs than in genomic RNAs' genomic RNA is technically an mRNA. Better to use 'More densely installed in spliced viral mRNA than in genomic RNAs'

Author Response to Reviewer 2 minor comment 3.

We modified our manuscript accordingly as follows:

The Summary section:

PREVIOUS: (Lines 34-35)

“More densely installed in viral mRNAs than in genomic RNAs, these m⁶As play a crucial role in maintaining normal levels of RNA splicing and translation.”

New : (Lines 34-36)

*“More densely installed in **spliced** viral mRNAs than in genomic RNAs, these m⁶As play a crucial role in maintaining normal levels of RNA splicing and translation.”*

Reviewer 2 minor comment 4.

- Supplementary note, p1, l39. Correct 1973 to 9173

Author Response to Reviewer 2 minor comment 4.

Thank you. The typo is corrected accordingly.

Reviewer 2 minor comment 5.

- I546. Which reference? Supply genbank id or similar

Author Response to Reviewer 2 minor comment 5.

HIV-1 genome reference sequence AF324493.2 was used. The manuscript is modified as follows:

Methods

PREVIOUS: (Lines 544-546)

“The sequence read length was extracted by aligning sequences against the HIV-1 ~~transcriptome~~ reference using minimap2 (v.2.24), retaining multiple secondary alignments (parameters -p 0 -N 10) and counting the number of unique read IDs among mapped alignments.”

New : (Lines 669-672)

*“The sequence read length was extracted by aligning sequences against the **HIV-1 coding sequences retrieved from HIV-1 genome reference sequence AF324493.2 from NCBI** using minimap2 (v.2.24), retaining multiple secondary alignments (parameters -p 0 -N 10) and counting the number of unique read IDs among mapped alignments.”*

Reviewer 2 minor comment 6.

- I563. Which aligner is used?

Author Response to Reviewer 2 minor comment 6.

minimap2 (v.2.24) is used. The manuscript is modified as follows:

Methods

PREVIOUS: (Lines 563-564)

“Reads were aligned against the HIV genome AF324493.2 from the NCBI in spliced mapping mode using a kmer size of 14.”

NEW : (Lines 688-689)

*“Reads were aligned **using minimap2 (v.2.24)** against the HIV genome AF324493.2 from the NCBI in spliced mapping mode using a kmer size of 14.”*Reviewer #3 comment 1

Major comments:

1. I agreed with the authors that they are reporting a technical advancement in HIV transcript sequencing, and that they have observed redundancy in m6A sites. However, the authors used “bet-hedging” to explain their observations in both the abstract and in the manuscript main text. In my understanding, bet-hedging means that a genotype was selected despite decreased fitness but the genotype would allow the organism in exchange for increased fitness in stressful conditions (in this case, it would be the high mutation rate of HIV leading to potential loss of any one of the m6A sites). However, in this specific study, the authors did not directly show that redundancy was associated with decreased fitness, and instead referenced previous publications (their citations 9 and 15) and extrapolated that there may be a decrease in fitness. I understand this hypothesis cannot be easily tested in this specific context, so my suggestion, if I have understood the context correctly, would be to tone down the focus on the word bet-hedging and instead focus on discussing the redundancy that they observed.

Response to Reviewer #3 comment 1

While a diverse range of bet-hedging strategies has been described, identifying the traits of bet-hedging remains challenging due to conceptual and practical difficulties in examining these traits (Simons, 2011), and we agree that additional experiments are needed to clearly link our finding to Bet-hedging.

We modified our manuscript to tone-down the focus on the word bet-hedging and instead focus on more clearly addressing the functional redundancy. Our response to this comment is largely identical to **Author Response to Reviewer 1 comment #3** and **Author Response to Reviewer 2 comment 1** above.

Abstract

Before: (Lines 38-41)

“Our ~~single-RNA-level~~ study demonstrates that HIV-1 tolerates functionally redundant m⁶As to provide stability and resilience to viral replication while minimizing ~~the risk of unpredictable mutagenesis~~—a novel RNA-level strategy similar to bet-hedging in evolutionary biology.”

New: (Lines 38-41)

*“Our **single-molecule-level** study demonstrates that HIV-1 tolerates functionally redundant m⁶As to provide stability and resilience to viral replication while minimizing **the risk of losing m⁶As due to random mutations** – potentially a novel RNA-level evolutionary strategy for HIV-1”*

Results

PREVIOUS: (Lines 251-270)

“Given the functional redundancy of these m⁶As, we then asked why HIV-1 maintains excessive m⁶As on its RNAs. First, having multiple m⁶As may be necessary to sustain normal levels of viral replication. In such a scenario, our single mutations would show either a selection for RNAs with multiple m⁶As or an increase in m⁶A stoichiometry at intact DRACH sites to compensate for the loss of a dominant m⁶A. However, we found indistinguishable or only moderate differences in m⁶A stoichiometry (Fig. 6b,c) and m⁶A landscape (Extended Data Fig. 10a) between the single mutants and the WT HIV-1. Like WT RNAs, all RNA subspecies of single mutants, regardless of m⁶A status, showed no significant bias toward the utilization of certain splicing donors or acceptors (Extended Data Fig. 10b-d).

Given that there is no clear evidence of adaptive selection or changes in m⁶A functions, a more favorable scenario may be that HIV-1 spreads the risk of losing m⁶A functions⁶⁴. HIV-1 may tolerate these multiple redundant m⁶As to minimize the risk of unpredictable random mutagenesis (10⁻⁵ to 10⁻³ mutations/bp/cycle for HIV-1⁶²) that may knockout a major m⁶As, analogous to bet-hedging in evolutionary biology reducing risks against unpredictably fluctuating environments⁶⁴. Interestingly, all the single mutants maintained at least one of the three m⁶As in most of their CS (91.5-95.8%) and PS (89.1%-95.0%) RNAs, levels comparable to the WT HIV-1 (Fig. 6d). Despite causing substantial loss of m⁶A at the population level, single mutations had only a marginal effect on overall viral fitness. The loss of all three, however, eroded the potential of RNA communities to sustain their control over splicing and translation, and adversely affected the various stages of the HIV-1 life cycle (Fig. 3).”

NEW: (Lines 266-292)

“Given the functional redundancy of these m⁶As, we then asked why HIV-1 maintains excessive m⁶As on its RNAs. To explore potential additive or synergistic effects of the three major m⁶As, we hypothesized that having multiple of these m⁶As on the majority of individual RNA molecules (‘subspecies A-D’ in Fig. 5c) is vital for HIV-1 to maintain normal levels of viral replication. Given that all single mutants exhibited no significant reduction in most of their replication stages (Fig. 3), we investigated whether (i) the single mutants deposit new m⁶As at other DRACH sites in response to the loss of a dominant m⁶A and/or (ii) selectively enrich multiple-m⁶A-containing RNAs in their RNA pool. However, we found indistinguishable or only moderate differences in the m⁶A stoichiometry at each of the three major m⁶A sites (Fig. 6b, c) and m⁶A landscape (Extended Data Fig. 10a) between the single mutants and the WT HIV-1. These results suggest no significant additive effects of the three m⁶As on viral replication, except for a moderate increase in alternative splicing in the single mutants.

Our read-level splicing analysis also showed that all RNA subspecies of single mutants exhibited no significant bias toward the utilization of specific splicing donors or acceptors (Extended Data Fig. 10b-d), similar to what was observed in WT (Fig. 6a-v), suggesting no functional changes of the three m⁶As in single mutants.

In the context of RNA-population-level evolutionary responses, our data suggest that the functional redundancy of m⁶As on the HIV-1 RNA best aligns with the bet-hedging mode among the three core modes of evolutionary response, including adaptive tracking, plasticity in phenotype (or function), and bet-hedging⁶⁰. HIV-1 may tolerate these multiple redundant m⁶As to minimize the risk of unpredictable random mutagenesis (10⁻⁵ to 10⁻³ mutations/bp/cycle for HIV-1⁶¹) that may knockout a major m⁶As. Interestingly, all the single mutants maintained at least one of the three m⁶As in most of their CS (87.7-94.7%) and PS (81.9-93.9%) RNAs, levels comparable to the WT HIV-1 (Fig. 6d). Despite causing substantial loss of m⁶A at the population level, single mutations had only a marginal effect on overall viral fitness. The loss of all three, however, eroded the potential of RNA communities to sustain their control over splicing and translation, and adversely affected the various stages of the HIV-1 life cycle (Fig. 3)”

Reviewer #3 comment 2

2. All experiments were done using 293T cells, which is not a T cell line. All the viral transcripts also seemed to be isolated after transfection of the 293T cells. To increase biological relevance of the study, I think the authors can consider validating their triple mutant findings by infecting primary CD4+ T cells or at least a T cell line with their viral stocks.

Response to Reviewer #3 comment 2

New T-cell infection data are provided. The RNA modification landscape in T cells is nearly identical to that of HEK293T cells after transfection (Extended Data Figure 11d). T-cell infection with triple mutant resulted in a significant reduction of US RNA compared to WT-infected cells at 96 hour post infection (Extended Data Figure 6b-ii-. The manuscript is modified as follows:

Results 1.

New: (Line 173)

“HIV-1-infected CD4+ T cells (Jurkat) also showed similar reduction of US RNA (Extended Data Fig.6b).”

Results 2.

New: (Lines 119-120)

“Similar modification signals were also observed in HIV-1 infected CD4+ T cells (Extended Data Fig. 11d).”

Extended Data Figure 11d (bottom panel)

Figure legend for Extended Data Figure 11d

New: (Lines 913-916)

“(d) Potential Modification Sites in Nanopore DRS Data. In this section, we present potential modification sites detected in Nanopore DRS data for HEK293T cells transfected with pNL4-3 (upper panel) and those for WT-infected CD4+ T cells (CEM-SS) 96 hour post infection (lower panel; result of single cycle infection, see methods for details).”

Extended Data Figure 6b-ii-

(ii) Jurkat cells 96 hpi with WT or Triple mutant viruses

Figure legend for Extended Data Figure 6b-ii-

Before: (Line 689)

“The US and total HIV-1 RNA were measured by digital PCR.”

New: (Lines 815-817)

“The US and total HIV-1 RNA were measured by digital PCR. HEK293T cells were analyzed 72 hour post-transfection with pNL4.3 plasmids (i) and Jurkat T cells were analyzed 96 hour post infection (hpi) with MOI of 1-2 of HIV-1_{NL4.3} (ii).”

Methods:

New: (Lines 643-659)

“Single-cycle infection of CEM-SS cells:

2 X 10⁶ CEM-SS cells were infected with WT or Triple mutant virus at 2 MOI in 1 mL RPMI-1640 with 1% penicillin/streptomycin (P/S) and 10% fetal bovine serum(FBS). Cells were incubated with viruses for 1 h, swirling every 20 min, and then transferred to T25 flask with 10 mL RPMI-1640 (1% P/S & 10% FBS). At 24 h post-infection, the cells were washed with PBS and the culture medium was exchanged with RPMI-1640 media with drugs (1% P/S, 10% FBS, 100nM T20, and 100nM IDV). At 96 h post-infection, the single-cycle infected cells were collected and total cellular RNAs were extracted with TRI Reagent (Sigma-aldrich, T9424), following manufacturer’s instructions.”

“Jurkat cell infection:

6 X 10⁶ Jurkat cells were infected with WT or Triple mutant virus at 1 MOI (first experiment) or 2 MOI (second and third experiments) in 2 mL RPMI-1640 (1% P/S and 10% FBS) for 1 h, swirling every 20 min. Cells were then transferred to T75 flask, adding RPMI-1640 (1% P/S and 10% FBS) up to 30 mL. At 24 h post-infection, the cells were washed with PBS and the culture medium was exchanged with fresh RPMI-1640 (1% P/S and 10% FBS). At 96 post-infection, total cellular RNAs were extracted with TRI Reagent (Sigma-aldrich, T9424), following manufacturer’s instructions.”

Reviewer #3 comment 3

3. Nanopore direct RNA sequencing has a very high error rate compared to other Nanopore library prep methods and even more so compared to other long read sequencing method such as PacBio. The authors should discuss how sequencing errors were handled.

Author Response to Reviewer #3 comment 3:

We applied “two-step noise removal procedure” to precisely identify RNA modification sites (Supplemental Note 1). Given the majority of errors in DRS are due to the signal alterations by the local context of RNA molecules surrounding the read head – e.g. local nucleotide

48sequences, secondary/tertiary structures and chemical modifications (<https://github.com/nanoporetech/tombo>)(Stoiber et al., 2017), – we first employed the “model-sample-compare” (MSC) option of Tombo using HIV-1 in vitro transcribed (IVT) RNA DRS data. The MSC option reduces HIV-1-RNA specific noises (due to local RNA contexts) by adjusting the expected values of position-specific k-mers using HIV-1 IVT RNA reads. Additionally, we also employed the second “noise-removal” step that further reduces HIV-1-specific noises using HIV-1 IVT subreads (this IVT subread dataset was separately processed as a test sample and not used as IVT canonical control for MSC analysis; see Supplemental Note 1.2.2). We demonstrated that these two-step noise removal procedures are effective for not only for Tombo analysis, but also for other analytic tools that use “pre-trained” control datasets to determine RNA modification, enabling an accurate and reproducible detection of RNA modifications for HIV-1 RNA (Supplemental Note Fig.SN2, SN3, SN5, and SN6).

The two-step noise removal procedures are elaborated in more detail in Supplemental Note 1.

Reviewer #3 Minor comment 1:

1. Line 73. The language here can be modified to make it more concise. “site-specific” and “three distinct locations” are redundant.

Author Response to Reviewer #3 Minor comment 1:

We modified our manuscript accordingly as follows:

Results

PREVIOUS: (Lines 73-75)

“We demonstrate dominant and site-specific m⁶A modifications at three distinct locations on the RNA genome and characterize their functional significance in regulating HIV-1 replication at the individual RNA level.”

NEW: (Lines 72-74)

“We found three dominant and site-specific m⁶A modifications on the RNA genome and characterize their functional significance in regulating HIV-1 replication at the individual RNA level.”

Reviewer #3 Minor comment 2:

2. Line 104. “149, 167, 156” It is unclear why the authors select different cutoff values for each tool.

Author Response to Reviewer #3 Minor comment 2:

We chose these numbers for three reasons:

1. Previous mass-spectrometry studies have estimated approximately 80-200 modifications of various kinds exist per HIV-1 RNA genome (Courtney et al., 2019; McIntyre et al., 2018),
2. Nanopore DRS can potentially detect chemical modifications of various kinds (Begik et al., 2021; Jenjaroenpun et al., 2021; Leger et al., 2021; Stoiber et al., 2017).
3. the top 149, 167, and 156 modification signals for Tombo, Eligos2, and Nanocompore, respectively, are chosen above the typical threshold of determining modification sites for these tools (Jenjaroenpun et al., 2021; Leger et al., 2021; Stoiber et al., 2017) and these numbers are within 80-200 modifications that mass-spectrometry studies estimated.

The details of these approaches can be found in Supplemental Note 1. Additionally, we modified the manuscript as follows:

Results 1

NEW: (Lines 92-94)

“Previous mass-spectrometry studies have estimated approximately 80-200 modifications of various kinds exist per HIV-1 RNA genome^{4, 14}. The locations and the functions of site-specific modifications, however, remain unclear due to the challenges to identify their precise locations.”

Results 2

PREVIOUS: (Lines 101-105)

“To identify ~~the most likely modification sites~~, we compared the modification signals generated by three different tools – Tombo, Eligos2, and Nanocompore¹⁻³, each analyzing different aspects of DRS signals, such as ionic current levels, base-calling error rates, and dwell time⁷. We selected the top 149, 167, and 156 modification signals, respectively, from these tools and cross-compared them (Supplementary Note 1.6).”

NEW: (Lines 104-109)

*“To identify **the most consistent and significant site-specific modifications**, we compared the modification signals generated by three different tools – Tombo, Eligos2, and Nanocompore⁴¹⁻⁴³, each analyzing different aspects of DRS signals, such as ionic current levels, base-calling error rates, and dwell time⁴⁵. **Given these tools can detect various kinds of chemical modifications**, we selected the top 149, 167, and 156 modification signals, respectively, from these tools and cross-compared them (Supplementary Note 1.6).”*

Reviewer #3 Minor comment 3:

3. Line 108. The citation for DRACH consensus should be added

Author Response to Reviewer #3 Minor comment 3:

We added the following citation for the DRACH consensus: Linder et al, 2015, *Nat Methods*.

Results

PREVIOUS: (Lines 107-108)

“Notably, among the seven peaks, five were located at or adjacent to the known m⁶A motifs (DRACH: D = A/G/U, R = A/G, H = A/C/U).”

NEW: (Lines 111-112)

“Notably, among the seven peaks, five were located at or adjacent to the known m⁶A motifs (DRACH: D = A/G/U, R = A/G, H = A/C/U)⁴⁶”

Reviewer #3 Minor comment 4:

4. Line 137. “m⁶A at A8110, however, remains unclear.” Please clarify.

Author Response to Reviewer #3 Minor comment 4:

The status of m⁶A at A8110 remains unclear for several reasons. Firstly, the impact of ALKBH5 treatment on the DRS signal was unclear (Fig. 2a). Additionally, m⁶A methylation at A8110 could not be confirmed by LC-MS/MS, despite multiple attempts. Enriching a sufficient amount of RNA fragments containing A8110 failed for unclear reasons. Read-level quantification assays, including m⁶Anet, and Nanom⁶A, suggest that m⁶A at A8110 is a relatively low-stoichiometry methylation, if it is present at all (Supplementary Note Fig. SN6). To clarify this issue, we modified the manuscript as follows:

Results

PREVIOUS: (Line 137)

“m⁶A at A8110, however, remains unclear.”

NEW: (Lines 149-153)

“Despite multiple attempts, we could not confirm m⁶A methylation at A8110 by LC-MS/MS due to the insufficient enrichment of RNA fragments containing A8110. Read-level quantification assays, including m⁶Anet and Nanom⁶A (Supplementary Note Fig. SN6), suggest that m⁶A at A8110 is a relatively low-stoichiometry methylation, if it is present at all.”

Reviewer #3 Minor comment 5:

5. Fig1b. Please clarify the differences between oligos1, 2, 3, and 4.

Author Response to Reviewer #3 Minor comment 5:

These numbers (oligos1, 2, 3, and 4) indicate repeated experiments. We modified the manuscript as follows:

Figure 1b:

Figure 1 legend:

NEW: (Lines 399-400)

"The rep1-3 (left) and rep1-4 (right) denote repeated experiments using independently prepared samples."

Reviewer #3 Minor comment 6:

6. Fig1e. It was unclear that the logo plots were from the Los Alamos Database (as opposed to experimental data) within the figure legend.

Author Response to Reviewer #3 Minor comment 6:

The figure legend is modified accordingly:

Figure 1(e) legend

PREVIOUS: (Lines 314-315)

“The magnified HIV-1 genome from 7916 to 9172 of HIV-1 genome (NL4-3 strain) and sequence logo plots are shown.”

NEW: (Lines 414-416)

*“The magnified HIV-1 genome from 7916 to 9172 of HIV-1 genome (NL4-3 strain) and sequence logo plots **for circulating HIV-1 (Los Alamos Database; <https://www.hiv.lanl.gov/>)** are shown.”*

Reviewer #3 Minor comment 7:

7. Line 543. The authors mentioned mapping reads to GRCh38.p13 but none of the human transcript results were shown.

Author Response to Reviewer #3 Minor comment 7:

The mapping results were used for Extended Data Fig.7a (to determine the quality of DRS run). We modified the manuscript to clarify this issue as follows:

Methods

PREVIOUS: (Lines 540-542)

“Fastqs were aligned to the HIV-1 genome reference sequence AF324493.2 from the NCBI or the human reference sequence (Human genome assembly GRCh38.p13) with the options “-ax map-ont” using minimap2 (v2.24).”

NEW: (Lines 664-667)

*“Fastqs were aligned to the HIV-1 genome reference sequence AF324493.2 from the NCBI or the human reference sequence (Human genome assembly GRCh38.p13 **for Extended Data Fig.7a**) with the options “-ax map-ont” using minimap2 (v2.24).”*

Reviewer #3 Minor comment 8:

8. Line 575. Did the authors see spliced transcripts that lacked the D1- junction?

Author Response to Reviewer #3 Minor comment 8:

Nearly all (93.5%-94.6%) spliced RNA uses D1 donor; 5.3% to 6.3% use D1c; and the remainder (less than 0.5%) uses mostly D4 (D2 or D3 using RNA was also detected).

We added Extended Figure 7d-iii- to clarify this issue as follow:

Extended Data Figure 7d.

Extended Data Fig. 7d legend:

New (Lines 835-836)

“(iii) First donor sites. Nearly all (93.5%-94.6%) spliced RNA uses D1 donor; 5.3% to 6.3% use D1c; and less than 0.5% use other donors (D2, D3 or D4) for the first splicing.”

Reviewer #3 Minor comment 9:

9. Did the authors look at whether m6A patterns in US transcripts are associated with Cap1G, 2G, 3G as described in Brown Science 2020 etc?

Author Response to Reviewer #3 Minor comment 9:

We were unable to read the 5' end sequence of HIV-1 RNA because of the technical limitation of Nanopore DRS reading 5' end (Ibrahim et al., 2021; Workman et al., 2019).

References

- Begik, O., Lucas, M.C., Pryszcz, L.P., Ramirez, J.M., Medina, R., Milenkovic, I., Cruciani, S., Liu, H., Vieira, H.G.S., Sas-Chen, A., *et al.* (2021). Quantitative profiling of pseudouridylation dynamics in native RNAs with nanopore sequencing. *Nat Biotechnol* 39, 1278-1291.
- Choi, J., Jeong, K.-W., Demirci, H., Chen, J., Petrov, A., Prabhakar, A., O'Leary, S.E., Dominissini, D., Rechavi, G., Soltis, S.M., *et al.* (2016). N6-methyladenosine in mRNA disrupts tRNA selection and translation-elongation dynamics. *Nature Structural & Molecular Biology* 23, 110-115.
- Courtney, D.G., Tsai, K., Bogerd, H.P., Kennedy, E.M., Law, B.A., Emery, A., Swanstrom, R., Holley, C.L., and Cullen, B.R. (2019). Epitranscriptomic Addition of m(5)C to HIV-1 Transcripts Regulates Viral Gene Expression. *Cell Host Microbe* 26, 217-227 e216.

He, P.C., Wei, J., Dou, X., Harada, B.T., Zhang, Z., Ge, R., Liu, C., Zhang, L.S., Yu, X., Wang, S., *et al.* (2023). Exon architecture controls mRNA m(6)A suppression and gene expression. *Science* 379, 677-682.

Jenjaroenpun, P., Wongsurawat, T., Wadley, T.D., Wassenaar, T.M., Liu, J., Dai, Q., Wanchai, V., Akel, N.S., Jamshidi-Parsian, A., Franco, A.T., *et al.* (2021). Decoding the epitranscriptional landscape from native RNA sequences. *Nucleic Acids Res* 49, e7.

Kennedy, E.M., Bogerd, H.P., Kornepati, A.V., Kang, D., Ghoshal, D., Marshall, J.B., Poling, B.C., Tsai, K., Gokhale, N.S., Horner, S.M., *et al.* (2016). Posttranscriptional m(6)A Editing of HIV-1 mRNAs Enhances Viral Gene Expression. *Cell Host Microbe* 19, 675-685.

Kessl, J.J., Kutluay, S.B., Townsend, D., Rebensburg, S., Slaughter, A., Larue, R.C., Shkriabai, N., Bakouche, N., Fuchs, J.R., Bieniasz, P.D., *et al.* (2016). HIV-1 Integrase Binds the Viral RNA Genome and Is Essential during Virion Morphogenesis. *Cell* 166, 1257-1268.e1212.

Kutluay, S.B., Zang, T., Blanco-Melo, D., Powell, C., Jannain, D., Errando, M., and Bieniasz, P.D. (2014). Global changes in the RNA binding specificity of HIV-1 gag regulate virion genesis. *Cell* 159, 1096-1109.

Lasman, L., Krupalnik, V., Viukov, S., Mor, N., Aguilera-Castrejon, A., Schneir, D., Bayerl, J., Mizrahi, O., Peles, S., Tawil, S., *et al.* (2020). Context-dependent functional compensation between Ythdf m(6)A reader proteins. *Genes Dev* 34, 1373-1391.

Leger, A., Amaral, P.P., Pandolfini, L., Capitanchik, C., Capraro, F., Miano, V., Migliori, V., Toolan-Kerr, P., Sideri, T., Enright, A.J., *et al.* (2021). RNA modifications detection by comparative Nanopore direct RNA sequencing. *Nat Commun* 12, 7198.

Lu, M., Zhang, Z., Xue, M., Zhao, B.S., Harder, O., Li, A., Liang, X., Gao, T.Z., Xu, Y., Zhou, J., *et al.* (2020). N(6)-methyladenosine modification enables viral RNA to escape recognition by RNA sensor RIG-I. *Nat Microbiol* 5, 584-598.

McIntyre, W., Netzband, R., Bonenfant, G., Biegel, J.M., Miller, C., Fuchs, G., Henderson, E., Arra, M., Canki, M., Fabris, D., *et al.* (2018). Positive-sense RNA viruses reveal the complexity and dynamics of the cellular and viral epitranscriptomes during infection. *Nucleic Acids Res* 46, 5776-5791.

Murakami, S., and Jaffrey, S.R. (2022). Hidden codes in mRNA: Control of gene expression by m(6)A. *Mol Cell* 82, 2236-2251.

Pereira-Montecinos, C., Toro-Ascuy, D., Ananias-Saez, C., Gaete-Argel, A., Rojas-Fuentes, C., Riquelme-Barrios, S., Rojas-Araya, B., Garcia-de-Gracia, F., Aguilera-Cortes, P., Chnaiderman, J., *et al.* (2022). Epitranscriptomic regulation of HIV-1 full-length RNA packaging. *Nucleic Acids Res* 50, 2302-2318.

Qi, S.T., Ma, J.Y., Wang, Z.B., Guo, L., Hou, Y., and Sun, Q.Y. (2016). N6-Methyladenosine Sequencing Highlights the Involvement of mRNA Methylation in Oocyte Meiotic Maturation and

Embryo Development by Regulating Translation in *Xenopus laevis*. *J Biol Chem* 291, 23020-23026.

Shi, H., Wang, X., Lu, Z., Zhao, B.S., Ma, H., Hsu, P.J., Liu, C., and He, C. (2017). YTHDF3 facilitates translation and decay of N(6)-methyladenosine-modified RNA. *Cell Res* 27, 315-328.

Simons, A.M. (2011). Modes of response to environmental change and the elusive empirical evidence for bet hedging. *Proc Biol Sci* 278, 1601-1609.

Slobodin, B., Han, R., Calderone, V., Vrieling, J., Loayza-Puch, F., Elkon, R., and Agami, R. (2017). Transcription Impacts the Efficiency of mRNA Translation via Co-transcriptional N6-adenosine Methylation. *Cell* 169, 326-337.e312.

Stoiber, M. (2017). De novo Identification of DNA Modifications Enabled by Genome-Guided Nanopore Signal Processing. . bioRxiv.

Tirumuru, N., Zhao, B.S., Lu, W., Lu, Z., He, C., and Wu, L. (2016). N(6)-methyladenosine of HIV-1 RNA regulates viral infection and HIV-1 Gag protein expression. *Elife* 5.

Uzonyi, A., Dierks, D., Nir, R., Kwon, O.S., Toth, U., Barbosa, I., Burel, C., Brandis, A., Rossmannith, W., Le Hir, H., *et al.* (2023). Exclusion of m6A from splice-site proximal regions by the exon junction complex dictates m6A topologies and mRNA stability. *Mol Cell* 83, 237-251.e237.

Wang, X., Zhao, B.S., Roundtree, I.A., Lu, Z., Han, D., Ma, H., Weng, X., Chen, K., Shi, H., and He, C. (2015). N(6)-methyladenosine Modulates Messenger RNA Translation Efficiency. *Cell* 161, 1388-1399.

Xu, C., Liu, K., Tempel, W., Demetriades, M., Aik, W., Schofield, C.J., and Min, J. (2014). Structures of human ALKBH5 demethylase reveal a unique binding mode for specific single-stranded N6-methyladenosine RNA demethylation. *J Biol Chem* 289, 17299-17311.

Yang, X., Triboulet, R., Liu, Q., Sendinc, E., and Gregory, R.I. (2022). Exon junction complex shapes the m(6)A epitranscriptome. *Nat Commun* 13, 7904.

Zaccara, S., and Jaffrey, S.R. (2020). A Unified Model for the Function of YTHDF Proteins in Regulating m(6)A-Modified mRNA. *Cell* 181, 1582-1595 e1518.

D'Ascenzo, L., Popova, A.M., Abernathy, S., Sheng, K., Limbach, P.A., and Williamson, J.R. (2022). Pytheas: a software package for the automated analysis of RNA sequences and modifications via tandem mass spectrometry. *Nat Commun* 13, 2424.

Ibrahim, F., Oppelt, J., Maragkakis, M., and Mourelatos, Z. (2021). TERA-Seq: true end-to-end sequencing of native RNA molecules for transcriptome characterization. *Nucleic Acids Res* 49, e115.

Workman, R.E., Tang, A.D., Tang, P.S., Jain, M., Tyson, J.R., Razaghi, R., Zuzarte, P.C., Gilpatrick, T., Payne, A., Quick, J., *et al.* (2019). Nanopore native RNA sequencing of a human poly(A) transcriptome. *Nat Methods* 16, 1297-1305.

Lavender, C.A., Gorelick, R.J., and Weeks, K.M. (2015). Structure-Based Alignment and Consensus Secondary Structures for Three HIV-Related RNA Genomes. *PLoS Comput Biol* 11, e1004230.

Reuter, J.S., and Mathews, D.H. (2010). RNAstructure: software for RNA secondary structure prediction and analysis. *BMC Bioinformatics* 11, 129.

Safari, M., Jayaraman, B., Yang, S., Smith, C., Fernandes, J.D., and Frankel, A.D. (2022). Functional and structural segregation of overlapping helices in HIV-1. *Elife* 11.

Sükösd, Z., Andersen, E.S., Seemann, S.E., Jensen, M.K., Hansen, M., Gorodkin, J., and Kjems, J. (2015). Full-length RNA structure prediction of the HIV-1 genome reveals a conserved core domain. *Nucleic Acids Res* 43, 10168-10179.

Watts, J.M., Dang, K.K., Gorelick, R.J., Leonard, C.W., Bess, J.W., Jr., Swanstrom, R., Burch, C.L., and Weeks, K.M. (2009). Architecture and secondary structure of an entire HIV-1 RNA genome. *Nature* 460, 711-716.

Wilkinson, K.A., Gorelick, R.J., Vasa, S.M., Guex, N., Rein, A., Mathews, D.H., Giddings, M.C., and Weeks, K.M. (2008). High-throughput SHAPE analysis reveals structures in HIV-1 genomic RNA strongly conserved across distinct biological states. *PLoS Biol* 6, e96.

Decision Letter, first revision:

Message: 22nd November 2023

Dear Sanggu,

Thank you for your patience while your manuscript "Single-molecule analysis of full-length HIV-1 RNAs reveals functional redundancy of m6As" was under peer-review at Nature Microbiology. It has now been seen by 2 referees, whose expertise and comments you will find at the end of this email. Unfortunately, reviewer #1 was not available at this time, thus we asked reviewer #2 to assess whether these comments were addressed satisfactorily. Although they find your work of some potential interest, they have raised a number of concerns that will need to be addressed before we can consider publication of the work in Nature Microbiology.

57In particular, reviewer #2 feels that performing experiments with the control mutants, as suggested by reviewer #1, are required. In addition, please remove the term functional redundancy and use an alternative wording, as suggested by both reviewers.

Should further experimental data allow you to address these criticisms, we would be happy to look at a revised manuscript.

Please include a data availability statement as a separate section after Methods but before references, under the heading "Data Availability". This section should inform readers about the availability of the data used to support the conclusions of your study. This information includes accession codes to public repositories (data banks for protein, DNA or RNA sequences, microarray, proteomics data etc...), references to source data published alongside the paper, unique identifiers such as URLs to data repository entries, or data set DOIs, and any other statement about data availability. At a minimum, you should include the following statement: "The data that support the findings of this study are available from the corresponding author upon request", mentioning any restrictions on availability. If DOIs are provided, we also strongly encourage including these in the Reference list (authors, title, publisher (repository name), identifier, year). For more guidance on how to write this section please see: <http://www.nature.com/authors/policies/data/data-availability-statements-data-citations.pdf>

* If you have not done so already we suggest that you begin to revise your manuscript so that it conforms to our Article format instructions at <http://www.nature.com/nmicrobiol/info/final-submission>. Refer also to any guidelines provided in this letter.

When submitting the revised version of your manuscript, please pay close attention to our [href="https://www.nature.com/nature-portfolio/editorial-policies/image-integrity">Digital Image Integrity Guidelines](https://www.nature.com/nature-portfolio/editorial-policies/image-integrity). and to the following points below:

Note: This url links to your confidential homepage and associated information about manuscripts you may have submitted or be reviewing for us. If you wish to forward this e-mail to co-authors, please delete this link to your homepage first.

Nature Microbiology is committed to improving transparency in authorship. As part of our efforts in this direction, we are now requesting that all authors identified as 'corresponding author' on published papers create and link their Open Researcher and Contributor Identifier (ORCID) with their account on the Manuscript Tracking System (MTS), prior to acceptance. This applies to primary research papers only. ORCID helps the scientific community achieve unambiguous attribution of all scholarly contributions. You can create and link your ORCID from the home page of the MTS by clicking on 'Modify my Springer Nature account'. For more information please visit please visit www.springernature.com/orcid.

If you wish to submit a suitably revised manuscript we would hope to receive it within 6 months. If you cannot send it within this time, please let us know. We will be happy to consider your revision, even if a similar study has been accepted for publication at Nature Microbiology or published elsewhere (up to a maximum of 6 months).

Yours sincerely,

Reviewer Expertise:

Referee #1: withdrawn
Referee #2: HIV, RNA biology
Referee #3: HIV

Reviewer Comments:

Reviewer #2 (Remarks to the Author):

#Reviewer 2

In this revised manuscript, Back et al provide additional data and discussion to support their claims of a functional role for m6A modifications in HIV-1.

This was already an excellent manuscript and the revision is further improved. Even though the mechanism is unclear, this study provides an important insight for the HIV-1 field to follow up. I enjoyed the updated discussion, and the authors raise an additional point that some of the effects on unspliced RNA may be due to indirect effects on Rev expression. This could indeed explain some of seemingly conflicting data, e.g. why m6A is required for US production even though it is depleted on US RNA.

--

Minor recommendations

- The authors provide additional replicates in Fig 4d. Whilst the new results are n.s., the data trends still suggest additive, rather than functionally redundant effect. Also, functional differences in the single mutants are seen in Fig 3.

This point is only important because the authors strongly claim 'functional redundancy' (in title). I would not be surprised to see replication defects of the single mutants under more stringent assays e.g. using competition assays. Given that these experiments are beyond the scope of the current manuscript, one cannot say. Nevertheless, the authors may 'future proof' their conclusions by caveating some critical statements in the manuscript.

e.g. l297, "Our analysis revealed that HIV-1 maintains functionally redundant m6As at a few positions near the 3' end of its RNAs to control splicing and translation" could be

60

caveated with “although we cannot exclude the possibility that single mutants may exhibit phenotypes under more stringent assay conditions”

e.g. I369, “Additionally, single mutants did not exhibit any notable differences in various features of viral RNA, including m6A methylation, alternative splicing, 3' polyadenylation, and translation of viral RNAs.” - this is not strictly true as some single mutants did show significant effects in Fig 3. I recommend to tone this down.

The title could be altered to “Single-RNA-level analysis of full-length HIV-1 RNAs reveals functional *role* of m6As”

Other minor

I360, “particularly rev” should be Rev, as the authors refer to a protein

--

Reviewer 1

I was asked to comment on response to Reviewer 1, due to their unavailability.

The authors have partially addressed several concerns raised, but an important point remains.

- The authors were asked how their dRNA-seq analysis matches with existing short read data sets. They provide Extended Data Fig 11 showing the results of dRNA and antibody short read sequencing largely overlap, and the new figure highlights the resolution of nanopore analysis over antibody based approaches.

- The authors were requested to provide additional controls. In the rebuttal, the authors argue that the experiments may not work out as intended due to the complexities of RNA folding and function. Whilst this may be true, it is not a good reason to ignore testing the requested mutants. After all, the same could be said for the A8079G, A8975C, and A8989T mutants, and one would want to rule out that the effects are due to changes in protein sequence (or G quadruplex formation)

The authors should show that mutations disrupting the DRACH motif at least leads to loss of the methylation signal in dRNA-seq, along with the expected effects for at least some of the mutants. The reviewer also suggested to test additional mutations outside of the DRACH motif, which is also a good idea. Alternatively, the authors may consider mutating the DRACH motif whilst maintaining the DRACH context to assess whether the changes are specific for the methylation.

- The authors were questioned on their use of 'functional redundancy'. Indeed, all three

61

reviewers independently picked up on this point so it seems important to address. The recommendation for the authors are the same as above i.e. to continue to tone down / caveat some of the conclusions.

Reviewer #3 (Remarks to the Author):

I really appreciate that the authors went above and beyond to address the comments. Their conclusion is currently appropriate given the data. It is also very encouraging to see that the T cell lines data are consistent with the 293T profiles, supported by a formal statistical test. Again, I think the RNA linearization method (Figure 1a) leading to an increased yield is an important technical advancement and is a major highlight of this manuscript. I support its publication.

Major comment:

- I have previously agreed with the use of the word "redundancy" to describe the authors' observation that the triple mutant, not the singles, reduced US RNA levels, p24, etc. However, in this resubmission, based on reviewer #1's comment, the authors also explored the term "additive effects" which made me re-examine the concept of redundancy. I agree with the authors that the effects are not additive. However, I think indeed redundancy may not be the most appropriate term here. What about "co-dependency"?

Minor comments:

- The authors have clearly explained their rationale for choosing the A8079G mutation (Line 364). Please provide rationale for A8975C and A8989T (why C and T?)
- Coordinates 8079, 8975 and 8989 are currently expressed in NL4-3 coordinates (Line 682-685). Please include the HXB2-equivalent coordinates in the main text.

Author Rebuttal, first revision:

Editorial comments

on 22nd

November 2023

Dear Sanggu,

Thank you for your patience while your manuscript "Single-molecule analysis of full-length HIV-1 RNAs reveals functional redundancy of m6As" was under peer-review at Nature Microbiology. It has now been seen by 2 referees, whose expertise and comments you will find at the end of this

62email. Unfortunately, reviewer #1 was not available at this time, thus we asked reviewer #2 to assess whether these comments were addressed satisfactorily. Although they find your work of some potential interest, they have raised a number of concerns that will need to be addressed before we can consider publication of the work in Nature Microbiology.

In particular, reviewer #2 feels that performing experiments with the control mutants, as suggested by reviewer #1, are required. In addition, please remove the term functional redundancy and use an alternative wording, as suggested by both reviewers.

Author Response to the Editor's comments.

Thank you for the opportunity to address the reviewers' comments on our revised manuscript. We appreciate the supportive comments from the two reviewers, noting, "This was already an excellent manuscript, and the revision is further improved" (reviewer 2), and "I really appreciate that the authors went above and beyond to address the comments" (reviewer 3). We also value their additional suggestions to enhance the manuscript.

Based on the reviewers' suggestions, we have carefully modified our manuscript to refine and "*caveat some critical statements*" regarding the functional redundancy of m6As and to address minor issues and suggestions.

However, we would like to express our reservations about reviewer #2's suggestion to "perform experiments with the control mutants." This was suggested by reviewer #1 as an experimental strategy to address the potential impact of mutagenesis itself, apart from the impact of the removal of m6A methylation. Given the current technical limitations, we believe the suggested control mutations are not suitable for HIV-1 study. Implementing the control mutations is beyond the scope of this manuscript and will not benefit making our major conclusions for the following reasons:

Firstly, predicting or interpreting the phenotype of the suggested mutations in HIV-1 RNA is highly challenging due to the complexity of the 9 Kb HIV-1 RNA, potentially resulting in various compounding effects. The whole purpose of a control experiment is to establish a benchmark for comparing our experimental results, and this mutation strategy may not be suitable for that purpose.

Secondly, the m6A DRACH motifs have already been well-characterized and cross-validated by multiple groups. Characterizing DRACH motifs on HIV-1 RNA is beyond the scope of this manuscript. Additionally, we have demonstrated that mutating the m6A DRACH motifs

abolishes m6A deposition on HIV-1 RNA using DRS and multiple software tools proven effective (Fig.2b).

Most importantly, we were asked to address the potential impact of mutagenesis on the phenotype that we observed with the triple mutant. However, please note that our single mutations (A8079G, A8975C, and A8989T) barely showed any phenotype when tested individually. Only the triple mutation exhibited a significant phenotype. Therefore, each of our single mutants by itself serves as a background control to assess the phenotype of the triple mutant. Whereas the suggested control mutation strategy (designed to examine the three sites individually) may generate more problems than solutions. Moreover, the suggested strategies are rather an indirect assessment, mutating nearby nucleotides that may cause a phenotype not directly related to the specific m6A sites (A8079, A8975, and A8989).

While we express our reservation on the control mutations, we believe we have thoroughly and reasonably addressed the main issue raised by reviewer #1 (addressing the impact of mutagenesis itself). We have provided additional triplicated DRS data to strengthen our background controls (refer to the updated data in Figures 4, 5, 6, Extended Data Figures 8, and 10) and added a new discussion paragraph in the revised manuscript (see Lines 366-378 in the Discussion). Furthermore, to alleviate any confusion that prompted this comment from reviewer #1, we have included two new discussion paragraphs addressing potential mechanisms by which m6As may affect HIV-1 RNA splicing (see Lines 348-365 in the Discussion). The original comment from reviewer #1 can be found below.

In this revised response, we have provided more explicit and specific explanations on these issues. Implementing the control mutations may cause more confusion and a significant delay without adding any benefit to our manuscript. We believe these responses convey our perspective more clearly on these matters and help facilitate the timely publication of our findings.

Sincerely,

Sanggu Kim, PhD

“Reviewer 1 comment 2

The genetic disruption of methylation sites - individually and combined - is a great experiment. Nonetheless, the point-mutation experiments could, in principle, also impact aspects other than the methylated site (e.g. an RNA-binding protein binding site). To more conclusively be able to

64establish that it is indeed the absence of m6A that underlies the phenotype rather than an unrelated consequence of disrupting the sequence, in our view additional mutants disrupting m6A (e.g. by disrupting the 'C' downstream of the methylated adenosine) would be required. In addition, control mutants, impacting the same general region (but not impacting m6A) would help in establishing the specificity of the phenotype. This question is all the more pertinent as (1) no mechanism is proposed for how presence of m6A impacts splicing, and (2) in mammalian cells in general, m6A is by and large not thought to impact splicing, as shown by different studies that acutely inhibited METTL3 and saw only very limited impact on splicing. Needless to say, working out the mechanism may be beyond the scope of this study, but given the absence of such a mechanism in my view additional evidence would be required here for supporting this proposed causal relationship.

Reviewer Comments:

Reviewer #2 (Remarks to the Author):

#Reviewer 2

In this revised manuscript, Back et al provide additional data and discussion to support their claims of a functional role for m6A modifications in HIV-1.

This was already an excellent manuscript and the revision is further improved. Even though the mechanism is unclear, this study provides an important insight for the HIV-1 field to follow up. I enjoyed the updated discussion, and the authors raise an additional point that some of the effects on unspliced RNA may be due to indirect effects on Rev expression. This could indeed explain some of seemingly conflicting data, e.g. why m6A is required for US production even though it is depleted on US RNA.

—

Reviewer 2 comment #1

Minor recommendations

- The authors provide additional replicates in Fig 4d. Whilst the new results are n.s., the data trends still suggest additive, rather than functionally redundant effect. Also, functional differences in the single mutants are seen in Fig 3.

This point is only important because the authors strongly claim ‘functional redundancy’ (in title). I would not be surprised to see replication defects of the single mutants under more stringent assays e.g. using competition assays. Given that these experiments are beyond the scope of the current manuscript, one cannot say. Nevertheless, the authors may ‘future proof’ their conclusions by caveating some critical statements in the manuscript.

e.g. I297, “Our analysis revealed that HIV-1 maintains functionally redundant m6As at a few positions near the 3’ end of its RNAs to control splicing and translation” could be caveated with “although we cannot exclude the possibility that single mutants may exhibit phenotypes under more stringent assay conditions”

66e.g. I369, “Additionally, single mutants did not exhibit any notable differences in various features of viral RNA, including m6A methylation, alternative splicing, 3’ polyadenylation, and translation of viral RNAs.” - this is not strictly true as some single mutants did show significant effects in Fig 3. I recommend to tone this down.

The title could be altered to “Single-RNA-level analysis of full-length HIV-1 RNAs reveals functional *role* of m6As”

Author Response to Reviewer 2 comment #1

Accordingly, we have modified our manuscript to tone down and caveat some of the conclusions for the functional redundancy of m6As as follows:

The following statement is included in the **Discussion**

New (Lines 301-302):

“We cannot, however, exclude the possibility that single mutants may exhibit phenotypes under more stringent assay conditions.”

The following statement is modified as follows:

Previous (Lines 369-371):

“Additionally, single mutants did not exhibit any notable differences in various features of viral RNA, including m⁶A methylation, alternative splicing, 3’ polyadenylation, and translation of viral RNAs”

New (Lines 369-371):

“Additionally, single mutants exhibited insignificant or only marginal differences in various features of their RNA, including m⁶A methylation, alternative splicing, 3’ polyadenylation, and translation of viral RNAs”

The title

Previous (Line 1):

“Single-molecule analysis of full-length HIV-1 RNAs reveals functional ~~redundancy~~ of m⁶As“

New (Line 1):

*“Single-molecule analysis of full-length HIV-1 RNAs reveals functional **roles of site-specific m⁶As**”*

Reviewer 2 comment #2

Other minor

I360, “particularly *rev*” should be *Rev*, as the authors refer to a protein

Author Response to Reviewer 2 comment #1

Thank you. We modified it accordingly:

Previous (Line 360)

“particularly ~~*rev*~~”

New (Line 362)

“particularly ***Rev***”

Reviewer 1

I (Reviewer 2) was asked to comment on response to Reviewer 1, due to their unavailability.

The authors have partially addressed several concerns raised, but an important point remains.

Comment #1*

- The authors were requested to provide additional controls. In the rebuttal, the authors argue that the experiments may not work out as intended due to the complexities of RNA folding and function. Whilst this may be true, it is not a good reason to ignore testing the requested mutants. After all, the same could be said for the A8079G, A8975C, and A8989T mutants, and one would want to rule out that the effects are due to changes in protein sequence (or G quadruplex formation)

The authors should show that mutations disrupting the DRACH motif at least leads to loss of the methylation signal in dRNA-seq, along with the expected effects for at least some of the mutants. The reviewer also suggested to test additional mutations outside of the DRACH motif, which is also a good idea. Alternatively, the authors may consider mutating the DRACH motif whilst maintaining the DRACH context to assess whether the changes are specific for the methylation.

Author Response to Comment #1*

We were asked to address the potential impact of mutagenesis on the phenotype we observed with triple mutant. We would like to clarify that our single-site mutations (A8079G, A8975C, and A8989T) barely showed any phenotype when tested individually. Only the triple mutation exhibited a significant phenotype. From this aspect, our single mutants themselves serve as useful background controls to assess the phenotype of the triple mutant. Whereas the suggested control mutation strategy (designed to examine the three sites individually) may generate more problems than solutions. Moreover, the suggested strategies are rather an indirect approach, mutating nearby nucleotides that may cause a phenotype not directly related to the specific m6A sites (A8079, A8975, and A8989).

We believe we have thoroughly and adequately addressed the main issues (the potential impact of mutagenesis itself) raised by reviewer 1. Nevertheless, we acknowledge that our key points

69may not have been effectively conveyed in our initial responses. Here, we provide a more explicit and specific explanation of why additional mutation experiments are unsuitable for this study (see **-A-** below) and articulate the reasonableness of our new approaches to address the main issues raised in the original comment (see **-B-** below).

(-A-) The control mutations are not suitable for this study.

The logic behind mutating the 4th nucleotide “C” of DRA*CH is based on the idea that this mutation will abolish methylation (denoted as *) at the third nucleotide “A,” allowing for a comparison with the A8079G, A8975C, or A8989T phenotypes. However, this logic, while useful for simpler RNA molecules with easily predictable phenotypes, presents several challenges when applied to HIV-1 RNA.

First, given the primary goal of a control experiment is to establish a benchmark to compare our experimental results to, control mutations are unsuitable for HIV-1 RNA because predicting or interpreting the phenotypic results of these mutations is difficult. A8079G, A8975C, and A8989T mutations, when individually tested, exhibited insignificant or only marginal phenotype differences; only triple mutations showed a significant phenotype (in most cases). Given the complexity of HIV-1 RNA, it would be extremely difficult to predict the phenotype of reviewer-suggested control mutations or to interpret the results of these mutations. Mutating nearby nucleotides may also cause a phenotype that is not directly related to the specific m6A sites (A8079, A8975, and A8989).

Second, the 4th “C” mutation may not fully remove m6A modification. Unlike the 3rd “A” (DRACH) that is absolutely crucial for m6A deposition, the frequency of C at the 4th position is only 85% necessary [approximately 15% are not C at this position (*Linder et al., 2015*)]. Mutating the C may still allow m6A methylation for a fraction of RNA, further complicating result interpretation, particularly in our single-RNA-level data. Nucleotides at other sites (1st, 2nd, or 5th of the DRACH) are even more flexible.

Thirdly, given the aforementioned issues, control mutation experiments require clear target nucleotides with easily defined phenotypes. However, it is unclear which specific nucleotides need to be selected based on the reviewers’ comments, such as “testing additional mutations outside of the DRACH motif” or “mutating the DRACH motif while maintaining the DRACH context.” It is unclear what specific hypotheses are suggested to test. The m6A DRACH motif has been well characterized and cross-validated by multiple groups (*Chen et al., 2015; Linder et al., 2015; Liu et al., 2014; Uzonyi et al., 2023*). Additional characterization of the DRACH motif on HIV-1 RNA is not the focus of this manuscript and is unlikely to add significant value to this already compact manuscript.

We were also asked to demonstrate “*mutations disrupting the DRACH motif at least leads to loss of the methylation signal in dRNA-seq, along with the expected effects for at least some of the mutants.*” In fact, our Fig.2b address this issue clearly using appropriate controls and multiple analytic tools that were proven effective.

The reviewer-suggested control mutations would require months of focused efforts and significant research resources to prepare multiple mutations and conduct triplicated DRS experiments for each mutation. However, we feel these experiments would not directly strengthen the conclusion of this manuscript. These specific questions may need to be addressed by follow-up studies.

(-B-) Providing a reasonable approach to address the main issue.

In the original comment from the reviewer #1, we were asked to address the potential impact of mutagenesis itself on “aspects other than the methylated site (e.g. an RNA-binding protein binding site)” (see the quote below). Reviewer #1 noted that the phenotype (“over-splicing” by the triple mutant) was confusing because of the conflicting reports on whether m6A affects splicing in the mammalian system, which we have addressed in Lines 348-365 in the Discussion.

“Reviewer 1 comment 2

The genetic disruption of methylation sites - individually and combined - is a great experiment. Nonetheless, the point-mutation experiments could, in principle, also impact aspects other than the methylated site (e.g. an RNA-binding protein binding site). To more conclusively be able to establish that it is indeed the absence of m6A that underlies the phenotype rather than an unrelated consequence of disrupting the sequence, in our view additional mutants disrupting m6A (e.g. by disrupting the ‘C’ downstream of the methylated adenosine) would be required. In addition, control mutants, impacting the same general region (but not impacting m6A) would help in establishing the specificity of the phenotype. This question is all the more pertinent as (1) no mechanism is proposed for how presence of m6A impacts splicing, and (2) in mammalian cells in general, m6A is by and large not thought to impact splicing, as shown by different studies that acutely inhibited METTL3 and saw only very limited impact on splicing. Needless to say, working out the mechanism may be beyond the scope of this study, but given the absence of such a mechanism in my view additional evidence would be required here for supporting this proposed causal relationship.”

Measuring the direct impact of a single mutation (A8079G, A8975C, or A8989T) on HIV-1 RNA-protein or RNA-RNA interactions is highly challenging with currently available techniques (as described in “**Author Response to Reviewer 1 comment #2a**” in our previous rebuttal letter).

Given the importance of RNA-RNA or RNA-protein interactions in regulating RNA post-transcriptional modifications (Fu and Ares, 2014; Kessl et al., 2016; Knoener et al., 2021; Kutluay et al., 2014; Singh et al., 2022), we reasoned that the impact of a nucleotide mutation on RNA-RNA or RNA-protein interactions may be indirectly assessed by evaluating the potential phenotypic changes of RNA molecules, including m⁶A methylation, alternative splicing, 3' polyadenylation, and translation among mutant viruses. To this end, we provided new triplicated DRS data for single mutations, enabling us to statistically conclude the impact of mutations on various HIV-1 RNA features (see the updated data in **Figures 4, 5, 6, Extended Data Figures 8 and 10**).

In addition to RNA-level phenotypes, A8079G is synonymous for Rev. A8079G changes one amino acid of Env at position 771 (Glutamine to Glycine), but this change does not significantly affect Env expression or viral replication (Fig.3). A consistent result was also reported in a previous in vitro test (Safari et al., 2022). A8975C and A8989T are in the 3' untranslated region and did not show a significant reduction in viral fitness when individually tested. See the paragraph in the newly added discussion section (Lines 366-378) below.

“Discussion: (Lines 366-378)

*“The A8079G mutation is in the overlapping region of rev and env genes and is silent for Rev but induces the substitution of Glutamine with Glycine at position 771 in Env. This change induces only a moderate reduction in HIV-1 fitness in vitro¹¹. A8975C and A8989T are located in the 3' UTR of the HIV-1 RNAs. **The predicted local structures near the mutation sites (A8079G, A8975C and A8989T) vary among different studies^{38, 112-114}**. As expected, we found no significant reduction in viral protein production and infectivity between the single mutants and the wild type. Additionally, single mutants **exhibited insignificant or only marginal differences in various features of their RNAs**, including m⁶A methylation, alternative splicing, 3' polyadenylation, and translation of viral RNAs. Given the significance of RNA-RNA or RNA-protein interactions in regulating these features^{74-77, 115}, it appears that these mutations are well-tolerated and unlikely have had significant impacts on the RNA-RNA or RNA-protein interactions that underlie the maturation and function of viral RNAs. **The A8079, A8975, and A8989 positions in HIV-1 NL4-3 is equivalent to A8089, A8985 and A8999 of HIV-1 HBX2 strain, respectively.**”*

We also provided the following statements to discuss how HIV-1 m⁶As may affect alternative splicing (the production of unspliced HIV-1 RNA).

“Discussion: (Lines 348-365)

The connection between m⁶As and RNA splicing is also intricate and likely context-dependent. The impact of m⁶As on individual genes seems to be heterogeneous in whole-transcriptome

studies⁹²⁻⁹⁴. The timing of m⁶A deposition (occurring before or after splicing) is key to understanding the connection, but it appears to be complex for cellular RNAs^{64-66, 80, 83, 93-95}. Notably, gene-specific or virus-specific investigations have established clearer links between m⁶As and alternative splicing, involving m⁶A writers⁹⁶⁻⁹⁹, readers (YTHDC1)^{16, 17, 100}, and erasers^{15, 101, 102}, as well as interactions with splicing regulatory elements^{20, 21, 23, 24, 92, 103}. In HIV-1, the impact of YTHDC1 knockdown on alternative splicing remained partially controversial in two recent studies^{16, 17}. G-quadruplexes (G4s), co-localized with the three major m⁶A sites, might also influence alternative splicing¹⁰⁴⁻¹⁰⁸. m⁶A-G4 co-localization has been reported for other viruses¹⁰⁹, but its role in viral RNA regulation remains unclear.

Besides the direct involvement through cellular splicing regulators, the three m⁶As might indirectly affect alternative splicing considering that the production of partially spliced and unspliced HIV-1 RNAs depends on the production of viral regulatory protein (Tat and Rev)^{51, 79}. m⁶A-mediated translational enhancement of these regulatory proteins, particularly Rev, could significantly affect the production of HIV-1 unspliced RNA^{51, 79, 110}. In-depth mechanistic investigations into the roles of the site-specific m⁶As are warranted to better understand the balance between viral self-regulation and host-regulation.”

Comment #2*

- The authors were questioned on their use of 'functional redundancy'. Indeed, all three reviewers independently picked up on this point so it seems important to address. The recommendation for the authors are the same as above i.e. to continue to tone down / caveat some of the conclusions.

Author Response to Comment #2*

Our response to this comment is essentially identical to “**Author Response to Reviewer 2 comment #1**” above.

Reviewer #3 (Remarks to the Author):

I really appreciate that the authors went above and beyond to address the comments. Their conclusion is currently appropriate given the data. It is also very encouraging to see that the T cell lines data are consistent with the 293T profiles, supported by a formal statistical test. Again, I think the RNA linearization method (Figure 1a) leading to an increased yield is an important technical advancement and is a major highlight of this manuscript. I support its publication.

Reviewer 3 comment #1

Major comment:

- I have previously agreed with the use of the word “redundancy” to describe the authors’ observation that the triple mutant, not the singles, reduced US RNA levels, p24, etc. However, in this resubmission, based on reviewer #1’s comment, the authors also explored the term “additive effects” which made me re-examine the concept of redundancy. I agree with the authors that the effects are not additive. However, I think indeed redundancy may not be the most appropriate term here. What about “co-dependency”?

Author Response to Reviewer 3 comment #1

We agree to tone down and caveat some of the conclusions, but we believe “redundancy” is general enough to describe what we observed in most cases, but the term “co-dependency” is rather unclear to use in this case. Our response to this comment is basically same as in “**Author Response to Reviewer 2 comment #1**” above.

We modified our manuscript as follows:

The Discussion:

The following statement is included (Lines 301-302):

“We cannot, however, exclude the possibility that single mutants may exhibit phenotypes under more stringent assay conditions.”

The following statement is modified:

Previous (Lines 369-371)

“Additionally, ~~single mutants did not exhibit any notable differences in various features of viral RNA, including m⁶A methylation, alternative splicing, 3’ polyadenylation, and translation of viral RNAs~~”

New (Lines 369-371)

*“Additionally, **single mutants exhibited insignificant or only marginal differences in various features of their RNA**, including m⁶A methylation, alternative splicing, 3’ polyadenylation, and translation of viral RNAs”*

The title is modified:

Previous (Line 1)

“~~Single-molecule analysis of full-length HIV-1 RNAs reveals functional redundancy of m⁶As~~”

New (Line 1)

*“**Single-molecule analysis of full-length HIV-1 RNAs reveals functional roles of site-specific m⁶As**”*

Reviewer 3 comment #2

Minor comments:

- The authors have clearly explained their rationale for choosing the A8079G mutation (Line 364). Please provide rationale for A8975C and A8989T (why C and T?)

Author Response to Reviewer 2 comment #2

Given the G-rich local structure near A8975 and A8989, we chose not to mutate them to G. We initially chose A8975C and A8989T to maintain the structures predicted by a minimal free energy structure prediction tool (Reuter and Mathews, 2010). We would like to note that the predicted local structures near the mutation sites (A8079G, A8975C and A8989T) vary among different studies (Lavender et al., 2015; Sükösd et al., 2015; Watts et al., 2009; Wilkinson et al., 2008) and the structures we used to design A8975C and A8989T remain to be validated. A8975C and A8989T mutant viruses replicate normally.

The following statement (**yellow highlighted**; Lines 369-370) is included in the **Discussion**:

75“The A8079G mutation is in the overlapping region of rev and env genes and is silent for Rev but induces the substitution of Glutamine with Glycine at position 771 in Env. This change induces only a moderate reduction in HIV-1 fitness in vitro¹¹¹. A8975C and A8989T are located in the 3' UTR of the HIV-1 RNAs. **The predicted local structures near the mutation sites (A8079G, A8975C and A8989T) vary among different studies^{38, 112-114}.** As expected, we found no significant reduction in viral protein production and infectivity between the single mutants and the wild type. Additionally, single mutants exhibited insignificant or only marginal differences in various features of their RNAs, including m⁶A methylation, alternative splicing, 3' polyadenylation, and translation of viral RNAs. Given the significance of RNA-RNA or RNA-protein interactions in regulating these features^{74-77, 115}, it appears that these mutations are well-tolerated and unlikely have had significant impacts on the RNA-RNA or RNA-protein interactions that underlie the maturation and function of viral RNAs. The A8079, A8975, and A8989 positions in HIV-1 NL4-3 is equivalent to A8089, A8985 and A8999 of HIV-1 HBX2 strain, respectively.”

Reviewer 3 comment #3

- **Coordinates 8079, 8975 and 8989 are currently expressed in NL4-3 coordinates (Line 682-685). Please include the HXB2-equivalent coordinates in the main text.**

Author Response to Reviewer 2 comment #3

Discussion section:

Following statement is included:

New (Lines 377-378)

“The A8079, A8975, and A8989 positions in HIV-1_{NL4-3} is equivalent to A8089, A8985 and A8999 of HIV-1_{HXB2} strain, respectively.”

References

Chen, T., Hao, Y.-J., Zhang, Y., Li, M.-M., Wang, M., Han, W., Wu, Y., Lv, Y., Hao, J., Wang, L., *et al.* (2015). m6A RNA Methylation Is Regulated by MicroRNAs and Promotes Reprogramming to Pluripotency. *Cell Stem Cell* *16*, 289-301.

Fu, X.D., and Ares, M., Jr. (2014). Context-dependent control of alternative splicing by RNA-binding proteins. *Nat Rev Genet* *15*, 689-701.

Kessl, J.J., Kutluay, S.B., Townsend, D., Rebensburg, S., Slaughter, A., Larue, R.C., Shkriabai, N., Bakouche, N., Fuchs, J.R., Bieniasz, P.D., *et al.* (2016). HIV-1 Integrase Binds the Viral RNA Genome and Is Essential during Virion Morphogenesis. *Cell* *166*, 1257-1268.e1212.

Knoener, R., Evans, E., 3rd, Becker, J.T., Scalf, M., Benner, B., Sherer, N.M., and Smith, L.M. (2021). Identification of host proteins differentially associated with HIV-1 RNA splice variants. *Elife* *10*.

Kutluay, S.B., Zang, T., Blanco-Melo, D., Powell, C., Jannain, D., Errando, M., and Bieniasz, P.D. (2014). Global changes in the RNA binding specificity of HIV-1 gag regulate virion genesis. *Cell* *159*, 1096-1109.

Lavender, C.A., Gorelick, R.J., and Weeks, K.M. (2015). Structure-Based Alignment and Consensus Secondary Structures for Three HIV-Related RNA Genomes. *PLoS Comput Biol* *11*, e1004230.

Linder, B., Grozhik, A.V., Olarerin-George, A.O., Meydan, C., Mason, C.E., and Jaffrey, S.R. (2015). Single-nucleotide-resolution mapping of m6A and m6Am throughout the transcriptome. *Nat Methods* *12*, 767-772.

Liu, J., Yue, Y., Han, D., Wang, X., Fu, Y., Zhang, L., Jia, G., Yu, M., Lu, Z., Deng, X., *et al.* (2014). A METTL3-METTL14 complex mediates mammalian nuclear RNA N6-adenosine methylation. *Nat Chem Biol* *10*, 93-95.

Reuter, J.S., and Mathews, D.H. (2010). RNAstructure: software for RNA secondary structure prediction and analysis. *BMC Bioinformatics* *11*, 129.

Safari, M., Jayaraman, B., Yang, S., Smith, C., Fernandes, J.D., and Frankel, A.D. (2022). Functional and structural segregation of overlapping helices in HIV-1. *Elife* *11*.

Singh, G., Seufzer, B., Song, Z., Zucko, D., Heng, X., and Boris-Lawrie, K. (2022). HIV-1 hypermethylated guanosine cap licenses specialized translation unaffected by mTOR. *Proc Natl Acad Sci U S A* *119*.

Sükösd, Z., Andersen, E.S., Seemann, S.E., Jensen, M.K., Hansen, M., Gorodkin, J., and Kjems, J. (2015). Full-length RNA structure prediction of the HIV-1 genome reveals a conserved core domain. *Nucleic Acids Res* *43*, 10168-10179.

Uzonyi, A., Dierks, D., Nir, R., Kwon, O.S., Toth, U., Barbosa, I., Burel, C., Brandis, A., Rossmannith, W., Le Hir, H., *et al.* (2023). Exclusion of m6A from splice-site proximal regions by

the exon junction complex dictates m6A topologies and mRNA stability. *Mol Cell* 83, 237-251.e237.

Watts, J.M., Dang, K.K., Gorelick, R.J., Leonard, C.W., Bess, J.W., Jr., Swanstrom, R., Burch, C.L., and Weeks, K.M. (2009). Architecture and secondary structure of an entire HIV-1 RNA genome. *Nature* 460, 711-716.

Wilkinson, K.A., Gorelick, R.J., Vasa, S.M., Guex, N., Rein, A., Mathews, D.H., Giddings, M.C., and Weeks, K.M. (2008). High-throughput SHAPE analysis reveals structures in HIV-1 genomic RNA strongly conserved across distinct biological states. *PLoS Biol* 6, e96.

Decision Letter, second revision:

Message: Our ref: NMICROBIOL-23030581B

23rd January 2024

Dear Sanggu,

Thank you for your patience as we've prepared the guidelines for final submission of your Nature Microbiology manuscript, "Single-molecule analysis of full-length HIV-1 RNAs reveals functional roles of site-specific m6As" (NMICROBIOL-23030581B). Sorry for my late response to your previous email.

Please carefully follow the step-by-step instructions provided in the attached file, and add a response in each row of the table to indicate the changes that you have made. Please also check and comment on any additional marked-up edits we have proposed within the text. Ensuring that each point is addressed will help to ensure that your revised manuscript can be swiftly handed over to our production team.

Since you mentioned the deadline of 17th Feb, I'd suggest to carefully address the points listed in the attached guidelines and submit your revised paper by 7th Feb latest. This will allow us to have enough time to check whether all the points were revised accordingly, and for you to perhaps change things that might be needed. Please also let me know via email once you resubmitted the paper. Does that sound okay to you?

78When you upload your final materials, please include a point-by-point response to any remaining reviewer comments.

In recognition of the time and expertise our reviewers provide to Nature Microbiology's editorial process, we would like to formally acknowledge their contribution to the external peer review of your manuscript entitled "Single-molecule analysis of full-length HIV-1 RNAs reveals functional roles of site-specific m6As". For those reviewers who give their assent, we will be publishing their names alongside the published article.

Nature Microbiology offers a Transparent Peer Review option for new original research manuscripts submitted after December 1st, 2019. As part of this initiative, we encourage our authors to support increased transparency into the peer review process by agreeing to have the reviewer comments, author rebuttal letters, and editorial decision letters published as a Supplementary item. When you submit your final files please clearly state in your cover letter whether or not you would like to participate in this initiative. Please note that failure to state your preference will result in delays in accepting your manuscript for publication.

Cover suggestions

COVER ARTWORK: We welcome submissions of artwork for consideration for our cover. For more information, please see our guide for cover artwork.

Nature Microbiology has now transitioned to a unified Rights Collection system which will allow our Author Services team to quickly and easily collect the rights and permissions required to publish your work. Approximately 10 days after your paper is formally accepted, you will receive an email in providing you with a link to complete the grant of rights. If your paper is eligible for Open Access, our Author Services team will also be in touch regarding any additional information that may be required to arrange payment for your article.

Please note that *Nature Microbiology* is a Transformative Journal (TJ). Authors may publish their research with us through the traditional subscription access route or make their paper immediately open access through payment of an article-processing charge (APC). Authors

will not be required to make a final decision about access to their article until it has been accepted. Find out more about Transformative Journals

Best regards,

Reviewer #2:

Remarks to the Author:

Whilst the authors have decided against testing additional mutants (arguing that the effects may be difficult to interpret), the authors have been careful to caveat their claims.

I look forward to seeing the current work published so that it can stimulate future mechanistic studies to understand the important phenomenon described therein.

Reviewer #3:

Remarks to the Author:

I only have one minor comment.

In the authors' rebuttal letter, they have clearly addressed the rationale behind the A8975C and A8989T mutations were to "maintain the structures predicted by a minimal free energy structure prediction tool (Reuter and Mathews, 2010)... replicate normally." I think this

80should go into the main text of the manuscript.

Final Decision Letter:

Message 15th February 2024

:

Dear Sanggu,

I am delighted to accept your Article "Single-molecule epitranscriptomic analysis of full-length HIV-1 RNAs reveals functional roles of site-specific m6As" for publication in Nature Microbiology. Thank you for having chosen to submit your work to us and many congratulations.

You may wish to make your media relations office aware of your accepted publication, in case they consider it appropriate to organize some internal or external publicity. Once your paper has been scheduled you will receive an email confirming the publication details. This is normally 3-4 working days in advance of publication. If you need additional notice of the date and time of publication, please let the production team know when you receive the proof of your article to ensure there is sufficient time to coordinate. Further information on our embargo policies can be found here:

<https://www.nature.com/authors/policies/embargo.html>

Acceptance of your manuscript is conditional on all authors' agreement with our publication

81policies (see <https://www.nature.com/nmicrobiol/editorial-policies>). In particular your manuscript must not be published elsewhere.

Please note that *Nature Microbiology* is a Transformative Journal (TJ). Authors may publish their research with us through the traditional subscription access route or make their paper immediately open access through payment of an article-processing charge (APC). Authors will not be required to make a final decision about access to their article until it has been accepted. Find out more about Transformative Journals

To assist our authors in disseminating their research to the broader community, our SharedIt initiative provides you with a unique shareable link that will allow anyone (with or without a subscription) to read the published article. Recipients of the link with a

subscription will also be able to download and print the PDF.

Congrats again to you and your co-authors. I'm looking forward to seeing your paper published!

All the best,